# Robustly Learning Monotone Single-Index Models

**Puqian Wang** *
UW Madison
pwang333@wisc.edu

**Nikos Zarifis** *
UW Madison
zarifis@wisc.edu

**Ilias Diakonikolas**
UW Madison
ilias@cs.wisc.edu

**Jelena Diakonikolas**
UW Madison
jelena@cs.wisc.edu

## Abstract

We consider the basic problem of learning Single-Index Models with respect to the square loss under the Gaussian distribution in the presence of adversarial label noise. Our main contribution is the first computationally efficient algorithm for this learning task, achieving a constant factor approximation, that succeeds for the class of *all* monotone activations with bounded moment of order $2 + \zeta$, for $\zeta > 0$. This class in particular includes all monotone Lipschitz functions and even discontinuous functions like (possibly biased) halfspaces. Prior work for the case of unknown activation either does not attain constant factor approximation or succeeds for a substantially smaller family of activations. The main conceptual novelty of our approach lies in developing an optimization framework that steps outside the boundaries of usual gradient methods and instead identifies a useful vector field to guide the algorithm updates by directly leveraging the problem structure, properties of Gaussian spaces, and regularity of monotone functions.

## 1 Introduction

Single-index models (SIMs) [Ich93, HJS01, HMS$^+$04, DJS08, KS09, KKSK11, DH18] represent a fundamental class of supervised learning models, widely used and studied in machine learning and statistics. The SIM framework captures scenarios where the output value depends solely on a one-dimensional projection of the input, i.e., it contains functions of the form $f(\mathbf{x}) = \sigma(\mathbf{w} \cdot \mathbf{x})$, where $\sigma : \mathbb{R} \to \mathbb{R}$ is an *unknown* activation (or link) function, and $\mathbf{w} \in \mathbb{R}^d$ is an unknown parameter vector. While the activation function is generally unknown, it is often assumed to lie in a "well-behaved" family, e.g., it is monotone and/or Lipschitz. In addition to being well-motivated from the aspect of applications, such regularity assumptions are also necessary for ensuring statistical and computational tractability of the underlying learning task. Indeed, without such assumptions, learning SIMs can be information-theoretically impossible (see, e.g., [ZWDD25]) or computationally intractable [SZB21]—even for Gaussian data. Classical works [KS09, KKSK11] demonstrated that SIMs with monotone and Lipschitz activations can be learned efficiently in the realizable case (i.e., with clean labels) or zero-mean label noise, under any distribution on the unit ball.

In this paper, we consider the task of learning SIMs in the (more challenging) agnostic model [Hau92, KSS94], in which the labels may be arbitrarily corrupted and no structural assumptions are made on the noise. The goal of an agnostic learner for a target class $\mathcal{C}$ is to find a predictor that is competitive with the best function in $\mathcal{C}$. More concretely, let $\mathcal{D}$ denote a distribution over labeled pairs $(\mathbf{x}, y) \in \mathbb{R}^d \times \mathbb{R}$, and let the squared loss of a predictor $f : \mathbb{R}^d \to \mathbb{R}$ be given by $\mathcal{L}_2(f) =$

---

*Equal contribution.

39th Conference on Neural Information Processing Systems (NeurIPS 2025).

$\mathbf{E}_{(\mathbf{x},y)\sim\mathcal{D}}[(f(\mathbf{x})-y)^2]$. Given access to i.i.d. samples from $\mathcal{D}$, the learner aims to output a hypothesis with error close to the minimum loss $\mathrm{OPT} := \inf_{f\in\mathcal{C}}\mathcal{L}_2(f)$ attainable by any function in the class $\mathcal{C}$.

In our context, the target class $\mathcal{C}$ will refer to the class of SIMs with unknown activation and parameter vector, namely functions of the form $f(\mathbf{x}) = \sigma(\mathbf{w}\cdot\mathbf{x})$. The only assumptions we make are that the norm of the parameter (weight) vector is bounded, i.e., $\|\mathbf{w}\|_2 \leq W$ for some $W > 0$, and that the activation function belongs to a known family of structured monotone functions. For a pair $(\mathbf{w},\sigma)$ defining the SIM $f(\mathbf{x}) = \sigma(\mathbf{w}\cdot\mathbf{x})$, we write $\mathcal{L}_2(\mathbf{w};\sigma) = \mathbf{E}_{(\mathbf{x},y)\sim\mathcal{D}}[(\sigma(\mathbf{w}\cdot\mathbf{x})-y)^2]$ to denote the squared error of $f$. We now formally define our learning task.

**Problem 1.1** (Robustly Learning SIMs). *Fix a family $\mathcal{F}$ of univariate activations. Let $\mathcal{D}$ be a distribution of $(\mathbf{x},y)\in\mathbb{R}^d\times\mathbb{R}$ such that its $\mathbf{x}$-marginal $\mathcal{D}_\mathbf{x}$ is the standard normal. We say that an algorithm is a $C$-approximate proper SIM learner, for some $C \geq 1$, if given $\epsilon > 0$, $W > 0$, and i.i.d. samples from $\mathcal{D}$, the algorithm outputs an activation $\widehat{\sigma}\in\mathcal{F}$ and a vector $\widehat{\mathbf{w}}\in\mathbb{R}^d$ such that with high probability it holds $\mathcal{L}_2(\widehat{\mathbf{w}};\widehat{\sigma}) \leq C\cdot\mathrm{OPT} + \epsilon$, where $\mathrm{OPT} \triangleq \min_{\|\mathbf{w}\|_2\leq W,\sigma\in\mathcal{F}}\mathcal{L}_2(\mathbf{w};\sigma)$.*

The focus of this work is on developing polynomial-time algorithms that achieve a *constant factor* approximation to the optimal loss—i.e., $C = O(1)$—independent of the dimension or any other problem parameters. Achieving $C = 1$ (for the case of Gaussian marginals studied here) is ruled out by computational hardness results [DKZ20, GGK20, DKPZ21, DKR23]. Moreover, even for a constant-factor approximation, strong distributional assumptions are required [DKMR22, GGKS23].

Motivated by the pioneering work of [KKSK11], a central open question in the algorithmic theory of SIMs has been to design an efficient constant-factor approximate SIM learner that succeeds for the class of monotone and Lipschitz activations. While significant algorithmic progress has been made on natural special cases of this task [DGK+20, DKTZ22b, DKTZ22a, ATV23, WZDD23, GGKS23, ZWDD24, GV24, ZWDD25], the general question has remained open:

> *Does there exist an efficient constant-factor approximation algorithm*
> *for learning monotone & Lipschitz SIMs under Gaussian inputs?*

As our main contribution, we resolve this question in the affirmative.

**Theorem 1.2** (Robustly Learning Monotone & Lipschitz SIMs). *There exists a universal constant $C > 1$ such that the following holds. Let $\mathcal{C}$ be the class of all SIMs on $\mathbb{R}^d$ with a monotone and $L$-Lipschitz activation. There is an algorithm that, given $\epsilon > 0$ and $W > 0$, draws $N = d^2\mathrm{poly}(1/\epsilon,W,L)$ samples, runs in $\mathrm{poly}(N)$ time, and returns a predictor $(\widehat{\sigma},\widehat{\mathbf{w}})$ such that, with high probability, $\mathcal{L}_2(\widehat{\mathbf{w}};\widehat{\sigma}) \leq C\mathrm{OPT} + \epsilon$.*

We reiterate that the approximation ratio of our algorithm is a universal constant—independent of the dimension, the desired accuracy, the Lipschitz constant, and the radius of the space.

It is worth noting that our algorithm does not require the Lipschitz assumption on the unknown activation. In fact, it applies for the broader class of monotone activations with bounded $(2 + \zeta)$ moment, for any $\zeta > 0$ (Corollary 2.4). This in particular implies that the case of Linear Threshold Functions (LTFs) fits in our class. As pointed out in [ZWDD25], the bounded moment assumption on top of monotonicity is essentially information-theoretically necessary (in particular, bounded second moment alone does not suffice).

**Comparison to Prior Work** The most directly related prior work is [ZWDD25], which gave an efficient constant-factor approximate learner for the *known activation* version of our problem (i.e., for a known monotone activation with bounded $(2 + \zeta)$ moment). Independently, [GV24] developed an efficient constant-factor learner for the special case of a general (biased) ReLU. Interestingly, even for the special case of known activation (e.g., a general LTF or ReLU), obtaining an efficient constant-factor approximation for more general structured distributions (beyond the Gaussian) remains open. The special case of LTFs, where the first such constant factor approximation was obtained in [DKS18], appears to be a significant bottleneck to go beyond the Gaussian case.

Regarding the SIM version of the problem, prior work either did not achieve a constant-factor approximation or succeeded for a significantly smaller class of activations. Specifically, [GGKS23] gave an efficient SIM learner for monotone 1-Lipschitz activations, for any distribution with bounded second moment, with error guarantee $O(W\sqrt{\mathrm{OPT}}) + \epsilon$ (under the technical assumption that the labels are bounded in $[0,1]$). In addition to the sub-optimal dependence on OPT, the multiplicative factor inside

the big-O scales with the radius $W$ of the space. Subsequently, [ZWDD24] developed an efficient constant-factor SIM learner under structured distributions (including the Gaussian), albeit for a much smaller family of activations. Specifically, for parameters $a, b > 0$, the activation family considered in [ZWDD24] contains all functions $\sigma : \mathbb{R} \to \mathbb{R}$ such that $|\sigma'(z)| \leq b$ everywhere and $\sigma'(z) \geq a > 0$ for $z \geq 0$. The final error bound of the algorithm in [ZWDD24] is $O\big(\mathrm{poly}(b/a)\big)\mathrm{OPT} + \epsilon$. This guarantee is vacuous as $a \to 0$, i.e., for the class of monotone $b$-Lipschitz functions.

## 1.1 Technical Overview

Before getting to the details of our algorithm and its analysis, we first provide "the big picture" of the conceptual novelty of our approach. Prior work on agnostic learning of GLMs and SIMs has as one of its main components a gradient-based algorithm applied to either the square loss [DKTZ22a], its smoothing [ZWDD25], or a suitable surrogate [GGKS23, WZDD23, WZDD24, ZWDD24]. Such approaches are reasonable, considering the long history of gradient-based optimization methods and their applications in learning. However, the considered problems are not black-box, and if we think about optimization algorithms as choosing vector fields to guide the algorithm updates, then the "negative gradient" appearing in gradient-based methods is but one possible choice that could work.

On a conceptual level, our work makes the case for stepping outside the usual boundaries of gradient-based methods and instead looking to directly design a vector field that can be computed from the information given to the algorithm and that carries useful information about the location of target solutions. In the spirit of prior work such as [WZDD23, WZDD24, ZWDD24, ZWDD25], this useful information is captured by the alignment of the chosen vector field and a target parameter vector $\mathbf{w}^*$.

We point out here that this a rather nontrivial goal: even in the case of GLMs (known activation), the negative gradient of the square loss (as used in e.g., [DKTZ22a]) or a standard surrogate loss (as used in e.g., [WZDD23]) can "point in the wrong direction" on some monotone Lipschitz functions (see the discussion in [ZWDD25]). This issue was addressed in the recent work [ZWDD25] by demonstrating that the negative (Riemannian) gradient of the squared loss with Gaussian-smoothed activation, $\mathcal{L}_\rho(\mathbf{w}; \sigma) = (1/(2\rho)) \mathbf{E}_{(\mathbf{x},y)\sim\mathcal{D}}[(y - \mathrm{T}_\rho\sigma(\mathbf{w}\cdot\mathbf{x}))^2]$, correlates with $\mathbf{w}^*$ for an appropriate choice of the smoothing parameter $\rho \in (0, 1)$. Here $\mathrm{T}_\rho\sigma$, $\rho \in (0, 1)$, is the smoothed activation using the Ornstein–Uhlenbeck semi-group; see Section 1.2 and Appendix A for details.

Briefly, [ZWDD25] shows that when $\mathbf{w}$ is still far away from the target $\mathbf{w}^*$, the Riemannian gradient $-\nabla_\mathbf{w}\mathcal{L}_\rho(\mathbf{w}; \sigma) = \mathbf{E}_{(\mathbf{x},y)\sim\mathcal{D}}[(y - \mathrm{T}_\rho\sigma(\mathbf{w}\cdot\mathbf{x}))\mathrm{T}_\rho\sigma'(\mathbf{w}\cdot\mathbf{x})\mathbf{x}^{\perp\mathbf{w}}] = \mathbf{E}_{(\mathbf{x},y)\sim\mathcal{D}}[y\mathrm{T}_\rho\sigma'(\mathbf{w}\cdot\mathbf{x})\mathbf{x}^{\perp\mathbf{w}}]$[1] possesses the 'gradient alignment' property, that is (denoting $\theta := \theta(\mathbf{w}, \mathbf{w}^*)$),

$$-\nabla_\mathbf{w}\mathcal{L}_\rho(\mathbf{w}; \sigma) \cdot \mathbf{w}^* \geq \sin^2\theta \, \mathbf{E}_{\mathbf{x}\sim\mathcal{D}_\mathbf{x}}[\mathrm{T}_{\cos\theta}\sigma'(\mathbf{w}\cdot\mathbf{x})\mathrm{T}_\rho\sigma'(\mathbf{w}\cdot\mathbf{x})] - \sqrt{\mathrm{OPT}}\sin\theta\|\mathrm{T}_\rho\sigma'\|_{L_2}.$$

Then, by carefully and adaptively updating the smoothing parameter $\rho$ so that $\rho \approx \cos\theta$ (for simplicity, we take $\rho = \cos\theta$ below), we have $-\nabla_\mathbf{w}\mathcal{L}_{\cos\theta}(\mathbf{w}; \sigma) \cdot \mathbf{w}^* \gtrsim \sin^2\theta\|\mathrm{T}_{\cos\theta}\sigma'\|_2^2 - \sqrt{\mathrm{OPT}}\sin\theta\|\mathrm{T}_{\cos\theta}\sigma'\|_{L_2}$,[2] which implies that $-\nabla_\mathbf{w}\mathcal{L}_{\cos\theta}(\mathbf{w}; \sigma) \cdot \mathbf{w}^* \gtrsim \sin^2\theta\|\mathrm{T}_{\cos\theta}\sigma'\|_2^2$ when $\sin\theta \gtrsim \sqrt{\mathrm{OPT}}/\|\mathrm{T}_{\cos\theta}\sigma'\|_{L_2}$; in other words, $-\nabla_\mathbf{w}\mathcal{L}_{\cos\theta}(\mathbf{w}; \sigma)$ *aligns* well with the target direction $\mathbf{w}^*$ when $\theta(\mathbf{w}, \mathbf{w}^*)$ is large. Consequently, [ZWDD25] proved that the algorithm linearly converges to $\mathbf{w}^*$ until $\mathbf{w}$ is too close to $\mathbf{w}^*$ (and the alignment condition fails), in which case a $C\mathrm{OPT} + \epsilon$ error solution is found.

A critical issue arises, however, when trying to adapt the methods from [ZWDD25] to the SIM setting: the correlation between $\mathbf{w}^*$ and the Riemannian gradient $\nabla_\mathbf{w}\mathcal{L}_{\cos\theta}(\mathbf{w}; u)$ becomes uncontrollable, even if we choose $u$ as the "best-fit" activation given $\mathbf{w}$. To see this, a simple calculation shows that

$$-\nabla_\mathbf{w}\mathcal{L}_{\cos\theta}(\mathbf{w}; u) \cdot \mathbf{w}^* = \mathbf{E}_{(\mathbf{x},y)\sim\mathcal{D}}[y\mathrm{T}_{\cos\theta}u'(\mathbf{w}\cdot\mathbf{x})\mathbf{x}^{\perp\mathbf{w}} \cdot \mathbf{w}^*]$$

$$\gtrsim \sin^2\theta \, \mathbf{E}_{\mathbf{x}\sim\mathcal{D}_\mathbf{x}}[\mathrm{T}_{\cos\theta}\sigma'(\mathbf{w}\cdot\mathbf{x})\mathrm{T}_{\cos\theta}u'(\mathbf{w}\cdot\mathbf{x})] - \sqrt{\mathrm{OPT}}\sin\theta\|\mathrm{T}_{\cos\theta}u'\|_{L_2}.$$

Even though it is possible to control the $L_2^2$ distance between $u(z)$ and $\sigma(z)$ by $\theta(\mathbf{w}, \mathbf{w}^*)$ following similar steps as in [ZWDD24], it is unclear how to show that $\mathrm{T}_{\cos\theta}u'(z)$ is close to $\mathrm{T}_{\cos\theta}\sigma'(z)$ so

---

[1] $\mathbf{x}^{\perp\mathbf{w}}$ here denotes the projection of $\mathbf{x}$ on the orthogonal complement of $\mathbf{w} : \mathbf{x}^{\perp\mathbf{w}} = (\mathbf{I} - \mathbf{w}\mathbf{w}^\top)\mathbf{x}$. Note that $\mathbf{x}^{\perp\mathbf{w}}$ is independent of $\mathbf{w}\cdot\mathbf{x}$ (as $\mathbf{x}$ is standard Gaussian), hence $\mathbf{E}_{\mathbf{x}\sim\mathcal{D}_\mathbf{x}}[\mathrm{T}_\rho\sigma(\mathbf{w}\cdot\mathbf{x})\mathrm{T}_\rho\sigma'(\mathbf{w}\cdot\mathbf{x})\mathbf{x}^{\perp\mathbf{w}}] = 0$.

[2] Here, $A \gtrsim B$ means that there exists a universal positive constant $C$ so that $A \geq CB$.

that $-\nabla_{\mathbf{w}}\mathcal{L}_\rho(\mathbf{w};u)\cdot\mathbf{w}^* \gtrsim \sin^2\theta\|\mathrm{T}_{\cos\theta}\sigma'\|_{L_2}^2$ and the arguments of [ZWDD25] can go through. Intuitively, the smoothing operator $\mathrm{T}_{\cos\theta}$ filters out the high-order components of the derivative of the activation $\sigma'$ in the Hermite basis while keeping only the low-order components. Thus, to ensure $\|\mathrm{T}_{\cos\theta}u' - \mathrm{T}_{\cos\theta}\sigma'\|_{L_2}$ is small, it necessitates approximating the low degree coefficients of $\sigma'$ (under adversarial noise and without the knowledge of $\sigma$), imposing formidable technical challenges. In particular, it is unclear whether prior SIM learning frameworks [KKSK11, ZWDD24, HTY25]—alternating between the "best-fit" updates for activation $u$ and gradient-style updates for $\mathbf{w}$—can resolve this issue. Hence, our work represents a departure from this seemingly natural approach.

**Alignment with a New Vector Field** Our method deviates from prior works in that we no longer cling to the gradient field of a particular loss function, but rather, we identify a vector field that aligns with $\mathbf{w}^*$ *without the need to estimate the target function $\sigma$ on the run*. Revisiting the correlation between the gradient of the smoothed $L_2^2$ loss and the target vector $\mathbf{w}^*$: $-\nabla_{\mathbf{w}}\mathcal{L}_{\cos\theta}(\mathbf{w};\sigma)\cdot\mathbf{w}^* = \mathbf{E}_{(\mathbf{x},y)\sim\mathcal{D}}[y\mathrm{T}_{\cos\theta}\sigma'(\mathbf{w}\cdot\mathbf{x})\mathbf{x}^{\perp\mathbf{w}}\cdot\mathbf{w}^*]$, we observe that the right-hand side of the equation consists of three parts: the label $y$, a random variable $\mathbf{x}^{\perp\mathbf{w}}\cdot\mathbf{w}^*$ that is independent of $\mathbf{w}\cdot\mathbf{x}$, and a function $\mathrm{T}_{\cos\theta}\sigma'(\mathbf{w}\cdot\mathbf{x})$ of $\mathbf{w}\cdot\mathbf{x}$ that is not available when $\sigma$ is unknown. Critically, we replace the unknown function $\mathrm{T}_{\cos\theta}\sigma'(z)$ with any function $h(z)$ such that $\|h(z)\|_{L_2}=1$, and we ask: which function $h^*(z)$ maximizes the correlation $K(h) := \mathbf{E}_{(\mathbf{x},y)\sim\mathcal{D}}[y\mathbf{x}^{\perp\mathbf{w}}h(\mathbf{w}\cdot\mathbf{x})]\cdot\mathbf{w}^*$? Let $h_0(z) := \mathrm{T}_{\cos\theta}\sigma'(z)/\|\mathrm{T}_{\cos\theta}\sigma'\|_{L_2}$. By maximality of $h^*$, we know that the correlation $K(h^*)$ is at least as large as $K(h_0) = -\nabla_{\mathbf{w}}\mathcal{L}_{\cos\theta}(\mathbf{w};\sigma)\cdot\mathbf{w}^*/\|\mathrm{T}_{\cos\theta}\sigma'\|_{L_2}$, indicating the vector field $\mathbf{H}_{\mathbf{w}}^* := \mathbf{E}_{(\mathbf{x},y)\sim\mathcal{D}}[y\mathbf{x}^{\perp\mathbf{w}}h^*(\mathbf{w}\cdot\mathbf{x})]$ is at least as good as the gradient $\nabla_{\mathbf{w}}\mathcal{L}_{\cos\theta}(\mathbf{w};\sigma)$ of the smoothed loss with respect to the *target activation $\sigma$*, after normalization.

In fact, letting $\mathbf{v}_{\mathbf{w}}^* = (\mathbf{w}^*)^{\perp\mathbf{w}}/\|(\mathbf{w}^*)^{\perp\mathbf{w}}\|_2$ and $\mathbf{w}^* = \cos\theta\mathbf{w} + \sin\theta\mathbf{v}_{\mathbf{w}}^*$, we can write $K(h)$ as:

$$K(h) = \underset{(\mathbf{x},y)\sim\mathcal{D}}{\mathbf{E}}[yh(\mathbf{w}\cdot\mathbf{x})\mathbf{w}^*\cdot\mathbf{x}^{\perp\mathbf{w}}] = \sin\theta\underset{\mathbf{x}\sim\mathcal{D}_{\mathbf{x}}}{\mathbf{E}}\Big[\underset{(\mathbf{x},y)\sim\mathcal{D}}{\mathbf{E}}[y(\mathbf{v}_{\mathbf{w}}^*\cdot\mathbf{x})\mid\mathbf{w}\cdot\mathbf{x}]h(\mathbf{w}\cdot\mathbf{x})\Big]. \quad (1)$$

Consider the $L_2$ space of the standard Gaussian random variable $\mathbf{w}\cdot\mathbf{x}$ equipped with the inner product $\langle a,b\rangle = \mathbf{E}_{\mathbf{w}\cdot\mathbf{x}\sim\mathcal{N}}[a\cdot b]$; it is known from duality that $\langle a,b\rangle \leq \|a\|_{L_2}\|b\|_{L_2}$ with equality if $a=b$ almost surely. Hence, the choice $h^*(\mathbf{w}\cdot\mathbf{x}) = \mathbf{E}[y\mathbf{v}_{\mathbf{w}}^*\cdot\mathbf{x}\mid\mathbf{w}\cdot\mathbf{x}]/\|\mathbf{E}[y\mathbf{v}_{\mathbf{w}}^*\cdot\mathbf{x}\mid\mathbf{w}\cdot\mathbf{x}]\|_{L_2}$ maximizes Equation (1). Having access to such $h^*$ would guarantee that the corresponding update rule using $\mathbf{H}_{\mathbf{w}}^* = \mathbf{E}_{(\mathbf{x},y)\sim\mathcal{D}}[y\mathbf{x}^{\perp\mathbf{w}}h^*(\mathbf{w}\cdot\mathbf{x})]$ performs at least as well as the gradient descent on the smoothed loss $\mathcal{L}_{\cos\theta}(\mathbf{w};\sigma)$–which is precisely the update of [ZWDD25] used in the case of known $\sigma$. Note that by the definition of $K(h)$, we have $K(h^*) = \sin\theta\mathbf{v}_{\mathbf{w}}^*\cdot\mathbf{H}_{\mathbf{w}}^*$.

There are two obstacles in finding such an $h^*$: 1) the learner does not have knowledge of $\mathbf{w}^*$, and 2) even if the learner knew $\mathbf{w}^*$, it would not be possible to estimate $h^*(z) = \mathbf{E}[y\mathbf{v}_{\mathbf{w}}^*\cdot\mathbf{x}\mid\mathbf{w}\cdot\mathbf{x}=z]/\|\mathbf{E}[y\mathbf{v}_{\mathbf{w}}^*\cdot\mathbf{x}\mid\mathbf{w}\cdot\mathbf{x}]\|_{L_2}$ on any desired point $\mathbf{w}\cdot\mathbf{x}=z$ (as the probability of observing a single point twice is zero). For this reason, we need to consider a different way of finding an update rule.

**The Spectral Subroutine** For the first obstacle, our critical observation is that instead of estimating $h^*$, we can approximate the vector $\mathbf{H}_{\mathbf{w}}^*$ directly via spectral methods. Let us define $\mathbf{g}_{\mathbf{w}}^*(z) := \mathbf{E}_{(\mathbf{x},y)\sim\mathcal{D}}[y\mathbf{x}^{\perp\mathbf{w}}\mid\mathbf{w}\cdot\mathbf{x}=z]$ and consider the matrix $\mathbf{M}_{\mathbf{w}}^* = \mathbf{E}_{z\sim\mathcal{N}}[\mathbf{g}_{\mathbf{w}}^*(z)\mathbf{g}_{\mathbf{w}}^*(z)^\top]$. Observe that

$$(\mathbf{v}_{\mathbf{w}}^*)^\top\mathbf{M}_{\mathbf{w}}^*\mathbf{v}_{\mathbf{w}}^* = \underset{\mathbf{x}\sim\mathcal{D}_{\mathbf{x}}}{\mathbf{E}}\Big[\Big(\underset{(\mathbf{x},y)\sim\mathcal{D}}{\mathbf{E}}[y\mathbf{x}^{\perp\mathbf{w}}\cdot\mathbf{v}_{\mathbf{w}}^*\mid\mathbf{w}\cdot\mathbf{x}]\Big)^2\Big] = (\mathbf{v}_{\mathbf{w}}^*\cdot\mathbf{H}_{\mathbf{w}}^*)\|\mathbf{E}[y\mathbf{v}_{\mathbf{w}}^*\cdot\mathbf{x}\mid\mathbf{w}\cdot\mathbf{x}]\|_{L_2}$$

$$= \frac{K(h^*)}{\sin\theta}\|\mathbf{E}[y\mathbf{v}_{\mathbf{w}}^*\cdot\mathbf{x}\mid\mathbf{w}\cdot\mathbf{x}]\|_{L_2} \geq \frac{K(\mathrm{T}_{\cos\theta}\sigma')}{\sin\theta\|\mathrm{T}_{\cos\theta}\sigma'\|_{L_2}}\|\mathbf{E}[y\mathbf{v}_{\mathbf{w}}^*\cdot\mathbf{x}\mid\mathbf{w}\cdot\mathbf{x}]\|_{L_2},$$

where in the last inequality we used the maximality of $K(h^*)$. The first equality above also implies $\mathbf{v}_{\mathbf{w}}^*\cdot\mathbf{M}_{\mathbf{w}}^*\mathbf{v}_{\mathbf{w}}^* = \|\mathbf{E}[y\mathbf{v}_{\mathbf{w}}^*\cdot\mathbf{x}\mid\mathbf{w}\cdot\mathbf{x}]\|_{L_2}^2$; therefore, we have $(\mathbf{v}_{\mathbf{w}}^*\cdot\mathbf{M}_{\mathbf{w}}^*\mathbf{v}_{\mathbf{w}}^*)^{1/2} = \mathbf{v}_{\mathbf{w}}^*\cdot\mathbf{H}_{\mathbf{w}}^*$. Since we know that $\mathbf{H}_{\mathbf{w}}^*$ is at least as aligned with $\mathbf{w}^*$ as the gradient vector $-\nabla_{\mathbf{w}}\mathcal{L}_{\cos\theta}(\mathbf{w};\sigma)$ and we know from [ZWDD25] that $K(\mathrm{T}_{\cos\theta}\sigma') = -\nabla_{\mathbf{w}}\mathcal{L}_\rho(\mathbf{w};\sigma)\cdot\mathbf{w}^* \gtrsim \sin^2\theta\|\mathrm{T}_{\cos\theta}\sigma'\|_{L_2}^2 \gg 0$ since the gradient alignment condition holds, we get that both $\mathbf{v}_{\mathbf{w}}^*\cdot\mathbf{M}_{\mathbf{w}}^*\mathbf{v}_{\mathbf{w}}^*$ and $\mathbf{v}_{\mathbf{w}}^*\cdot\mathbf{H}_{\mathbf{w}}^*$ are far away from 0. This implies that much of the information on $\mathbf{H}_{\mathbf{w}}^*$ is contained in $\mathbf{v}_{\mathbf{w}}^*$ and, in addition, $\mathbf{v}_{\mathbf{w}}^*$ is contained in the eigenspace of the large eigenvectors of $\mathbf{M}_{\mathbf{w}}^*$. Furthermore, we show that for any direction $\mathbf{u}$ that is orthogonal to $\mathbf{v}_{\mathbf{w}}^*$, both $\mathbf{u}\cdot\mathbf{M}_{\mathbf{w}}^*\mathbf{u} \lesssim \mathrm{OPT}$ and $\mathbf{H}_{\mathbf{w}}^*\cdot\mathbf{u} \lesssim \sqrt{\mathrm{OPT}}$ are small. In other words, $\mathbf{H}_{\mathbf{w}}^*$ is almost completely captured by $\mathbf{v}_{\mathbf{w}}^*$, and $\mathbf{v}_{\mathbf{w}}^*$ is effectively contained in the space of the highest eigenvalues. Consequently, $\mathbf{H}_{\mathbf{w}}^*$ is approximated by the top eigenvectors of $\mathbf{M}_{\mathbf{w}}^*$.

**Approximation and Regularity of Monotone Functions**    The second obstacle is to estimate the function $\mathbf{g}_{\mathbf{w}}^*(z) = \mathbf{E}_{(\mathbf{x},y)\sim\mathcal{D}}[y\mathbf{x}^{\perp\mathbf{w}} \mid \mathbf{w}\cdot\mathbf{x} = z]$ that constructs the matrix $\mathbf{M}_{\mathbf{w}}^*$. This is not a trivial task even in the noiseless setting because we are conditioning on a hyperplane that has measure zero in the space. One would hope that conditioning on small bands suffices for this purpose; namely, that if $\mathbf{w}\cdot\mathbf{x} \in (a,b)$ with $|b-a| = \mathrm{poly}(\epsilon)$, then for all $z \in (a,b)$ it would be $\mathbf{E}[y\mathbf{x}^{\perp\mathbf{w}} \mid \mathbf{w}\cdot\mathbf{x} = z] \approx \mathbf{E}[y\mathbf{x}^{\perp\mathbf{w}} \mid \mathbf{w}\cdot\mathbf{x} \in (a,b)]$. Unfortunately, since the labels $y$ are adversarially corrupted, the adversary could corrupt $y$ for each value of $\mathbf{w}\cdot\mathbf{x} = z$ —in which case such an approximation would not yield an accurate estimate. Instead, we proceed as follows: consider restricting $h(z)$ to be a piecewise-constant function on a fixed set of small bands $\mathcal{E}_i = [a_i, a_{i+1}), i \in [I]$. We show that the argument about maximizing $K(h)$ on continuous functions $h$ can be carried out similarly to piecewise constants. Let $\widetilde{h}^*$ be the piecewise constant function that maximizes $K(h)$. We further show that $\mathrm{T}_{\cos\theta}\sigma'$ can be approximated by a fixed value on each band $\mathcal{E}_i$. Hence, using a piecewise-constant approximate function $\widetilde{\mathrm{T}}_{\cos\theta}\sigma'$, we have that $0 \ll K(\mathrm{T}_{\cos\theta}\sigma') \approx K(\widetilde{\mathrm{T}}_{\cos\theta}\sigma') \leq K(\widetilde{h}^*)$, and thus the argument that large eigenvectors of $\mathbf{M}_{\mathbf{w}}^* = \mathbf{E}_{z\sim\mathcal{N}}[\mathbf{g}_{\mathbf{w}}^*(z)\mathbf{g}_{\mathbf{w}}^*(z)^\top]$ contain information about $\mathbf{H}_{\mathbf{w}}^*$ can be extended to $\mathbf{M}_{\mathbf{w}} = \mathbf{E}_{z\sim\mathcal{N}}[\mathbf{g}_{\mathbf{w}}(z)\mathbf{g}_{\mathbf{w}}(z)^\top]$, where $\mathbf{g}_{\mathbf{w}}(z)$ is the piecewise constant version of $\mathbf{g}_{\mathbf{w}}^*(z)$, which can now be efficiently estimated.

**Optimization via Random Walk**    A final obstacle in this approach is that both $\mathbf{u}$ and $-\mathbf{u}$ are eigenvectors that correspond to the maximum eigenvalue and we cannot determine whether $\mathbf{u}$ or $-\mathbf{u}$ correlates positively with $\mathbf{w}^*$. To address this issue, at each iteration, we pick the direction from $\{\mathbf{u}, -\mathbf{u}\}$ at random. Consequently, with probability $1/2$, the algorithm will decrease the angle with $\mathbf{w}^*$. Consider the random variable $Z_t = \theta(\mathbf{w}^{(t)}, \mathbf{w}^*)$—the angle between $\mathbf{w}_t$ and $\mathbf{w}^*$. Let $\theta^*$ be the largest angle such that if $Z_t \leq \theta^*$ then $\mathcal{L}_2(\mathbf{w}^{(t)}; \sigma) \leq O(\mathrm{OPT}) + \epsilon$. Furthermore, let $\theta_0 := \theta(\mathbf{w}^{(0)}, \mathbf{w}^*)$. Assume without loss of generality that the initialized vector $\mathbf{w}^{(0)}$ is not a constant approximate vector, hence $\theta_0 \geq \theta^*$. Now let $\tau_1 = \inf_t\{t \geq 1 \mid Z_t \leq \theta^*\}$, i.e., $\tau_1$ is the first iteration that has $Z_{\tau_1} \leq \theta^*$, and let $\tau_2 = \inf_t\{t \geq 1 \mid Z_t \geq \theta_0\}$. If $\mathbf{Pr}[\tau_1 < \tau_2] \geq \alpha > 0$, then repeating the process $O(1/\alpha)$ times guarantees that with high probability the event $\tau_1 < \tau_2$ happens. Note that: $\mathbf{Pr}[\tau_1 < \tau_2] \geq \mathbf{Pr}[\text{chooses the correct direction for all the } T \text{ steps until } Z_t \leq \theta^*] = 2^T$. We will show that $T \lesssim \log(BL/\epsilon)$. Thus, we have $\mathbf{Pr}[\tau_1 < \tau_2] \geq 1/2^T = \mathrm{poly}(\epsilon, 1/B, 1/L)$. Therefore, repeating the algorithm $2^T = \mathrm{poly}(1/\epsilon, B, L)$ times suffices.

## 1.2   Notation and Preliminaries

For $n \in \mathbb{Z}_+$, let $[n] := \{1, \ldots, n\}$. We use bold lowercase letters to denote vectors and bold uppercase letters for matrices. For $\mathbf{x} \in \mathbb{R}^d$, $\|\mathbf{x}\|_2$ denotes the $\ell_2$-norm of $\mathbf{x}$. For a matrix $\mathbf{M} \in \mathbb{R}^{d\times d}$, $\|\mathbf{M}\|_2$ denotes the operator norm of $\mathbf{M}$. We use $\mathbf{x}\cdot\mathbf{y}$ for the dot product of $\mathbf{x}, \mathbf{y} \in \mathbb{R}^d$ and $\theta(\mathbf{x}, \mathbf{y})$ for the angle between $\mathbf{x}, \mathbf{y}$. We use $\mathbb{1}\{A\}$ to denote the characteristic function of the set $A$. For unit vectors $\mathbf{u}, \mathbf{v}$, we use $\mathbf{u}^{\perp\mathbf{v}}$ to denote the component of $\mathbf{u}$ that is orthogonal to $\mathbf{v}$ i.e., $\mathbf{u}^{\perp\mathbf{v}} = (\mathbf{I} - \mathbf{v}\mathbf{v}^\top)\mathbf{u}$. $\mathbb{S}^{d-1}$ denotes the unit sphere in $\mathbb{R}^d$ and $\mathbb{B}$ denotes the unit ball. For $(\mathbf{x}, y)$ distributed according to $\mathcal{D}$, we denote by $\mathcal{D}_{\mathbf{x}}$ the marginal distribution of $\mathbf{x}$. We use $\widehat{\mathcal{D}}_N$ to denote the empirical distribution constructed by $N$ i.i.d. samples from $\mathcal{D}$. We use the standard $O(\cdot), \Theta(\cdot), \Omega(\cdot)$ asymptotic notation and $\widetilde{O}(\cdot)$ to omit polylogarithmic factors in the argument. We write $E \gtrsim F$ for two non-negative expressions $E$ and $F$ to denote that *there exists* some positive universal constant $c > 0$ such that $E \geq cF$. $E \lesssim F$ is defined similarly. We write $E \approx F$ if $E \lesssim F$ and $E \gtrsim F$. We write $E \gg F$ if there exists a large universal constant $C > 0$ such that $E \geq CF$. $E \ll F$ is similarly defined.

Let $\mathcal{N}(\mathbf{0}, \mathbf{I})$ denote the standard $d$-dimensional normal distribution. The $L_2$ norm of a function $g$ with respect to the standard normal is $\|g\|_{L_2} = (\mathbf{E}_{\mathbf{x}\sim\mathcal{N}}[|g(\mathbf{x})|^2]^{1/2})$, while $\|g\|_{L_\infty}$ is the essential supremum of the absolute value of $g$. We denote by $L_2(\mathcal{N})$ the vector space of all functions $f : \mathbb{R}^d \to \mathbb{R}$ such that $\|f\|_{L_2} < \infty$. An important tool for our work is the Ornstein–Uhlenbeck semigroup, defined below.

**Definition 1.3** (Ornstein–Uhlenbeck Semigroup). *Let $\rho \in (0,1)$. The Ornstein–Uhlenbeck semigroup, denoted by $\mathrm{T}_\rho$, is a linear operator that maps a function $g \in L_2(\mathcal{N})$ to the function $\mathrm{T}_\rho g$ defined as:* $(\mathrm{T}_\rho g)(\mathbf{x}) := \mathbf{E}_{\mathbf{z}\sim\mathcal{N}}[g(\rho\mathbf{x} + \sqrt{1-\rho^2}\mathbf{z})].$

## 2 Robustly Learning SIMs

We consider distributions $\mathcal{D}$ over $(\mathbf{x}, y) \in \mathbb{R}^d \times \mathbb{R}$ with $\mathbf{x} \sim \mathcal{N}(\mathbf{0}, \mathbf{I})$, and predictors of the form $f_{\mathbf{w},\sigma}(\mathbf{x}) = \sigma(\mathbf{w} \cdot \mathbf{x})$, where $\|\mathbf{w}\| \leq W$ and $\sigma$ is monotone. We assume that the target activation $\sigma$ is $\epsilon$-close to a $(B, L)$-Regular function, that satisfies the following conditions:

**Definition 2.1** ($(B, L)$-Regular Monotone Activations). *Given parameters $B, L > 0$, we define the class of $(B, L)$-Regular activations, denoted by $\mathcal{H}(B, L)$, as the class containing all functions $\sigma : \mathbb{R} \to \mathbb{R}$ such that 1) $\|\sigma\|_{L_\infty} \leq B$ and 2) $\|\sigma'\|_{L_2} \leq L$. Given $\epsilon > 0$, we define the class of $\epsilon$-Extended $(B, L)$-Regular activations, denoted by $\mathcal{H}_\epsilon(B, L)$, as the class containing all activations $\sigma_1 : \mathbb{R} \to \mathbb{R}$ for which there exists $\sigma_2 \in \mathcal{H}(B, L)$ such that $\|\sigma_1 - \sigma_2\|_{L_2}^2 \leq \epsilon$.*

**Remark 2.2.** Instead of directly enforcing a norm bound $\|\mathbf{w}\| \leq W$, one can assume that $\mathcal{H}_\epsilon(B, L)$ that we compete against is chosen so that it contains all the activations $\sigma(z) \mapsto \sigma(\lambda z)$ for all $\lambda \leq W$. This lets us focus on the core statistical challenge without separately tracking a norm constraint on $\mathbf{w}$.

Learning with respect to the class of activations $\mathcal{H}_\epsilon(B, L)$ allows us to make the following simplifying assumption that comes at no loss of generality. We can assume that the labels $y$ are truncated in the interval $[-B, B]$, and, as a result, we can assume that $|y| \leq B$ (see Fact A.9). Our main result is that Algorithm 1 efficiently generates a solution pair that achieves $C \mathrm{OPT} + \epsilon$ error:

---

**Algorithm 1** Main algorithm

---

1: **Input:** Parameters $B, L, \epsilon, \delta$; Data access $(\mathbf{x}, y) \sim \mathcal{D}$, empty set $S^{\mathrm{sol}}$.
2: $S^{\mathrm{ini}} \leftarrow \mathrm{Initialziation}[B, \epsilon]$ (Algorithm 2), $S^{\mathrm{sol}} \leftarrow S^{\mathrm{ini}}$.
3: Sample $N$ i.i.d. samples $\{(\mathbf{x}^{(i)}, y^{(i)})\}_{i=1}^N$ from $\mathcal{D}$ and construct $\widehat{\mathcal{D}}_N$.
4: **for** $\mathbf{w}^{(0)} \in S^{\mathrm{ini}}$ **do**
5:      **for** $\bar{\theta} \in \Theta = \{k\epsilon/L : k \in [L/\epsilon]\}$ **do**
6:          Run $\mathrm{SpectralOptimization}[\bar{\theta}, \mathbf{w}^{(0)}, \widehat{D}_N]$ (Algorithm 3) and get $S$.
7:          $S^{\mathrm{sol}} \leftarrow S^{\mathrm{sol}} \cup S$.
8: $(\widehat{\mathbf{w}}, \widehat{u}) = \mathrm{Test}[S^{\mathrm{sol}}]$ (Algorithm 5).
9: **Return:** $(\widehat{\mathbf{w}}, \widehat{u})$.

---

**Theorem 2.3** (Main Result). *Let $\epsilon > 0$ and let $B, L > 0$. Let $\mathcal{D}$ be a distribution over $(\mathbf{x}, y) \in \mathbb{R}^d \times \mathbb{R}$ with $\mathbf{x} \sim \mathcal{N}(\mathbf{0}, \mathbf{I}_d)$. Let $(\mathbf{w}^*, \sigma) \in \mathbb{R}^d \times \mathcal{H}_\epsilon(B, L)$ be a pair of vector and monotone activation such that $\mathcal{L}_2(\mathbf{w}^*; \sigma) = \mathrm{OPT}$. Then Algorithm 1 draws $N = d^2 \mathrm{poly}(B, L, 1/\epsilon)$ samples, it runs in at most $\mathrm{poly}(N, d)$ time, and it returns a vector $\widehat{\mathbf{w}}$ and a monotone and Lipschitz activation $\widehat{u} : \mathbb{R} \to \mathbb{R}$, such that with probability at least $2/3$, it holds that $\mathcal{L}_2(\widehat{\mathbf{w}}; \widehat{u}) \leq O(\mathrm{OPT}) + \epsilon$.*

Using the fact that any monotone function $\sigma$ with bounded $2 + \zeta$ moment $\mathbf{E}_{z \sim \mathcal{N}}[|\sigma(z)|^{2+\zeta}] \leq B_\sigma$ is an $\epsilon$-Extended $(B, L)$-Regular with $B, L = \mathrm{poly}((B_\sigma/\epsilon)^{1/\zeta}, 1/\epsilon)$ (see Fact A.7), we have:

**Corollary 2.4.** *Let $\epsilon, \zeta > 0$. Let $(\mathbf{x}, y) \sim \mathcal{D}$ with $\mathcal{D}_\mathbf{x} = \mathcal{N}(\mathbf{0}, \mathbf{I})$. Let $\mathbf{w}^* \in \mathbb{R}^d$ be a unit vector and let $\sigma$ be a monotone function with bounded $(2 + \zeta)$ moment, i.e., $\mathbf{E}_{z \sim \mathcal{N}}[|\sigma(z)|^{2+\zeta}] \leq B_\sigma$, such that $\mathcal{L}_2(\mathbf{w}^*; \sigma) = \mathrm{OPT}$. Then, Algorithm 1 draws $N = d^2 \mathrm{poly}((B_\sigma/\epsilon)^{1/\zeta}, 1/\epsilon)$ samples, runs in at most $\mathrm{poly}(N, d)$ time, and returns a vector $\widehat{\mathbf{w}}$ and a monotone and Lipschitz activation $\widehat{u} : \mathbb{R} \to \mathbb{R}$, such that with probability at least $2/3$, it holds that $\mathcal{L}_2(\widehat{\mathbf{w}}; \widehat{u}) \leq O(\mathrm{OPT}) + \epsilon$.*

Note further that monotone $L$-Lipschitz functions belong to $\mathcal{H}_\epsilon(B, L)$ with $B = O(L \log(L/\epsilon))$; see Fact A.7.

The body of the section is organized as follows: in Section 2.1 we prove the correctness of the initialization subroutine; in Section 2.2 we present the main component of our algorithm, the spectral subroutine and show that it generates a pair of solution achieving small error; Section 2.3 presents the proof of the main theorem (Theorem 2.3).

### 2.1 Initialization

In this section, we present the initialization algorithm. The goal of our initialization is to find a vector $\mathbf{w}^{(0)}$ such that $\theta(\mathbf{w}^{(0)}, \mathbf{w}^*) \leq 1/M$, where $M$ is the smallest threshold such that

$$\mathbf{E}_{z \sim \mathcal{N}}[(\sigma(z) - \sigma(M))^2 \mathbb{1}\{|z| \geq M\}] \leq C(\mathrm{OPT} + \epsilon)$$

for some large absolute constant $C$; in other words, we can truncate the activation $\sigma(z)$ after $|z| \geq M$ without inducing much error. To find such a vector $\mathbf{w}^{(0)}$, we design a label transformation $\mathcal{T}(y) = \mathbb{1}\{y \geq t\}$ for a carefully chosen threshold $t$ and transform the regression problem to a robust halfspace learning problem, following the same procedure as in [ZWDD25] (see Section 4.3 of [ZWDD25]). Since (unlike in [ZWDD25]) $\sigma$ is unknown, neither this threshold $t$ nor the parameter $M$ are known to the learner. Our workaround is to construct a grid of possible thresholds $t_i$ (we argue at most $B/\sqrt{\epsilon}$ of values in the grid suffice) and argue that with high probability there exists a threshold $t^*$ such that the initialization succeeds. We store all the vectors generated by the initialization algorithm, based on all these thresholds. We find the correct parameter vector in the final testing stage of the main algorithm. The proof of the following lemma is deferred to Appendix B.

**Lemma 2.5** (Initialization)**.** *Let $\sigma(\mathbf{w}^* \cdot \mathbf{x})$ be a hypothesis that satisfies $\mathbf{E}_{(\mathbf{x},y)\sim\mathcal{D}}[(y-\sigma(\mathbf{w}^*\cdot\mathbf{x}))^2] \leq \mathrm{OPT}+\epsilon$, where $\sigma$ is a non-decreasing $\epsilon$-Extended $(B, L)$-Regular function. Suppose that no constant hypothesis, i.e., function of the form $\sigma(z) = c$ for any $c \in \mathbb{R}$, is a constant-factor approximate solution. Let $C > 1$ be a large absolute constant and let $M > 0$ be the smallest parameter such that $\mathbf{E}_{z\sim\mathcal{N}}[(\sigma(z) - \sigma(M))^2 \mathbb{1}\{z \geq M\}] \leq C(\mathrm{OPT} + \epsilon)$. Then Algorithm 2, using $O(d/\epsilon^2 \log(B/\epsilon))$ samples, with probability at least $99\%$, returns a list $S^{\mathrm{ini}}$ of $O(B/\sqrt{\epsilon})$ vectors that contains a vector $\mathbf{w}^{(0)}$ such that $\theta(\mathbf{w}^{(0)}, \mathbf{w}^*) \leq \min(1/M, \pi/16)$.*

---

**Algorithm 2** Initialization

1: **Input:** Parameters $B$, $\epsilon$; Data access $(\mathbf{x}, y) \sim \mathcal{D}$; $S \leftarrow \emptyset$.
2: **for** $i = 1, \ldots, \lceil B/\sqrt{\epsilon} \rceil + 1$ **do**
3: $\quad$ $t_i = i\sqrt{\epsilon}$, transform the data to $\mathcal{D}_i = (\mathbf{x}, \mathcal{T}(y; t_i))$ where $\mathcal{T}(y; t_i) = \mathbb{1}\{y \geq t_i\}$.
4: $\quad$ Run the Robust Halfspace Learning algorithm from Fact B.2, get parameter $\mathbf{w}^{(0,i)}$
5: $\quad$ $S \leftarrow S \cup \{\mathbf{w}^{(0,i)}\}$
6: **Return:** $S$.

---

## 2.2 The Spectral Subroutine

In this section, we present our main structural result (Proposition 2.6). We show that—even though the target activation $\sigma$ is unknown—we can identify a vector that has a strong correlation with an 'ideal descent direction' $\mathbf{v}_{\mathbf{w}}^* := (\mathbf{w}^*)^{\perp\mathbf{w}}/\|(\mathbf{w}^*)^{\perp\mathbf{w}}\|_2$. It is not hard to see that $\mathbf{v}_{\mathbf{w}}^*$ can be used to rotate $\mathbf{w}$ towards $\mathbf{w}^*$. The vector that we identify is a top eigenvector of a matrix $\mathbf{M}_{\mathbf{w}}$ that can be efficiently estimated using sample access to labeled data. We can only identify such a target vector up to its sign; however, as we argue later, this is sufficient for our argument to go through.

To build up this result, we need the following technical pieces: (1) the spectrum of the matrix $\mathbf{M}_{\mathbf{w}}$ contains information on $\mathbf{v}_{\mathbf{w}}^*$, i.e., $\mathbf{v}_{\mathbf{w}}^* \cdot \mathbf{M}_{\mathbf{w}} \mathbf{v}_{\mathbf{w}}^*$ is large (Lemma 2.7); (2) All the other directions $\mathbf{u}$ that are orthogonal to $\mathbf{v}_{\mathbf{w}}^*$ have small quadratic form value compared to $\mathbf{v}_{\mathbf{w}}^*$, i.e., $\mathbf{u} \cdot \mathbf{M}_{\mathbf{w}} \mathbf{u} \ll \mathbf{v}_{\mathbf{w}}^* \cdot \mathbf{M}_{\mathbf{w}} \mathbf{v}_{\mathbf{w}}^*$, and therefore, the direction $\mathbf{v}_{\mathbf{w}}^*$ stands out in the spectrum of the matrix $\mathbf{M}_{\mathbf{w}}$ (Lemma 2.8); (3) finally, we argue that the top eigenvector $\mathbf{v}_{\mathbf{w}}$ of $\mathbf{M}_{\mathbf{w}}$ correlates strongly with $\mathbf{v}_{\mathbf{w}}^*$ (Lemma 2.9).

---

**Algorithm 3** Spectral Optimization

1: **Input:** Parameter $\theta_0$; Initialization vector $\mathbf{w}^{(0)}$; Empirical Distribution $\widehat{\mathcal{D}}_N$;
2: $S^{\mathrm{sol}} \leftarrow \emptyset$, $\phi_t \leftarrow \bar{\theta}(1 - 1/128)^t$, $\eta_t \leftarrow \sin\phi_t/8$, $K \leftarrow \mathrm{poly}(1/\epsilon, L)$ and $T \leftarrow \Theta(\log(1/(\epsilon L)))$.
3: **for** $k = 1, \ldots, K$ **do**
4: $\quad$ **for** $t = 0, \ldots, T$ **do**
5: $\quad\quad$ Let $\widehat{\mathbf{g}}_{\mathbf{w}^{(t)}}^{(j)} \leftarrow \mathbf{E}_{(\mathbf{x},y)\sim\widehat{\mathcal{D}}_N}[y\mathbf{x}^{\perp\mathbf{w}^{(t)}}\mathbb{1}\{\mathbf{w}^{(t)} \cdot \mathbf{x} \in \mathcal{E}_j\}]$, $j \in [I]$.
6: $\quad\quad$ Compute the empirical matrix $\widehat{\mathbf{M}}_{\mathbf{w}^{(t)}} \leftarrow \sum_{j=1}^{I} \widehat{\mathbf{g}}_{\mathbf{w}^{(t)}}^{(j)}(\widehat{\mathbf{g}}_{\mathbf{w}^{(t)}}^{(j)})^\top/\mathbf{Pr}_{z\sim\mathcal{N}(0,1)}[z \in \mathcal{E}_j]$.
7: $\quad\quad$ Find the top eigenvector $\mathbf{u}$ of $\widehat{\mathbf{M}}_{\mathbf{w}^{(t)}}$, then randomly pick $\mathbf{v}^{(t)}$ from $\{\pm\mathbf{u}\}$.
8: $\quad\quad$ $\mathbf{w}^{(k+1)} \leftarrow \mathrm{proj}_{\mathbb{B}^d}(\mathbf{w}^{(t)} - \eta_t\mathbf{v}^{(t)})$.
9: $\quad\quad$ $S^{\mathrm{sol}} \leftarrow S^{\mathrm{sol}} \cup \{\mathbf{w}^{(k+1)}\}$.
10: **Return:** $S^{\mathrm{sol}}$.

---

**Proposition 2.6** (Spectral Alignment)**.** *Fix parameters $B, L > 0$ and $\epsilon > 0$. Let $\mathcal{D}$ be a distribution over $(\mathbf{x}, y) \in \mathbb{R}^d \times \mathbb{R}$ with $\mathcal{D}_{\mathbf{x}} = \mathcal{N}(\mathbf{0}, \mathbf{I}_d)$. Let $(\mathbf{w}^*, \sigma) \in \mathbb{R}^d \times \mathcal{H}_\epsilon(B, L)$ be a pair of vector*

and monotone activation such that $\mathcal{L}_2(\mathbf{w}^*; \sigma) = \text{OPT}$. If $N \geq d^2\text{poly}(B, L, 1/\epsilon)$, then with probability at least $99\%$ the following holds: for any unit vector $\mathbf{w} \in \mathbb{R}^d$ satisfying $\sin\theta(\mathbf{w}, \mathbf{w}^*) \geq 40\sqrt{\text{OPT}}/\|\mathrm{T}_{\cos\theta(\mathbf{w},\mathbf{w}^*)}\sigma'\|_{L_2}$, the top eigenvector $\mathbf{u} \in \mathbb{R}^d$ of the empirical matrix $\widehat{\mathbf{M}}_{\mathbf{w}}$ returned by Algorithm 3 satisfies $\mathbf{u} \cdot \mathbf{w} = 0$ and $|\mathbf{u} \cdot \mathbf{w}^*| \geq (\sqrt{2}/2)\sin\theta(\mathbf{w}, \mathbf{w}^*)$.

The rest of this subsection develops the machinery required to prove the proposition above. Further details and complete proofs are deferred to Appendix C.

[ZWDD25] showed that (Proposition 2.2, [ZWDD25]) given the target activation $\sigma$, the gradient vector of the smoothed $L_2^2$ loss correlates strongly with $\mathbf{w}^*$: when $\sin\theta \geq 3\sqrt{\text{OPT}}/\|\mathrm{T}_{\cos\theta}\sigma'\|_{L_2}$, $\nabla_{\mathbf{w}}\mathcal{L}_{\cos\theta}(\mathbf{w}; \sigma) \cdot \mathbf{w}^* = \mathbf{E}_{(\mathbf{x},y)\sim\mathcal{D}}[y\mathrm{T}_{\cos\theta}\sigma'(\mathbf{w}\cdot\mathbf{x})\mathbf{x}^{\perp\mathbf{w}}] \cdot \mathbf{w}^* \geq (2/3)\|\mathrm{T}_{\cos\theta}\sigma'\|_{L_2}^2\sin^2\theta$. However, the structural result above requires the knowledge of $\sigma$, which is not applicable to our setting. Instead, we argue that using the vector $\mathbf{g}_{\mathbf{w}}(z)$ defined below, the top eigenvector of the matrix $\mathbf{M}_{\mathbf{w}} := \mathbf{E}_{z\sim\mathcal{N}(0,1)}[\mathbf{g}_{\mathbf{w}}(z)\mathbf{g}_{\mathbf{w}}(z)^\top]$ correlates with the "ideal" update direction $\mathbf{v}_{\mathbf{w}}^* := (\mathbf{w}^*)^{\perp\mathbf{w}}/\|(\mathbf{w}^*)^{\perp\mathbf{w}}\|_2$:

$$\mathbf{g}_{\mathbf{w}}(z) := \sum_{i=1}^{I} \frac{\mathbf{E}_{(\mathbf{x},y)\sim\mathcal{D}}[y\mathbf{x}^{\perp\mathbf{w}}\mathbb{1}\{\mathbf{w}\cdot\mathbf{x} \in \mathcal{E}_i\}]}{\mathbf{Pr}[\mathbf{w}\cdot\mathbf{x} \in \mathcal{E}_i]}\mathbb{1}\{z \in \mathcal{E}_i\},$$

$$\text{where } \begin{cases} \mathcal{E}_i = [a_i, a_{i+1}) = [-M' + (i-1)\Delta, -M' + i\Delta), \Delta = \epsilon^2/(B^2L^2); \\ M' = O(\sqrt{\log(BL/\epsilon)}), I = O(M'B^2L^2/\epsilon^2) = \tilde{O}(B^2L^2/\epsilon^2). \end{cases} \quad \text{(Grad)}$$

We first show that the target direction $\mathbf{v}_{\mathbf{w}}^*$ lies in the space of eigenvectors of large eigenvalues.

**Lemma 2.7.** *Let $\mathbf{g}_{\mathbf{w}}(z)$ be the vector defined in (Grad), let $\mathbf{M}_{\mathbf{w}} := \mathbf{E}_{z\sim\mathcal{N}}[\mathbf{g}_{\mathbf{w}}(z)\mathbf{g}_{\mathbf{w}}(z)^\top]$, and $\mathbf{v}_{\mathbf{w}}^* := (\mathbf{w}^*)^{\perp\mathbf{w}}/\|(\mathbf{w}^*)^{\perp\mathbf{w}}\|_2$. Suppose $\epsilon \leq \text{OPT}$ and $\sin\theta \geq 4\sqrt{\text{OPT}}/\|\mathrm{T}_{\cos\theta}\sigma'(z)\|_{L_2}$. Then, $(\mathbf{v}_{\mathbf{w}}^*)^\top\mathbf{M}_{\mathbf{w}}\mathbf{v}_{\mathbf{w}}^* \geq (1/16)\sin^2\theta\|\mathrm{T}_{\cos\theta}\sigma'\|_{L_2}^2$.*

*Proof Sketch of Lemma 2.7.* Define the correltation $K(h(\mathbf{w}\cdot\mathbf{x})) := \mathbf{E}_{(\mathbf{x},y)\sim\mathcal{D}}[y\mathbf{w}^* \cdot \mathbf{x}^{\perp\mathbf{w}}h(\mathbf{w}\cdot\mathbf{x})]$ where $h(z) \in \mathcal{H}' = \{h : h(z) = \sum_{i=1}^{I}h_i\mathbb{1}\{z \in \mathcal{E}_i\}, \|h\|_{L_2} = 1\}$, and $\mathcal{E}_i, i = 1, \ldots, I$, are the same intervals as in the definition of $\mathbf{g}_{\mathbf{w}}(z)$, (Grad). One can show that $K(h)$ can be written as: $K(h(\mathbf{w}\cdot\mathbf{x})) = \sin\theta\langle h(z), \mathbf{v}_{\mathbf{w}}^* \cdot \mathbf{g}_{\mathbf{w}}(z)\rangle_{L_2(\mathcal{N}(0,1))}$, where $\langle\cdot,\cdot\rangle_{L_2(\mathcal{N}(0,1))}$ is the inner product defined on the $L_2$ space with respect to the standard Gaussian measure. Therefore, by the duality of norms in the Hilbert spaces, we have $h^*(z) = \mathbf{v}_{\mathbf{w}}^* \cdot \mathbf{g}_{\mathbf{w}}(z)/\|\mathbf{v}_{\mathbf{w}}^* \cdot \mathbf{g}_{\mathbf{w}}(z)\|_{L_2}$ maximizes $K(h)$. Next, observe that by the definition of $\mathbf{M}_{\mathbf{w}}$ and $\mathbf{g}_{\mathbf{w}}(z)$, it holds $(\mathbf{v}_{\mathbf{w}}^*)^\top\mathbf{M}_{\mathbf{w}}\mathbf{v}_{\mathbf{w}}^* = (\|\mathbf{v}_{\mathbf{w}}^*\cdot\mathbf{g}_{\mathbf{w}}(z)\|_{L_2}/\sin\theta)K(h^*(\mathbf{w}\cdot\mathbf{x}))$. Now define $\widetilde{\mathrm{T}}_{\cos\theta}\sigma'(z) = \sum_{i=1}^{I}\widetilde{\mathrm{T}}_{\cos\theta}\sigma'(a_i)\mathbb{1}\{z \in \mathcal{E}_i\}$ where $\mathcal{E}_i = [a_i, a_{i+1}]$ as defined in (Grad). Then, $h_0(z) := \widetilde{\mathrm{T}}_{\cos\theta}\sigma'(z)/\|\widetilde{\mathrm{T}}_{\cos\theta}\sigma'\|_{L_2} \in \mathcal{H}'$ and by the maximality of $h^*$ we have $(\mathbf{v}_{\mathbf{w}}^*)^\top\mathbf{M}_{\mathbf{w}}\mathbf{v}_{\mathbf{w}}^* \geq (\|\mathbf{v}_{\mathbf{w}}^* \cdot \mathbf{g}_{\mathbf{w}}(z)\|_{L_2}/\sin\theta)K(h_0(\mathbf{w}\cdot\mathbf{x}))$. One can show that it holds $K(h_0(\mathbf{w}\cdot\mathbf{x})) \gtrsim \sin^2\theta\|\mathrm{T}_{\cos\theta}\sigma'\|_{L_2}$ and that $\|\mathbf{v}_{\mathbf{w}}^* \cdot \mathbf{g}_{\mathbf{w}}(z)\|_{L_2} = ((\mathbf{v}_{\mathbf{w}}^*)^\top\mathbf{M}_{\mathbf{w}}\mathbf{v}_{\mathbf{w}}^*)^{1/2}$. Thus, we obtain that $(\mathbf{v}_{\mathbf{w}}^*)^\top\mathbf{M}_{\mathbf{w}}\mathbf{v}_{\mathbf{w}}^* \gtrsim \sin^2\theta\|\mathrm{T}_{\cos\theta}\sigma'\|_{L_2}^2$. $\qquad\square$

The next lemma shows that any vector $\mathbf{u}$ that is orthogonal to $\mathbf{v}_{\mathbf{w}}^*$ has a small quadratic form.

**Lemma 2.8.** *For any unit vector $\mathbf{u} \in \mathbb{R}^d$ orthogonal to $\mathbf{v}_{\mathbf{w}}^*$, we have $\mathbf{u}^\top\mathbf{M}_{\mathbf{w}}\mathbf{u} \leq 2\text{OPT}$.*

Then we show that the top eigenvector $\mathbf{v}_{\mathbf{w}}$ of $\mathbf{M}_{\mathbf{w}}$ correlates strongly with the target direction $\mathbf{v}_{\mathbf{w}}^*$.

**Lemma 2.9.** *Let $\mathbf{v}_{\mathbf{w}}$ be the top eigenvector of $\mathbf{M}_{\mathbf{w}}$. If $\sin\theta \geq 40\sqrt{\text{OPT}}/\|\mathrm{T}_{\cos\theta}\sigma'\|_{L_2}$, then $\mathbf{v}_{\mathbf{w}} \cdot \mathbf{v}_{\mathbf{w}}^* \geq \sqrt{3}/2$.*

What remains is to determine the required number of samples so that we can have an accurate approximation of $\mathbf{M}_{\mathbf{w}}$, which is characterized in the following lemma:

**Lemma 2.10** (Sample Complexity). *Draw $N \geq d^2\text{poly}(1/\epsilon, B, L)$ i.i.d. samples from $\mathcal{D}$, and let $\widehat{\mathbf{M}}_{\mathbf{w}}$ be constructed as in Algorithm 3. Then, with probability at least $99\%$ for all $\mathbf{w} \in \mathbb{S}^{d-1}$, it holds $\|\widehat{\mathbf{M}}_{\mathbf{w}} - \mathbf{M}_{\mathbf{w}}\|_2 \leq \epsilon$.*

We can now proceed to the proof sketch of the main structural result (Proposition 2.6).

*Proof Sketch of Proposition 2.6.* Let $\theta = \theta(\mathbf{w}, \mathbf{w}^*)$ and assume that $\sin\theta \geq 40\sqrt{\text{OPT}}\|\mathrm{T}_{\cos\theta}\sigma'\|_{L_2}$. In Lemma 2.9 we proved that for any unit vector $\mathbf{w}$, one of the top eigenvectors $\mathbf{v}_{\mathbf{w}}$ of $\mathbf{M}_{\mathbf{w}}$ correlates

with $\mathbf{v_w^*}$: $\theta(\mathbf{v_w^*}, \mathbf{v_w}) \leq \pi/6$. Next, in Lemma 2.10, we proved that using $N = O(d^2)\text{poly}(1/\epsilon, B, L)$ samples, for any unit vector $\mathbf{w}$, the empirical matrix $\widehat{\mathbf{M}}_\mathbf{w}$ satisfies $\|\widehat{\mathbf{M}}_\mathbf{w} - \mathbf{M}_\mathbf{w}\|_2 \leq \epsilon$. Furthermore, one can show that the eigengap of $\mathbf{M}_\mathbf{w}$ is greater than $60\epsilon$ (Lemma C.8). Therefore, using Wedin's theorem (Fact C.7), we know that for any vector $\mathbf{w}$, the top eigenvector $\widehat{\mathbf{v}}_\mathbf{w}$ of $\widehat{\mathbf{M}}_\mathbf{w}$ satisfies $\theta(\mathbf{v_w}, \widehat{\mathbf{v}}_\mathbf{w}) \leq 1/59$, indicating $\theta(\widehat{\mathbf{v}}_\mathbf{w}, \mathbf{v_w^*}) \leq \theta(\widehat{\mathbf{v}}_\mathbf{w}, \mathbf{v_w}) + \theta(\mathbf{v_w}, \mathbf{v_w^*}) \leq \pi/4$. Therefore, let $\mathbf{u}$ be such eigenvector $\widehat{\mathbf{v}}_\mathbf{w}$ of $\widehat{\mathbf{M}}_\mathbf{w}$ that correlates positively with $\mathbf{v_w^*}$. Note that by definition of $\widehat{\mathbf{M}}_\mathbf{w}$, it must hold $\mathbf{u} \perp \mathbf{w}$. Thus we have $\mathbf{u} \cdot \mathbf{w}^* = \mathbf{u} \cdot (\cos(\theta)\mathbf{w} + \sin(\theta)\mathbf{v_w^*}) = \sin(\theta)\mathbf{u} \cdot \mathbf{v_w^*} \geq (\sqrt{2}/2)\sin(\theta)$. $\quad\square$

## 2.3 Proof Sketch of Main Theorem (Theorem 2.3)

Full details are deferred to Appendix D. In this proof sketch, we assume that we are initialized at the correct $\mathbf{w}^{(0)}$ that satisfies $\theta(\mathbf{w}^{(0)}, \mathbf{w}^*) \leq 1/M$, where $M$ is the minimum value that satisfies $\mathbf{E}_{z \sim \mathcal{N}}[(\sigma(z) - \sigma(M))^2 \mathbb{1}\{|z| \geq M\}] \leq C(\text{OPT} + \epsilon)$. In other words, according to Fact A.10, for any vector $\mathbf{w}$ such that $\theta(\mathbf{w}, \mathbf{w}^*) \leq \theta(\mathbf{w}^{(0)}, \mathbf{w}^*)$, it holds $\mathbf{E}_{(\mathbf{x},y) \sim \mathcal{D}}[(y - \sigma(\mathbf{w} \cdot \mathbf{x}))^2] \leq C\text{OPT} + \sin^2\theta\|\mathrm{T}_{\cos\theta}\sigma'\|_{L_2}^2$. Denote the angle between $\mathbf{w}^{(t)}$ and $\mathbf{w}^*$ by $\theta_t = \theta(\mathbf{w}^{(t)}, \mathbf{w}^*)$. Furthermore, let $\phi_t = \bar{\theta}(1 - c^2/32)^t$ and $\eta_t = c\sin\phi_t/4$ where $c = 1/4 \leq \sqrt{2}/2$. We can assume without loss of generality that $\epsilon \leq \text{OPT}$. According to Proposition 2.6, as long as $\sin\theta_t \geq 40\sqrt{\text{OPT}}/\|\mathrm{T}_{\cos\theta_t}\sigma'\|_{L_2}$, with probability at least 99% the vector $\mathbf{v}^{(t)}$ returned at Line (7) of Algorithm 3 satisfies $|\mathbf{v}^{(t)} \cdot \mathbf{w}^*| \geq c\sin\theta_t$ and $\mathbf{v}^{(t)} \cdot \mathbf{w}^{(t)} = 0$. We denote by $\mathcal{P}_t$ the event that $\mathbf{v}^{(t)}$ negatively correlates with $\mathbf{w}^*$. We consider the following event $\mathcal{R}_t := \{\sin\theta_t \geq C\sqrt{\text{OPT}}/\|\mathrm{T}_{\cos\theta_t}\sigma'\|_{L_2}\}$ where $C > 0$ is an absolute constant.

We show that conditioning on the events $\mathcal{R}_t, \mathcal{P}_t, t \in [T]$, for all $t \in T$, it holds that $\phi_t \geq \theta_t$. We use induction. By assumption, we have that $\phi_0 \geq \theta_0$. Next, we assume that $\phi_t \geq \theta_t$. Let us study the distance between $\mathbf{w}^{(t)}$ and $\mathbf{w}^*$ after one iteration from $t$ to $t+1$. Since $\mathbf{v}^{(t)}$ is orthogonal to $\mathbf{w}^{(t)}$, it must be $\|\mathbf{w}^{(t)} - \eta_t\mathbf{v}^{(t)}\|_2 \geq 1$, therefore, $\mathbf{w}^{(t+1)} = \text{proj}_{\mathbb{B}}(\mathbf{w}^{(t)} - \eta_t\mathbf{v}^{(t)})$. By the non-expansiveness of the projection operator, we have

$$\|\mathbf{w}^{(t+1)} - \mathbf{w}^*\|_2^2 = \|\text{proj}_{\mathbb{B}}(\mathbf{w}^{(t)} - \eta_t\mathbf{v}^{(t)}) - \mathbf{w}^*\|_2^2 \leq \|\mathbf{w}^{(t)} - \eta_t\mathbf{v}^{(t)} - \mathbf{w}^*\|_2^2$$
$$= \|\mathbf{w}^{(t)} - \mathbf{w}^*\|_2^2 + \eta_t^2\|\mathbf{v}^{(t)}\|_2^2 - 2\eta_t\mathbf{v}^{(t)}(\mathbf{w}^{(t)} - \mathbf{w}^*) = \|\mathbf{w}^{(t)} - \mathbf{w}^*\|_2^2 + \eta_t^2 + 2\eta_t\mathbf{v}^{(t)} \cdot \mathbf{w}^*. \quad (2)$$

Note that $\|\mathbf{w}^{(t)} - \mathbf{w}^*\|_2^2 - \|\mathbf{w}^{(t+1)} - \mathbf{w}^*\|_2^2 = 2(\cos\theta_{t+1} - \cos\theta_t)$ and using the identity about the sum of cosines, we have $4\sin((\theta_{t+1} - \theta_t)/2)\sin((\theta_{t+1} + \theta_t)/2) \leq \eta_t^2 + 2\eta_t\mathbf{v}^{(t)} \cdot \mathbf{w}^*$.

First, consider the case where $2\theta_t \geq \phi_t \geq \theta_t$. From Proposition 2.6, we have that $\mathbf{v}^{(t)} \cdot \mathbf{w}^* \leq -c\sin\theta_t$ where $c > 0$ is an absolute constant. Hence, since we chose $\eta_t = c\sin\phi_t/4$ it holds $\eta_t^2 + 2\eta_t\mathbf{v}^{(t)} \cdot \mathbf{w}^* \leq -c^2\sin\phi_t\sin\theta_t/4$. Therefore, in this case, $\theta_{t+1} \leq \theta_t$, hence $\sin((\theta_{t+1} + \theta_t)/2) \leq \sin\theta_t$. Using the inequality $x/4 \leq \sin x \leq x$ for $x \in (0, \pi/2)$, we have $\theta_{t+1} \leq \theta_t(1 - c^2/32) \leq \phi_{t+1}$.

For the next case where $\phi_t \geq 2\theta_t$, if $\theta_{t+1} \leq \theta_t$ then $\theta_{t+1} \leq \phi_{t+1}$, so we need to consider the case where $\theta_{t+1} \geq \theta_t$. We need to bound the maximum increase of $\theta_{t+1}$. By the triangle inequality and the non-expansiveness of the projection operator, it holds

$$2\sin(\theta_{t+1}/2) = \|\mathbf{w}^{(t+1)} - \mathbf{w}^*\|_2 = \|\text{proj}_{\mathbb{B}}(\mathbf{w}^{(t)} - \eta_t\mathbf{v}^{(t)}(\mathbf{w}^{(t)})) - \mathbf{w}^*\|_2 \leq \|\mathbf{w}^{(t)} - \eta_t\mathbf{v}^{(t)} - \mathbf{w}^*\|_2$$
$$\leq \|\mathbf{w}^{(t)} - \mathbf{w}^*\|_2 + \eta_t\|\mathbf{v}^{(t)}\|_2 = 2\sin(\theta_t/2) + c\sin\phi_t/4 .$$

From the assumption we have $\theta_t \leq \phi_t/2$, therefore, choosing $c \leq 1/4$ and since $\sin(x) \leq x$ for $x \in (0, \pi/2)$ we have that $\sin(\theta_{t+1}/2) \leq \sin(\phi_t/4) + c\sin\phi_t/8 \leq 9\phi_t/32$. Therefore, since $(5/8)x \leq \sin x$ when $x \in (0, \pi/2)$ we have $\sin(\theta_{t+1}/2) \geq (5/16)\theta_{t+1}$ and thus, $\theta_{t+1} \leq (9/10)\phi_t \leq (1 - c^2/32)\phi_t \leq \phi_{t+1}$. This completes the induction argument that $\theta_{t+1} \leq \phi_{t+1}$.

Conditioning on the event that all $\mathcal{P}_t$'s are satisfied for $t \in [T]$, one can show that due to the contraction of $\theta_t$, i.e., $\theta_t \leq (1 - c^2/32)^t\phi_0 \leq (1 - c^2/32)^t\theta_0$, the algorithm will arrive at a vector $\widehat{\mathbf{w}}$ such that $\widehat{\theta} := \theta(\widehat{\mathbf{w}}, \mathbf{w}^*)$ satisfies $\sin\widehat{\theta} \leq \sqrt{\text{OPT}}/\|\mathrm{T}_{\cos\widehat{\theta}}\sigma'\|_{L_2}$ in at most $T_1 \leq T = C'\log(1/(L\epsilon))$ iterations, for some large absolute value $C'$. This implies that $\widehat{\mathbf{w}}$ satisfies $\mathbf{E}_{(\mathbf{x},y) \sim \mathcal{D}}[(y - \sigma(\widehat{\mathbf{w}} \cdot \mathbf{x}))^2] \leq C\text{OPT} + \epsilon$. Let $T_1 \leq T$ be the first time that $\mathcal{R}_t$ is not satisfied.

Next, we need to bound the probability that all $\mathcal{P}_t$ (correct direction choices) are satisfied. The events $\mathcal{P}_t$ are independent, and each occurs with probability at least $1/2$. The probability of $T_1$ such events

occurring is at least $(1/2)^{T_1}$. Since $T_1 \leq T = O(\log(L/\epsilon))$, this probability is bounded below by $\delta' = (1/2)^T = \text{poly}(\epsilon, 1/L)$. If we rerun the algorithm $K = O((1/\delta') \log(1/\delta))$ times (Line 3 of Algorithm 3), by standard Chernoff bounds, with probability at least $1 - \delta$, there will be at least one run where all $\mathcal{P}_t$ are satisfied for all $t \in [T_1]$.

Next, we show that given all the constructed candidate solutions, the testing subroutine (Algorithm 5) with high probability returns an activation and direction pair that achieves $O(\text{OPT}) + \epsilon$ error.

**Lemma 2.11** (Learning the Predictor and Testing). *Algorithm 5 given $n = \text{poly}(B, L, 1/\epsilon)$ samples and a set $S^{\text{sol}}$ of $\text{poly}(B, L, 1/\epsilon)$ vectors, with probability at least $99\%$ returns a solution pair $(\widehat{u}_{\widehat{\mathbf{w}}}, \widehat{\mathbf{w}})$, with $\widehat{u}_{\widehat{\mathbf{w}}}$ being Lipschitz and monotone, and $\widehat{\mathbf{w}} \in S^{\text{sol}}$, such that $\mathbf{E}_{(\mathbf{x},y) \sim \mathcal{D}}[(\widehat{u}_{\widehat{\mathbf{w}}}(\widehat{\mathbf{w}} \cdot \mathbf{x}) - y)^2] \leq C \min_{\mathbf{w} \in S^{\text{sol}}} \mathbf{E}_{(\mathbf{x},y) \sim \mathcal{D}}[(\sigma(\mathbf{w} \cdot \mathbf{x}) - y)^2] + \epsilon$ for some universal constant $C$.*

Using Lemma 2.11 and the fact that there exists $\widehat{\mathbf{w}} \in S^{\text{sol}}$ that satisfies $\mathbf{E}_{(\mathbf{x},y) \sim \mathcal{D}}[(y - \sigma(\widehat{\mathbf{w}} \cdot \mathbf{x}))^2] \leq C\text{OPT} + \epsilon$, Algorithm 5 with $n = \text{poly}(B, L, 1/\epsilon)$ new samples returns with probability at least $99\%$ a solution pair $(\widehat{u}_{\widehat{\mathbf{w}}}, \widehat{\mathbf{w}})$ where $\widehat{u}_{\widehat{\mathbf{w}}}$ is Lipschitz and monotone and $\widehat{\mathbf{w}} \in S^{\text{sol}}$ such that $\mathbf{E}_{(\mathbf{x},y) \sim \mathcal{D}}[(\widehat{u}_{\widehat{\mathbf{w}}}(\widehat{\mathbf{w}} \cdot \mathbf{x}) - y)^2] \leq C\text{OPT} + \epsilon$. This completes the proof.

# 3 Conclusions and Future Directions

This work resolves a recognized open problem in the algorithmic theory of learning SIMs, by developing the first polynomial-time, constant-factor robust SIM learner for monotone activations under Gaussian inputs. At the technical level, our alignment-based spectral framework bypasses the limitations of gradient-based methods and leads to a constant-factor approximation ratio—independent of dimension, radius of optimization, or noise level. An interesting direction for future work is to generalize our algorithmic guarantees beyond Gaussian marginals, e.g., to all isotropic log-concave distributions. This question is open even for agnostic learning of a general (i.e., with arbitrary bias) halfspace or ReLU, where all known efficient constant-factor learners critically leverage Gaussianity.

## Acknowledgments and Disclosure of Funding

PW was supported in part by NSF Award DMS-2023239 and by the Air Force Office of Scientific Research under award number FA9550-24-1-0076. NZ was supported in part by NSF Medium Award CCF-2107079 and ONR award number N00014-25-1-2268. ID was supported in part by NSF Medium Award CCF-2107079, ONR award number N00014-25-1-2268, and an H.I. Romnes Faculty Fellowship. JD was supported in part by the Air Force Office of Scientific Research under award number FA9550-24-1-0076, by the U.S. Office of Naval Research under contract number N00014-22-1-2348, and by the NSF CAREER Award CCF-2440563. Any opinions, findings and conclusions or recommendations expressed in this material are those of the author(s) and do not necessarily reflect the views of the U.S. Department of Defense.

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

# Appendix

**Organization** The appendix is organized as follows: in Appendix A we present basic properties of the Ornstein–Uhlenbeck-semigroup and discuss the ($\epsilon$-Extended) $(B, L)$-Regular activation class; in Appendix B we provide omitted details and proofs of Section 2.1; in Appendix C we provide the the full version of Section 2.2 on the spectral subroutine; in Appendix D we complete the details omitted from Section 2.3 on the proof of the main theorem Theorem 2.3.

## A Technical Backgroound

### A.1 Ornstein–Uhlenbeck Semigroup

The Ornstein–Uhlenbeck semigroup is extensively used in our work. Let us first give a formal definition of the Ornstein–Uhlenbeck semigroup and then record the properties that will be used frequently throughout this paper.

**Definition A.1** (Ornstein–Uhlenbeck Semigroup). *Let $\rho \in (0, 1)$, $g \in L_2(\mathcal{N})$. The Ornstein–Uhlenbeck semigroup, denoted by $\mathrm{T}_\rho$, is a linear operator that maps $g \in L_2(\mathcal{N})$ to the function $\mathrm{T}_\rho g$ defined as:*

$$(\mathrm{T}_\rho g)(\mathbf{x}) \coloneqq \mathop{\mathbf{E}}_{\mathbf{z} \sim \mathcal{N}} \left[ g(\rho \mathbf{x} + \sqrt{1 - \rho^2} \mathbf{z}) \right] .$$

*For simplicity of notation, we write $\mathrm{T}_\rho g(\mathbf{x})$ instead of $(\mathrm{T}_\rho g)(\mathbf{x})$.*

The following fact summarizes useful properties of the Ornstein–Uhlenbeck semigroup.

**Fact A.2** (see, e.g., [Bog98, O'D14]). *Let $f, g \in L_2(\mathcal{N})$.*

1. *For any $f, g \in L_2$ and any $t > 0$, $\mathbf{E}_{\mathbf{x} \sim \mathcal{N}}[(\mathrm{T}_t f(\mathbf{x})) g(\mathbf{x})] = \mathbf{E}_{\mathbf{x} \sim \mathcal{N}}[(\mathrm{T}_t g(\mathbf{x})) f(\mathbf{x})]$ .*

2. *For any $g : \mathbb{R}^d \to \mathbb{R}$, $g \in L_2$, all of the following statements hold.*

   (a) *For any $t, s > 0$, $\mathrm{T}_t \mathrm{T}_s g = \mathrm{T}_{ts} g$.*
   (b) *For any $\rho \in (0, 1)$, $\mathrm{T}_\rho g(\mathbf{x})$ is differentiable at every point $\mathbf{x} \in \mathbb{R}^d$.*
   (c) *For any $\rho \in (0, 1)$, $\mathrm{T}_\rho g(\mathbf{x})$ is $\|g\|_{L_\infty}/(1 - \rho^2)^{1/2}$-Lipschitz, i.e., $\|\nabla \mathrm{T}_\rho g(\mathbf{x})\|_{L_\infty} \leq \|g\|_{L_\infty}/(1 - \rho^2)^{1/2}, \forall \mathbf{x} \in \mathbb{R}^d$.*
   (d) *For any $\rho \in (0, 1)$, $\mathrm{T}_\rho g(\mathbf{x}) \in \mathcal{C}^\infty$.*
   (e) *For any $p \geq 1$, $\mathrm{T}_\rho$ is nonexpansive with respect to the norm $\| \cdot \|_{L_p}$, i.e., $\|\mathrm{T}_\rho g\|_{L_p} \leq \|g\|_{L_p}$.*
   (f) *$\|\mathrm{T}_\rho g(\mathbf{x})\|_{L_2}$ is non-decreasing w.r.t. $\rho$.*
   (g) *If $g$ is, in addition, a differentiable function, then for all $\rho \in (0, 1)$, it holds that: $\nabla_{\mathbf{x}} \mathrm{T}_\rho g(\mathbf{x}) = \rho \mathrm{T}_\rho \nabla_{\mathbf{x}} g(\mathbf{x})$, for any $\mathbf{x} \in \mathbb{R}^d$.*

The Ornstein–Uhlenbeck semigroup induces an operator $\mathrm{L}$ applying to functions $f \in L_2(\mathcal{N})$, defined below.

**Definition A.3** (Definition 11.24 in [O'D14]). *The Ornstein–Uhlenbeck operator is a linear operator that applies to functions $f \in L_2(\mathcal{N})$, defined by $\mathrm{L}f = \frac{\mathrm{d}\mathrm{T}_\rho f}{\mathrm{d}\rho} |_{\rho=1}$, provided that $\mathrm{L}f$ exists.*

**Fact A.4** ([O'D14]). *Let $f, g \in L_2(\mathcal{N})$, $\rho \in (0, 1)$. Then:*

1. *([Proposition 11.27]) $\frac{\mathrm{d}\mathrm{T}_\rho f}{\mathrm{d}\rho} = \frac{1}{\rho} \mathrm{L}\mathrm{T}_\rho f = \frac{1}{\rho} \mathrm{T}_\rho \mathrm{L}f$.*

2. *([Proposition 11.28]) $\mathbf{E}_{\mathbf{x} \sim \mathcal{N}}[f(\mathbf{x}) \mathrm{L}\mathrm{T}_\rho g(\mathbf{x})] = \mathbf{E}_{\mathbf{x} \sim \mathcal{N}}[\nabla f(\mathbf{x}) \nabla \mathrm{T}_\rho g(\mathbf{x})]$.*

The following fact from [ZWDD25] shows that the error incurred by smoothing is controlled by the smoothing parameter $\rho$ and the $L_2^2$ norm of the gradient:

**Fact A.5** (Lemma B.5 in [ZWDD25]). *Let $f \in L_2(\mathcal{N})$ be a continuous and (almost everywhere) differentiable function. Then $\mathbf{E}_{\mathbf{x} \sim \mathcal{N}}[(\mathrm{T}_\rho f(\mathbf{x}) - f(\mathbf{x}))^2] \leq 3(1 - \rho) \mathbf{E}_{\mathbf{x} \sim \mathcal{N}}[\|\nabla f(\mathbf{x})\|_2^2]$.*

## A.2 Regular Activations

Our main algorithm robustly learns SIMs whose activations are monotone and approximately regular. The definition of regularity and approximate regularity is given below.

**Definition A.6** (($B, L$)-Regular Activations, Definition 3.1 of [ZWDD25]). *Given parameters $B, L > 0$, we define the class of $(B, L)$-Regular activations, denoted by $\mathcal{H}(B, L)$, as the class containing all functions $\sigma : \mathbb{R} \to \mathbb{R}$ such that 1) $\|\sigma\|_{L_\infty} \leq B$ and 2) $\|\sigma'\|_{L_2} \leq L$. Given $\epsilon > 0$, we define the class of $\epsilon$-Extended $(B, L)$-Regular activations, denoted by $\bar{\mathcal{H}}_\epsilon(B, L)$, as the class containing all activations $\sigma_1 : \mathbb{R} \to \mathbb{R}$ for which there exists $\sigma_2 \in \mathcal{H}(B, L)$ such that $\|\sigma_1 - \sigma_2\|_{L_2}^2 \leq \epsilon$.*

**Examples of Monotone Regular Activations**    The class of Monotone $\epsilon$-extended $(B, L)$-regular activations is a broad family of functions, with illustrative examples provided in the fact below.

**Fact A.7** (Examples of $\epsilon$-Extended Regular Functions (Lemmas C.9 and C.12 in [ZWDD25])). *The following function classes are $\epsilon$-Extended Regular:*

1. *If $\sigma$ satisfies $\mathbf{E}_{z \sim \mathcal{N}}[\sigma(z)^{2+\zeta}] \leq B_\sigma$ for some $\zeta > 0$ and $\sigma$ is monotone, then $\sigma \in \mathcal{H}_\epsilon(c_1 D, c_2 D^4/\epsilon^2)$ where $D = (B_\sigma/4\epsilon)^{1/\zeta}$ and $c_1, c_2$ are absolute constants.*

2. *If $\sigma$ is $b$-Lipschitz and recentered so that $\sigma(0) = 0$, then $\sigma \in \mathcal{H}_\epsilon(cb \log^{1/2}(b/\epsilon), b)$, where $c$ is an absolute constant.*

3. *If $\sigma = \sigma_1 + \Phi$, where $\sigma_1 \in \mathcal{H}_\epsilon(B, L)$, $|\Phi(z)| \leq A$, and*

$$\Phi(z) = \sum_{i=1}^{m} A_i \mathbb{1}\{z \geq t_i\} + A_0 : A_0 \in \mathbb{R}; A_i > 0, \forall i \in [m]; m < \infty$$

   *then $\sigma \in \mathcal{H}_\epsilon(B + A, L + \max\{A^2 L/\sqrt{\epsilon}, A^4/\epsilon\})$.*

In particular, using Fact A.7, it follows that

1. General ReLUs $\sigma(z) = \max\{0, z + t\}$, $t \in \mathbb{R}$ are regular; namely, $\sigma \in \mathcal{H}_\epsilon(c \log^{1/2}(1/\epsilon), 1)$;

2. General Halfspaces $\sigma(z) = \mathbb{1}\{z + t \geq 0\}$, $t \in \mathbb{R}$, are regular; namely, $\sigma \in \mathcal{H}_\epsilon(1, 1/\epsilon)$.

In the next lemma, we show that, in fact, all monotone functions $f \in L_2(\mathcal{N})$ are $\epsilon$-Extended $(B, L)$-Regular. However, the parameters $B(\epsilon), L(\epsilon)$ depend on $f$ and $\epsilon$, which might not be a polynomial of $1/\epsilon$. Therefore, the lemma below does not violate the information-theoretic lower bound in [ZWDD25].

**Lemma A.8.** *Let $f$ be a monotone activation in $L_2(\mathcal{N})$. Then, for any $\epsilon > 0$, there exists $C(\epsilon) > 0$ so that $f \in \mathcal{H}_{\epsilon/2}(\mathrm{poly}(C(\epsilon)/\epsilon), \mathrm{poly}(C(\epsilon)/\epsilon))$.*

*Proof of Lemma A.8.* Using the assumption that $f \in L_2(\mathcal{N})$, we have that $\|f\|_{L_2}^2 \leq c < \infty$ for some $c > 0$. Therefore, we have that

$$\|f\|_{L_2}^2 = \int_0^\infty \mathbf{Pr}[f^2(z) \geq t] \mathrm{d}t \leq c .$$

Note that the function $\mathbf{Pr}[f^2(z) \geq t]$ is a nonnegative function of $t$. Therefore the sequence $a_n = \int_0^n \mathbf{Pr}[f^2(z) \geq t] \mathrm{d}t$ is non-decreasing for any $n \in N$ and by assumption the limit of the sequence $a_n$ as $n \to \infty$ exists. Therefore from the definition of the convergence of the limits, for any $\epsilon' \in (0, 1)$ there exists $n_{\epsilon'} \in \mathbb{N}$ so that for any $n \geq n_{\epsilon'}$, we have

$$\int_n^\infty \mathbf{Pr}[f^2(z) \geq t] \mathrm{d}t = \left| \int_0^\infty \mathbf{Pr}[f^2(z) \geq t] \mathrm{d}t - \int_0^n \mathbf{Pr}[f^2(z) \geq t] \mathrm{d}t \right| \leq \epsilon' .$$

Furthermore, note that for $f_2 = \text{sign}(f)\min(|f|, |f(n_{\epsilon'})|)$, we have that

$$\|f - f_2\|_{L_2}^2 = \int_0^\infty \mathbf{Pr}[|f(z) - f_2(z)|^2 \geq t]\mathrm{d}t$$

$$= \int_0^\infty \mathbf{Pr}[|f(z) - f_2(z)|^2 \geq t, |f| \geq |f(n_\epsilon)|]\mathrm{d}t$$

$$\leq \int_n^\infty \mathbf{Pr}[f(z)^2 \geq t]\mathrm{d}t \leq \epsilon' \ .$$

By applying Part 1 of Fact A.7 to $f_2$ and choose $\epsilon' = \epsilon/2$, we get the result for $C(\epsilon) = |f_2(n_{\epsilon'})|$. $\quad\square$

**Truncation of Regular Activations**    For a target activation $\sigma \in \mathcal{H}_\epsilon(B, L)$, we can make simplifying assumptions that come at no loss of generality. The first assumption is that there exists a finite parameter $\bar{M}$ such that outside the interval $[-\bar{M}, \bar{M}]$, the derivative $\sigma'$ of the target activation $\sigma$ is zero. In this case, we call the interval $[-\bar{M}, \bar{M}]$ the support of $\sigma'$ and say the support of $\sigma'$ is bounded by $\bar{M}$. Another way of viewing this assumption is as saying that $\sigma$ is "capped" (or truncated) at $\sigma(\bar{M})$. It turns out that such a truncation ensures all $O(\text{OPT})$ solutions are unaffected. Similarly, the labels can be truncated in the interval $[-B, B]$, and, as a result, we can assume w.l.o.g. that $|y| \leq B$. A formal statement is provided below.

**Fact A.9** (Lemma C.6, C.7 in [ZWDD25]). *Suppose that the target activation $\sigma \in \mathcal{H}_\epsilon(B, L)$. Let $\bar{y} = \text{sign}(y)\min\{|y|, B\}$. Then, $\mathbf{E}_{(\mathbf{x}, y) \sim \mathcal{D}}[(\bar{y} - \sigma(\mathbf{w}^* \cdot \mathbf{x}))^2] \leq \text{OPT} + \epsilon$ and for any $\hat{\mathbf{w}}$ such that $\mathbf{E}_{(\mathbf{x}, y) \sim \mathcal{D}}[(\bar{y} - \sigma(\hat{\mathbf{w}} \cdot \mathbf{x}))^2] = O(\text{OPT}) + \epsilon$, we have $\mathbf{E}_{(\mathbf{x}, y) \sim \mathcal{D}}[(y - \sigma(\hat{\mathbf{w}} \cdot \mathbf{x}))^2] = O(\text{OPT}) + \epsilon$.*

*Moreover, there exists $\tilde{\sigma} \in \mathcal{H}(B, L)$ such that $\|\tilde{\sigma} - \sigma\|_{L_2}^2 \leq \epsilon$, with support of $\tilde{\sigma}'$ bounded by $\bar{M} \leq \sqrt{2\log(4B^2/\epsilon) - \log\log(4B^2/\epsilon)}$. If $\hat{\mathbf{w}}$ satisfies $\mathbf{E}_{(\mathbf{x}, y) \sim \mathcal{D}}[(y - \tilde{\sigma}(\hat{\mathbf{w}} \cdot \mathbf{x}))^2] \leq O(\text{OPT}) + \epsilon$, then also $\mathbf{E}_{(\mathbf{x}, y) \sim \mathcal{D}}[(y - \sigma(\hat{\mathbf{w}} \cdot \mathbf{x}))^2] \leq O(\text{OPT}) + \epsilon$. Thus, one can replace $\sigma$ with $\tilde{\sigma}$ and $y$ by $\bar{y}$, and assume w.l.o.g. that the support of $\sigma'$ is bounded by $\bar{M}$ and $|y| \leq B$.*

**Error Bound**    In the next fact, we show that for any function $\sigma \in \mathcal{H}_\epsilon(B, L)$, we can bound the $L_2^2$ loss $\mathcal{L}_2(\mathbf{w}; \sigma)$ at vector $\mathbf{w}$ by $\sin^2\theta\|\mathrm{T}_{\cos\theta}\sigma'\|_{L_2}^2$, where $\theta := \theta(\mathbf{w}, \mathbf{w}^*)$.

**Fact A.10** (Error Bound, Lemma D.8 + Proposition 4.5 in [ZWDD25]). *Suppose that $\mathbf{E}_{(\mathbf{x}, y) \sim \mathcal{D}}[(\sigma(\mathbf{w}^* \cdot \mathbf{x}) - y)^2] = \text{OPT}$ holds for a monotone activation $\sigma \in \mathcal{H}_\epsilon(B, L)$ and a unit vector $\mathbf{w}^* \in \mathbb{R}^d$ and suppose the support of $\sigma'$ is bounded by $M > 0$. Given any unit vector $\mathbf{w} \in \mathbb{R}^d$, let $\theta := \theta(\mathbf{w}, \mathbf{w}^*)$. Then, if $\theta \leq c/M$ for an absolute constant $c$, we have $\mathbf{E}_{(\mathbf{x}, y) \sim \mathcal{D}}[(\sigma(\mathbf{w} \cdot \mathbf{x}) - y)^2] \leq C\text{OPT} + C\sin^2\theta\|\mathrm{T}_{\cos\theta}\sigma'\|_{L_2}^2$ for a universal constant $C > 1$.*

# B    Omitted Proofs and Details from Section 2.1

Note that the methodology of our initialization algorithm is to find a threshold $t$ such that after transforming the labels $y$ to $\mathcal{T}(y, t) = \mathbb{1}\{y \geq t\}$, we can reduce the regression problem to a robust halfspace learning problem and find a vector $\mathbf{w}^{(0)}$ satisfying $\theta(\mathbf{w}^{(0)}, \mathbf{w}) \leq 1/M$ via the robust halfspace learning algorithm from [DKTZ22b], where $M > 0$ is the smallest parameter such that $\mathbf{E}_{z \sim \mathcal{N}}[(\sigma(z) - \sigma(M))^2\mathbb{1}\{|z| \geq M\}] \leq C(\text{OPT} + \epsilon)$. The following fact guarantees the existence of a target threshold that ensures the desired initialization.

**Fact B.1** (Proposition F.19 and Lemma F.21 in [ZWDD25]). *Let $\sigma(\mathbf{w}^* \cdot \mathbf{x})$ be an optimal hypothesis that satisfies $\mathbf{E}_{(\mathbf{x}, y) \sim \mathcal{D}}[(y - \sigma(\mathbf{w}^* \cdot \mathbf{x}))^2] \leq \epsilon_0$ for $\epsilon_0 := \text{OPT} + \epsilon > 0$, where $\sigma$ is a non-decreasing $\epsilon$-Extended $(B, L)$-regular function. Suppose the constant hypothesis $\sigma(z) = c$ is not a constant factor approximate solution for any $c \in \mathbb{R}$. Let $C_1 > 0$ be a sufficiently large absolute constant. Then there exists a minimum $M > 0$ that satisfies $\|(\sigma(z) - \sigma(M))\mathbb{1}\{z \geq M\}\|_{L_2}^2 \leq C_1\epsilon_0$, such that $\mathbf{Pr}[\mathbb{1}\{y \geq \sigma(M)\} \neq \mathbb{1}\{\mathbf{w}^* \cdot \mathbf{x} \geq M\}] \leq \mathbf{Pr}[\mathbf{w}^* \cdot \mathbf{x} \geq M]/C_2$, where $C_2 = \sqrt{C_1/5}$.*

Suppose we are given the threshold $t = \sigma(M)$ with $M$ being the minimum value satisfying Fact B.1. After transforming the labels to $\tilde{y} = \mathcal{T}(y; t) = \mathbb{1}\{y \geq t\}$, we can apply the algorithm from [DKTZ22b, ZWDD25] to find a vector $\mathbf{w}^0$ such that $\theta(\mathbf{w}^0, \mathbf{w}^*) \leq 1/M$.

**Fact B.2** (Proposition F.19, Fact F.20 in [ZWDD25]). *Let $h^*(\mathbf{x}) = \mathbb{1}\{\mathbf{w}^* \cdot \mathbf{x} \geq M\}$ be a target Gaussian halfspace, i.e., $\mathbf{x} \sim \mathcal{D}_{\mathbf{x}}$ and $\mathcal{D}_{\mathbf{x}}$ is a standard Gaussian distribution. Let $(\mathbf{x}, \tilde{y}) \sim \mathcal{D}$ be a distribution of labeled samples with $\mathrm{OPT}'$- adversarial label noise—meaning that $\mathbf{Pr}[\tilde{y} \neq h^*(\mathbf{x})] \leq \mathrm{OPT}'$, with the noise rate satisfying $\mathrm{OPT}' \leq (1/C_2)(\exp(-M^2/2)/M) \approx (1/C_2)\mathbf{Pr}[h^*(\mathbf{x}) = 1]$, where $C_2$ is a large absolute constant. Then, there is an algorithm that uses $O(d/\epsilon^2 \log(1/\delta))$ samples and returns with probability at least $1 - \delta$ a vector $\mathbf{w}$ such that $\theta(\mathbf{w}, \mathbf{w}^*) \leq \min(\pi/16, 1/M)$.*

Therefore, what remains is to find such a threshold $t$. Since the target activation $\sigma$ is unknown, it is not possible to find $M$ and calculate $\mathbf{Pr}[z \geq M]$ and estimate $\mathbf{Pr}[\mathbb{1}\{y \geq \sigma(M)\} \neq \mathbb{1}\{z \geq M\}]$. Our strategy is to show that we can discretize the labels $y$ and the target activation $\sigma$ so that they only take a small number of values, which then implies that the target threshold $t = \sigma(M)$ must be an element of a small set. Then, we can brute force iterates through all the possible values (we show there are polynomially many), and run the robust halfspace learning algorithm from Fact B.2 on each of the label transformations $\mathcal{T}(y; t_i)$, $t_i = i\sqrt{\epsilon}$. Formally, we have:

**Claim B.3.** *Suppose $\|y - \sigma(\mathbf{w}^* \cdot \mathbf{x})\|_{L_2}^2 \leq \epsilon_0$ for some $\epsilon_0 > 0$. Define the truncation operator* $\mathrm{trunc}(\cdot)$ *by* $\mathrm{trunc}(z) = -B + i\sqrt{\epsilon}$ *if* $z \in [i\sqrt{\epsilon}, (i+1)\sqrt{\epsilon})$, $i \in [2B/\sqrt{\epsilon}]$. *Then* $\|\mathrm{trunc}(y) - \mathrm{trunc}(\sigma(\mathbf{w}^* \cdot \mathbf{x}))\|_{L_2}^2 \leq 9(\epsilon_0 + \epsilon)$.

*Proof.* Since $\sigma$ is $(B, L)$-regular (thus $\|\sigma\|_{L_\infty} \leq B$) and since w.l.o.g. $|y| \leq B$ (Fact A.9), we have $\|\mathrm{trunc}(y) - y\|_{L_2} \leq \sqrt{\epsilon_0}$ and $\|\sigma(\mathbf{w}^* \cdot \mathbf{x}) - \mathrm{trunc}(\sigma(\mathbf{w}^* \cdot \mathbf{x}))\|_{L_2} \leq \sqrt{\epsilon_0}$. Direct calculation yields:

$$\|\mathrm{trunc}(y) - \mathrm{trunc}(\sigma(\mathbf{w}^* \cdot \mathbf{x}))\|_{L_2} = \|\mathrm{trunc}(y) - y + y - \sigma(\mathbf{w}^* \cdot \mathbf{x}) + \sigma(\mathbf{w}^* \cdot \mathbf{x}) - \mathrm{trunc}(\sigma(\mathbf{w}^* \cdot \mathbf{x}))\|_{L_2}$$
$$\leq 2\sqrt{\epsilon} + \sqrt{\epsilon_0}.$$

Thus, we have $\|\mathrm{trunc}(y) - \mathrm{trunc}(\sigma(\mathbf{w}^* \cdot \mathbf{x}))\|_{L_2}^2 \leq 9(\epsilon + \epsilon_0)$. $\qquad\square$

Now we prove Lemma 2.5, restated below.

**Lemma 2.5** (Initialization). *Let $\sigma(\mathbf{w}^* \cdot \mathbf{x})$ be a hypothesis that satisfies $\mathbf{E}_{(\mathbf{x},y) \sim \mathcal{D}}[(y - \sigma(\mathbf{w}^* \cdot \mathbf{x}))^2] \leq \mathrm{OPT} + \epsilon$, where $\sigma$ is a non-decreasing $\epsilon$-Extended $(B, L)$-Regular function. Suppose that no constant hypothesis, i.e., function of the form $\sigma(z) = c$ for any $c \in \mathbb{R}$, is a constant-factor approximate solution. Let $C > 1$ be a large absolute constant and let $M > 0$ be the smallest parameter such that $\mathbf{E}_{z \sim \mathcal{N}}[(\sigma(z) - \sigma(M))^2 \mathbb{1}\{z \geq M\}] \leq C(\mathrm{OPT} + \epsilon)$. Then Algorithm 2, using $O(d/\epsilon^2 \log(B/\epsilon))$ samples, with probability at least 99%, returns a list $S^{\mathrm{ini}}$ of $O(B/\sqrt{\epsilon})$ vectors that contains a vector $\mathbf{w}^{(0)}$ such that $\theta(\mathbf{w}^{(0)}, \mathbf{w}^*) \leq \min(1/M, \pi/16)$.*

*Proof of Lemma 2.5.* By Claim B.3, we can discretize the labels and the target activation $\sigma$ so that it only takes a finite number of values: $i\sqrt{\epsilon}$, $i \in [\lceil 2B/\sqrt{\epsilon}\rceil + 1]$, while only inducing a small error. By Fact B.1, there exits a threshold $t \in \{i\sqrt{\epsilon}\}_{i=0}^{B/\sqrt{\epsilon}}$ so that $\mathbf{E}_{z \sim \mathcal{N}}[(\sigma(z) - t)^2 \mathbb{1}\{\sigma(z) \geq t\}] \leq C(\mathrm{OPT} + \epsilon)$ and $\mathbf{Pr}[\mathbb{1}\{y \geq t\} \neq \mathbb{1}\{\sigma(\mathbf{w}^* \cdot \mathbf{x}) \geq t\}] \leq \mathbf{Pr}[\sigma(\mathbf{w}^* \cdot \mathbf{x}) \geq t]/C_1$, where $C, C_1$ are large absolute constants. Then by Fact B.2 we know that using the algorithm from [DKTZ22b] we will obtain a vector $\mathbf{w}$ such that $\theta(\mathbf{w}, \mathbf{w}^*) \leq 1/M$ where $M = \sigma^{-1}(t)$. The algorithm requires $O(d/\epsilon^2 \log(1/\delta'))$ samples to return such a vector with probability $1 - \delta'$. Since we run the algorithm $B/\sqrt{\epsilon}$ times, let $\delta' = \sqrt{\epsilon}\delta/B$, and after a union bound we get that with $O(d/\epsilon^2 \log(B/(\epsilon\delta)))$ samples, the algorithm succeeds with probability $1 - \delta$ for all the iterations. Setting $\delta = 0.01$ completes the proof. $\qquad\square$

# C   Full Version of Section 2.2

In this section, we present and prove our main structural result (Proposition C.1). We show that—even though the target activation $\sigma$ is unknown—we can identify a vector that has a strong correlation with an 'ideal descent direction' $\mathbf{v}_{\mathbf{w}}^* := (\mathbf{w}^*)^{\perp \mathbf{w}}/\|(\mathbf{w}^*)^{\perp \mathbf{w}}\|_2$. It is not hard to see that $\mathbf{v}_{\mathbf{w}}^*$ can be used to rotate $\mathbf{w}$ towards $\mathbf{w}^*$. The vector that we identify is a top eigenvector of a matrix $\mathbf{M}_{\mathbf{w}}$ that can be efficiently estimated using sample access to labeled data. We can only identify such a target vector up to its sign; however, as we argue later, this is sufficient for our argument to go through.

To build up this result, we need the following technical pieces: (1) the spectrum of the matrix $\mathbf{M}_{\mathbf{w}}$ contains information on $\mathbf{v}_{\mathbf{w}}^*$, i.e., $\mathbf{v}_{\mathbf{w}}^* \cdot \mathbf{M}_{\mathbf{w}} \mathbf{v}_{\mathbf{w}}^*$ is large (Lemma C.3); (2) All the other directions $\mathbf{u}$ that

are orthogonal to $\mathbf{v}_\mathbf{w}^*$ have small quadratic form value compared to $\mathbf{v}_\mathbf{w}^*$, i.e., $\mathbf{u} \cdot \mathbf{M}_\mathbf{w} \mathbf{u} \ll \mathbf{v}_\mathbf{w}^* \cdot \mathbf{M}_\mathbf{w} \mathbf{v}_\mathbf{w}^*$, and therefore, the direction $\mathbf{v}_\mathbf{w}^*$ stands out in the spectrum of the matrix $\mathbf{M}_\mathbf{w}$ (Lemma C.5); (3) finally, we argue that the top eigenvector $\mathbf{v}_\mathbf{w}$ of $\mathbf{M}_\mathbf{w}$ correlates strongly with $\mathbf{v}_\mathbf{w}^*$ (Lemma C.6).

---

**Algorithm 4** Spectral Optimization

---

1: **Input:** Parameter $\theta_0$; Initialization vector $\mathbf{w}^{(0)}$; Empirical Distribution $\widehat{\mathcal{D}}_N$;
2: $S^{\mathrm{sol}} \leftarrow \emptyset$, $\phi_t \leftarrow \bar{\theta}(1 - 1/128)^t$, $\eta_t \leftarrow \sin \phi_t / 8$, $K \leftarrow \mathrm{poly}(1/\epsilon, L)$ and $T \leftarrow \Theta(\log(1/(\epsilon L)))$.
3: **for** $k = 1, \ldots, K$ **do**
4:      **for** $t = 0, \ldots, T$ **do**
5:          Let $\widehat{\mathbf{g}}_{\mathbf{w}^{(t)}}^{(j)} \leftarrow \mathbf{E}_{(\mathbf{x},y) \sim \widehat{\mathcal{D}}_N}[y \mathbf{x}^{\perp \mathbf{w}^{(t)}} \mathbb{1}\{\mathbf{w}^{(t)} \cdot \mathbf{x} \in \mathcal{E}_j\}]$, $j \in [I]$.
6:          Compute the empirical matrix $\widehat{\mathbf{M}}_{\mathbf{w}^{(t)}} \leftarrow \sum_{j=1}^I \widehat{\mathbf{g}}_{\mathbf{w}^{(t)}}^{(j)} (\widehat{\mathbf{g}}_{\mathbf{w}^{(t)}}^{(j)})^\top / \mathbf{Pr}_{z \sim \mathcal{N}(0,1)}[z \in \mathcal{E}_j]$.
7:          Find the top eigenvector $\mathbf{u}$ of $\widehat{\mathbf{M}}_{\mathbf{w}^{(t)}}$, then randomly pick $\mathbf{v}^{(t)}$ from $\{\pm \mathbf{u}\}$.
8:          $\mathbf{w}^{(k+1)} \leftarrow \mathrm{proj}_{\mathbb{B}^d}(\mathbf{w}^{(t)} - \eta_t \mathbf{v}^{(t)})$.
9:          $S^{\mathrm{sol}} \leftarrow S^{\mathrm{sol}} \cup \{\mathbf{w}^{(k+1)}\}$.
10: **Return:** $S^{\mathrm{sol}}$.

---

**Proposition C.1** (Spectral Alignment). *Fix parameters $B, L > 0$ and $\epsilon > 0$. Let $\mathcal{D}$ be a distribution over $(\mathbf{x}, y) \in \mathbb{R}^d \times \mathbb{R}$ with $\mathcal{D}_\mathbf{x} = \mathcal{N}(\mathbf{0}, \mathbf{I}_d)$. Let $(\mathbf{w}^*, \sigma) \in \mathbb{S}^{d-1} \times \mathcal{H}_\epsilon(B, L)$ be a pair of unit vector and monotone activation such that $\mathcal{L}_2(\mathbf{w}^*; \sigma) = \mathrm{OPT}$. If $N \geq d^2 \mathrm{poly}(B, L, 1/\epsilon)$, then with probability at least $99\%$ the following holds: for any unit vector $\mathbf{w} \in \mathbb{R}^d$ satisfying $\sin \theta(\mathbf{w}, \mathbf{w}^*) \geq 40\sqrt{\mathrm{OPT}}/\|\mathrm{T}_{\cos \theta(\mathbf{w}, \mathbf{w}^*)} \sigma'\|_{L_2}$, the top (unit) eigenvector $\mathbf{u} \in \mathbb{R}^d$ of the empirical matrix $\widehat{\mathbf{M}}_\mathbf{w}$ returned by Algorithm 4 satisfies $\mathbf{u} \cdot \mathbf{w} = 0$ and $|\mathbf{u} \cdot \mathbf{w}^*| \geq (\sqrt{2}/2) \sin \theta(\mathbf{w}, \mathbf{w}^*)$.*

The rest of this subsection is as follows: Appendix C.1 develops the machinery required to prove Proposition C.1. We prove Proposition C.1 in Appendix C.2. In Appendix C.3, we determine the sample complexity for the spectral subroutine.

## C.1    Technical Machinery for Proposition C.1

We start with the following fact from [ZWDD25] showing that given the target activation $\sigma$, the gradient vector of the smoothed $L_2^2$ loss correlates strongly with $\mathbf{w}^*$:

**Fact C.2** (Proposition 2.2, [ZWDD25]). *Fix an activation $\sigma : \mathbb{R} \to \mathbb{R}$. Fix vectors $\mathbf{w}^*, \mathbf{w} \in \mathbb{S}^{d-1}$ such that $\mathbf{E}_{(\mathbf{x},y) \sim \mathcal{D}}[(y - \sigma(\mathbf{w}^* \cdot \mathbf{x}))^2] = \mathrm{OPT}$ and let $\theta = \theta(\mathbf{w}^*, \mathbf{w})$. If $\sin \theta \geq 3\sqrt{\mathrm{OPT}}/\|\mathrm{T}_{\cos \theta} \sigma'\|_{L_2}$, then:*

$$\mathbf{E}_{(\mathbf{x},y) \sim \mathcal{D}}[y \mathrm{T}_{\cos \theta} \sigma'(\mathbf{w} \cdot \mathbf{x}) \mathbf{x}^{\perp \mathbf{w}}] \cdot \mathbf{w}^* \geq (2/3)\|\mathrm{T}_{\cos \theta} \sigma'\|_{L_2}^2 \sin^2 \theta.$$

The structural result above relies heavily on the knowledge of the target activation $\sigma$ and is thus not applicable to our setting. Instead, we argue (Lemma C.6) that using the vector $\mathbf{g}_\mathbf{w}(z)$ defined below, the top eigenvector of the matrix $\mathbf{M}_\mathbf{w} \coloneqq \mathbf{E}_{\mathbf{z} \sim \mathcal{N}(\mathbf{0}, \mathbf{I}_d)}[\mathbf{g}_\mathbf{w}(\mathbf{w} \cdot \mathbf{z}) \mathbf{g}_\mathbf{w}(\mathbf{w} \cdot \mathbf{z})^\top]$, correlates with the "ideal" update direction $\mathbf{v}_\mathbf{w}^* \coloneqq (\mathbf{w}^*)^{\perp \mathbf{w}}/\|(\mathbf{w}^*)^{\perp \mathbf{w}}\|_2$:

$$\mathbf{g}_\mathbf{w}(z) \coloneqq \sum_{i=1}^I \frac{\mathbf{E}_{(\mathbf{x},y) \sim \mathcal{D}}[y \mathbf{x}^{\perp \mathbf{w}} \mathbb{1}\{\mathbf{w} \cdot \mathbf{x} \in \mathcal{E}_i\}]}{\mathbf{Pr}[\mathbf{w} \cdot \mathbf{x} \in \mathcal{E}_i]} \mathbb{1}\{z \in \mathcal{E}_i\},$$

$$\text{where} \begin{cases} \mathcal{E}_i = [a_i, a_{i+1}) = [-M' + (i-1)\Delta, -M' + i\Delta), \ \Delta = \epsilon^2/(B^2 L^2); \\ M' = O(\sqrt{\log(BL/\epsilon)}) \text{ satisfies } \mathbf{Pr}_{z \sim \mathcal{N}}[|z| \geq M'] \leq \epsilon^2/(B^2 L^2); \\ I = O(M' B^2 L^2/\epsilon^2) = \tilde{O}(B^2 L^2/\epsilon^2). \end{cases} \quad \text{(Grad)}$$

Denote $\mathbf{E}_{(\mathbf{x},y) \sim \mathcal{D}}[y \mathbf{x}^{\perp \mathbf{w}} \mid \mathbf{w} \cdot \mathbf{x} \in \mathcal{E}_i] \coloneqq \mathbf{E}_{(\mathbf{x},y) \sim \mathcal{D}}[y \mathbf{x}^{\perp \mathbf{w}} \mathbb{1}\{\mathbf{w} \cdot \mathbf{x} \in \mathcal{E}_i\}]/\mathbf{Pr}[\mathbf{w} \cdot \mathbf{x} \in \mathcal{E}_i]$, so that $\mathbf{g}_\mathbf{w}(z)$ can be written as $\mathbf{g}_\mathbf{w}(z) = \sum_i \mathbf{E}_{(\mathbf{x},y) \sim \mathcal{D}}[y \mathbf{x}^{\perp \mathbf{w}} \mid \mathbf{w} \cdot \mathbf{x} \in \mathcal{E}_i] \mathbb{1}\{z \in \mathcal{E}_i\}$. We note that for any $z$ in interval $\mathcal{E}_i$, $\mathbf{g}_\mathbf{w}(z) = \mathbf{E}_{(\mathbf{x},y) \sim \mathcal{D}}[y \mathbf{x}^{\perp \mathbf{w}} \mid \mathbf{w} \cdot \mathbf{x} \in \mathcal{E}_i]$, i.e., it is a fixed vector that only depends on $\mathbf{w}$.

First, we show that the quadratic form $(\mathbf{v}_\mathbf{w}^*)^\top \mathbf{M}_\mathbf{w} \mathbf{v}_\mathbf{w}^*$ with respect to the target vector $\mathbf{v}_\mathbf{w}^*$ is large.

**Lemma C.3.** *Let* $\mathbf{g_w}(z)$ *be the vector defined in* (Grad), *let* $\mathbf{M_w} := \mathbf{E}_{z \sim \mathcal{N}}[\mathbf{g_w}(z)\mathbf{g_w}(z)^\top]$, *and* $\mathbf{v_w^*} := (\mathbf{w^*})^{\perp \mathbf{w}}/\|(\mathbf{w^*})^{\perp \mathbf{w}}\|_2$. *Suppose* $\epsilon \leq \mathrm{OPT}$ *and* $\sin\theta \geq 4\sqrt{\mathrm{OPT}}/\|\mathrm{T}_{\cos\theta}\sigma'(z)\|_{L_2}$. *Then,* $(\mathbf{v_w^*})^\top \mathbf{M_w}\mathbf{v_w^*} \geq (1/16)\sin^2\theta\|\mathrm{T}_{\cos\theta}\sigma'\|_{L_2}^2$.

*Proof.* Observe that $\mathbf{v_w^*}\sin\theta = (\mathbf{w^*})^{\perp\mathbf{w}}$. Consider the following quantity:

$$K(h(\mathbf{w}\cdot\mathbf{x})) := \mathop{\mathbf{E}}_{(\mathbf{x},y)\sim\mathcal{D}}[y\mathbf{w}^* \cdot \mathbf{x}^{\perp\mathbf{w}}h(\mathbf{w}\cdot\mathbf{x})].$$

Define the function class $\mathcal{H}'$ by

$$\mathcal{H}' = \left\{ h : \mathbb{R} \to \mathbb{R} \mid h(z) = \sum_{i=1}^{I} h_i \mathbb{1}\{z \in \mathcal{E}_i\}, \|h\|_{L_2} = 1 \right\},$$

where $\mathcal{E}_i, i = 1, \ldots, I$, are the same intervals as in the definition of $\mathbf{g_w}(z)$, (Grad). Our goal is to find $h \in \mathcal{H}'$ that maximizes $K(h(\mathbf{w}\cdot\mathbf{x}))$. Observe that for $h \in \mathcal{H}'$, $K(h(\mathbf{w}\cdot\mathbf{x}))$ can be written as

$$K(h(\mathbf{w}\cdot\mathbf{x})) = \mathop{\mathbf{E}}_{(\mathbf{x},y)\sim\mathcal{D}}\left[ y\mathbf{w}^* \cdot \mathbf{x}^{\perp\mathbf{w}} \sum_{i=1}^{I} h_i \mathbb{1}\{\mathbf{w}\cdot\mathbf{x} \in \mathcal{E}_i\} \right]$$

$$\overset{(i)}{=} \sum_{i=1}^{I} h_i \sin\theta \mathop{\mathbf{E}}_{(\mathbf{x},y)\sim\mathcal{D}}[y(\mathbf{v_w^*}\cdot\mathbf{x})\mathbb{1}\{\mathbf{w}\cdot\mathbf{x} \in \mathcal{E}_i\}]$$

$$\overset{(ii)}{=} \sum_{i=1}^{I} h_i \sin\theta \left( \mathbf{v_w^*} \cdot \mathop{\mathbf{E}}_{(\mathbf{x},y)\sim\mathcal{D}}[y\mathbf{x}^{\perp\mathbf{w}} \mid \mathbf{w}\cdot\mathbf{x} \in \mathcal{E}_i] \right) \mathbf{Pr}[\mathbf{w}\cdot\mathbf{x} \in \mathcal{E}_i]$$

$$= \sum_{i=1}^{I} h_i \sin\theta \left( \mathbf{v_w^*} \cdot \mathop{\mathbf{E}}_{(\mathbf{x},y)\sim\mathcal{D}}[y\mathbf{x}^{\perp\mathbf{w}} \mid \mathbf{w}\cdot\mathbf{x} \in \mathcal{E}_i] \right) \mathbf{Pr}[z \in \mathcal{E}_i] \qquad (3)$$

$$\overset{(iii)}{=} \sin\theta \mathop{\mathbf{E}}_{z\sim\mathcal{N}}\left[ \left( \sum_{i=1}^{I}\left(h_i\mathbb{1}\{z\in\mathcal{E}_i\}\right) \right)\left( \mathbf{v_w^*} \cdot \mathop{\mathbf{E}}_{(\mathbf{x},y)\sim\mathcal{D}}[y\mathbf{x}^{\perp\mathbf{w}} \mid \mathbf{w}\cdot\mathbf{x} \in \mathcal{E}_i]\mathbb{1}\{z\in\mathcal{E}_i\} \right) \right]$$

$$= \sin\theta \mathop{\mathbf{E}}_{z\sim\mathcal{N}}\left[ h(z)(\mathbf{v_w^*}\cdot\mathbf{g_w}(z)) \right] = \sin\theta \Big\langle h(z), \mathbf{v_w^*}\cdot\mathbf{g_w}(z) \Big\rangle_{L_2(\mathcal{N}(0,1))},$$

where $(i)$ is by $\mathbf{w}^* \cdot \mathbf{x}^{\perp\mathbf{w}} = (\mathbf{w}^*)^{\perp\mathbf{w}} \cdot \mathbf{x}^{\perp\mathbf{w}} = \mathbf{v_w^*} \cdot \mathbf{x}^{\perp\mathbf{w}}\sin\theta$, $(ii)$ is by the definition of $\mathbf{E}_{(\mathbf{x},y)\sim\mathcal{D}}[y\mathbf{x}^{\perp\mathbf{w}} \mid \mathbf{w}\cdot\mathbf{x} \in \mathcal{E}_i]$, $(iii)$ is by $\mathbf{Pr}[z \in \mathcal{E}_i] = \mathbf{E}_{\mathbf{x}\sim\mathcal{D}_\mathbf{x}}[\mathbb{1}\{z \in \mathcal{E}_i\}] = \mathbf{E}_{z\sim\mathcal{N}}[(\mathbb{1}\{z \in \mathcal{E}_i\})^2]$ and an appropriate grouping of terms, and the remaining inequalities use the definition of the inner product and that $\mathbf{w}\cdot\mathbf{z} \sim \mathcal{N}(0,1)$. Because $K(h(\mathbf{w}\cdot\mathbf{x}))$ is an inner product in the space of $L_2(\mathcal{N})$ functions, from the $L_2$ norm (self-)duality, this is maximized for $h(z) = (\mathbf{g_w}(z)\cdot\mathbf{v_w^*})/\|\mathbf{g_w}(z)\cdot\mathbf{v_w^*}\|_{L_2}$.

Now we study $(\mathbf{v_w^*})^\top\mathbf{M_w}\mathbf{v_w^*}$. Using the result that $K(h(\mathbf{w}\cdot\mathbf{x}))$ is maximized at $h^*(z) = \mathbf{v_w^*}\cdot\mathbf{g_w}(z)/\|\mathbf{v_w^*}\cdot\mathbf{g_w}(z)\|_{L_2}$ from the discussion above, and recalling that $\mathbf{Pr}_{\mathbf{x}\sim\mathcal{D}_\mathbf{x}}[\mathbf{w}\cdot\mathbf{x}\in\mathcal{E}_j] = \mathbf{Pr}_{z\sim\mathcal{N}}[z\in\mathcal{E}_j]$ (since $\mathbf{x}$ follows the standard Gaussian distribution), we have that

$$(\mathbf{v_w^*})^\top\mathbf{M_w}\mathbf{v_w^*} = \mathop{\mathbf{E}}_{z\sim\mathcal{N}}\left[ (\mathbf{g_w}(z)\cdot\mathbf{v_w^*})^2 \right] \qquad (4)$$

$$= \mathop{\mathbf{E}}_{z\sim\mathcal{N}}\left[ \left( \sum_{i=1}^{I} \mathop{\mathbf{E}}_{(\mathbf{x},y)\sim\mathcal{D}}[y(\mathbf{v_w^*}\cdot\mathbf{x}) \mid \mathbf{w}\cdot\mathbf{x}\in\mathcal{E}_i]\mathbb{1}\{z\in\mathcal{E}_i\} \right)^2 \right]$$

$$= \sum_{i=1}^{I}\left( \mathop{\mathbf{E}}_{(\mathbf{x},y)\sim\mathcal{D}}[y(\mathbf{v_w^*}\cdot\mathbf{x}) \mid \mathbf{w}\cdot\mathbf{x}\in\mathcal{E}_i] \right)^2 \mathop{\mathbf{E}}_{z\sim\mathcal{N}}[\mathbb{1}\{z\in\mathcal{E}_i\}]$$

$$= \sum_{i=1}^{I}\left( \mathbf{v_w^*}\cdot\mathop{\mathbf{E}}_{(\mathbf{x},y)\sim\mathcal{D}}[y\mathbf{x}^{\perp\mathbf{w}} \mid \mathbf{w}\cdot\mathbf{x}\in\mathcal{E}_i] \right)^2 \mathbf{Pr}[z\in\mathcal{E}_i]$$

$$= \frac{\|\mathbf{v_w^*}\cdot\mathbf{g_w}(z)\|_{L_2}}{\sin\theta}K(h^*(\mathbf{w}\cdot\mathbf{x})), \qquad (5)$$

where in the last equality we used (3) and that $h_i^* = \mathbf{v_w^*} \cdot \mathbf{E}_{(\mathbf{x},y)\sim\mathcal{D}}[y\mathbf{x}^{\perp\mathbf{w}} \mid \mathbf{w}\cdot\mathbf{x} \in \mathcal{E}_i]/\|\mathbf{v_w^*} \cdot \mathbf{g_w}(z)\|_{L_2}$ by the definition of $h^*(z)$. Let us define

$$\widetilde{\mathrm{T}}_{\cos\theta}\sigma'(z) = \sum_{i=1}^{I} \mathrm{T}_{\cos\theta}\sigma'(a_i)\mathbb{1}\{z \in \mathcal{E}_i\}. \tag{6}$$

Recall that we have defined $a_i$ as the left endpoint of the interval $\mathcal{E}_i = [a_i, a_{i+1})$ in (Grad). We show that $\widetilde{\mathrm{T}}_{\cos\theta}\sigma'(\mathbf{w}\cdot\mathbf{x})$ is close to $\mathrm{T}_{\cos\theta}\sigma'(\mathbf{w}\cdot\mathbf{x})$ pointwise.

**Claim C.4.** *Suppose* $\sin\theta\|\mathrm{T}_{\cos\theta}\sigma'\|_{L_2} \gtrsim \sqrt{\mathrm{OPT}}$ *and* $\epsilon \leq \mathrm{OPT}$. *Consider the piecewise constant function* $\widetilde{\mathrm{T}}_{\cos\theta}\sigma'(z)$ *defined in Equation (6), with the intervals* $\mathcal{E}_i$, $i \in [I]$, *and parameter* $M'$ *following the construction in* (Grad). *Then,*

$$|\widetilde{\mathrm{T}}_{\cos\theta}\sigma'(z) - \mathrm{T}_{\cos\theta}\sigma'(z)|\mathbb{1}\{z \in [-M', M']\} \leq \epsilon/B\,;$$
$$|\|\mathrm{T}_{\cos\theta}\sigma'(z)\|_{L_2} - \|\widetilde{\mathrm{T}}_{\cos\theta}\sigma'(z)\|_{L_2}| \leq \epsilon\,.$$

*Proof of Claim C.4.* By Fact A.2 part (c), we know that for any $\rho \in (0,1)$, $\|(\mathrm{T}_\rho f(z))'\|_{L_\infty} \leq \|f\|_{L_\infty}/\sqrt{1-\rho^2}$. In addition, by Fact A.2 part (a) we have $\mathrm{T}_{\cos\theta}\sigma'(z) = \mathrm{T}_{\sqrt{\cos\theta}}(\mathrm{T}_{\sqrt{\cos\theta}}\sigma'(z))$. Therefore we claim that $\mathrm{T}_{\cos\theta}\sigma'(z)$ is a Lipschitz function:

$$\|(\mathrm{T}_{\cos\theta}\sigma'(z))'\|_{L_\infty} = \|(\mathrm{T}_{\sqrt{\cos\theta}}(\mathrm{T}_{\sqrt{\cos\theta}}\sigma'(z)))'\|_{L_\infty} \leq \frac{\|\mathrm{T}_{\sqrt{\cos\theta}}\sigma'(z)\|_{L_\infty}}{\sqrt{1-\cos\theta}} \overset{(i)}{=} \frac{\|(\mathrm{T}_{\sqrt{\cos\theta}}\sigma(z))'\|_{L_\infty}}{\sqrt{2\cos\theta}\sin(\theta/2)}$$

$$\leq \frac{\|\sigma(z)\|_{L_\infty}}{\sqrt{2\cos\theta}\sin(\theta/2)\sqrt{1-\cos\theta}} \overset{(ii)}{\leq} \frac{B}{2\sin^2(\theta/2)\sqrt{\cos\theta}},$$

where in $(i)$ we applied Fact A.2 part (g) and the fact that $1 - \cos\theta = 2\sin^2(\theta/2)$ and in $(ii)$ we used the fact that $\|\sigma\|_{L_\infty} \leq B$ since $\sigma$ is $\epsilon$-Extended $(B, L)$-regular. Furthermore, note that by our assumption that $\sin\theta \gtrsim \sqrt{\mathrm{OPT}}/\|\mathrm{T}_{\cos\theta}\sigma'\|_2$, we have $\sin^2\theta \geq \mathrm{OPT}/\|\mathrm{T}_{\cos\theta}\sigma'\|_{L_2}^2$. Using the fact that $\|\mathrm{T}_\rho f\|_{L_2}^2$ is an non-decreasing function with respect to $\rho$ (Fact A.2, (f)), we have $\|\mathrm{T}_{\cos\theta}\sigma'\|_{L_2}^2 \leq \|\sigma'\|_{L_2}^2 \leq L^2$, again by the assumption that $\sigma$ is $\epsilon$-Extended $(B, L)$-Regular. Hence $\sin^2\theta \geq \mathrm{OPT}/L^2$. Finally, our initialization subroutine guarantees that $\cos\theta \geq c$ for some small absolute constant $c > 0$. Therefore, in summary, we obtain that $\|(\mathrm{T}_{\cos\theta}\sigma'(z))'\|_{L_\infty} \lesssim BL^2/\mathrm{OPT}$.

Now for any $i \in [I]$, let $z \in [a_i, a_{i+1})$ be any value in the interval $\mathcal{E}_i$. Since we have proved in the last paragraph that $\mathrm{T}_{\cos\theta}\sigma'(z)$ is $O(BL^2/\mathrm{OPT})$-Lipschitz, we have that there exists a sufficiently large absolute constant $C'$ such that

$$|\mathrm{T}_{\cos\theta}\sigma'(z) - \mathrm{T}_{\cos\theta}\sigma'(a_i)| \leq C'BL^2/\mathrm{OPT}|z - a_i| \leq (C'BL^2/\mathrm{OPT})(\epsilon^2/(CB^2L^2)) \leq \epsilon/B.$$

Note that in the last inequality we used (by the definition of $\mathcal{E}_i = [a_i, a_{i+1})$ in (Grad)) that $a_{i+1} - a_i = \Delta \leq \epsilon^2/(CB^2L^2)$, where $C \geq C'$ is a sufficiently large absolute constant. Therefore, we conclude that $|\widetilde{\mathrm{T}}_{\cos\theta}\sigma'(z) - \mathrm{T}_{\cos\theta}\sigma'(z)|\mathbb{1}\{z \in [-M', M']\} \leq \epsilon/B$, proving the first part of the claim.

For the second part of the claim, note first that

$$\|\mathrm{T}_{\cos\theta}\sigma'(z) - \widetilde{\mathrm{T}}_{\cos\theta}\sigma'(z)\|_{L_2} \leq \|(\mathrm{T}_{\cos\theta}\sigma'(z) - \widetilde{\mathrm{T}}_{\cos\theta}\sigma'(z))\mathbb{1}\{z \in [-M', M']\}\|_{L_2}$$
$$+ \|\mathrm{T}_{\cos\theta}\sigma'(z)\mathbb{1}\{|z| \geq M'\}\|_{L_2}\,.$$

Using the first part of the claim, we obtain $\|(\mathrm{T}_{\cos\theta}\sigma'(z) - \widetilde{\mathrm{T}}_{\cos\theta}\sigma'(z))\mathbb{1}\{z \in [-M', M']\}\|_{L_2} \leq \epsilon$. Now applying Fact A.2(c) again we have $\|\mathrm{T}_{\cos\theta}\sigma'\|_{L_\infty} \leq B/(\cos\theta\sin\theta)$. We have argued in the previous paragraph that $\sin\theta \geq \sqrt{\mathrm{OPT}}/L \geq \sqrt{\epsilon}/L$ and $\cos\theta \geq c$ for some small absolute constant $c$ under our assumptions. Therefore, $\|\mathrm{T}_{\cos\theta}\sigma'\|_{L_\infty} \lesssim BL/\sqrt{\epsilon}$. Since $M' = O(\sqrt{\log(BL/\epsilon)})$ is chosen such that $\mathbf{Pr}[|z| \geq M'] \leq \epsilon^2/(CBL)^2$ for a large absolute constant $C$, where $z$ is a standard Gaussian random variable, we have

$$\mathop{\mathbf{E}}_{z\sim\mathcal{N}}[(\mathrm{T}_{\cos\theta}\sigma'(z))^2\mathbb{1}\{|z| \geq M\}] \leq (B^2L^2/\epsilon)\,\mathbf{Pr}[|z| \geq M'] \leq \epsilon.$$

Therefore, it holds $|\|\mathrm{T}_{\cos\theta}\sigma'(z)\|_{L_2} - \|\widetilde{\mathrm{T}}_{\cos\theta}\sigma'(z)\|_{L_2}| \leq 2\epsilon$. $\qquad\square$

Now observe that since $\widetilde{T}_{\cos\theta}\sigma'(z)/\|\widetilde{T}_{\cos\theta}\sigma'\|_{L_2} \in \mathcal{H}'$, we have that

$$K(h^*(\mathbf{w}\cdot\mathbf{x})) \geq K(\widetilde{T}_{\cos\theta}\sigma'(\mathbf{w}\cdot\mathbf{x})/\|\widetilde{T}_{\cos\theta}\sigma'\|_{L_2}) \,,$$

which implies (from Equation (4))

$$\mathbf{v}_{\mathbf{w}}^* \cdot \mathbf{M}_{\mathbf{w}}\mathbf{v}_{\mathbf{w}}^* \geq \frac{\|\mathbf{v}_{\mathbf{w}}^* \cdot \mathbf{g}_{\mathbf{w}}(z)\|_{L_2}}{\sin\theta} K(\widetilde{T}_{\cos\theta}\sigma'(\mathbf{w}\cdot\mathbf{x})/\|\widetilde{T}_{\cos\theta}\sigma'\|_{L_2})$$

$$= \frac{\|\mathbf{v}_{\mathbf{w}}^* \cdot \mathbf{g}_{\mathbf{w}}(z)\|_{L_2}}{\|\widetilde{T}_{\cos\theta}\sigma'\|_{L_2}\sin\theta} \sum_{i=1}^{I} \mathop{\mathbf{E}}_{(\mathbf{x},y)\sim\mathcal{D}}[y(\mathbf{w}^*\cdot\mathbf{x}^{\perp\mathbf{w}})T_{\cos\theta}\sigma'(a_i)\mathbb{1}\{\mathbf{w}\cdot\mathbf{x}\in\mathcal{E}_i\}] \,. \quad (7)$$

Note that by the definition of $\mathbf{M}_{\mathbf{w}}$, we have $(\mathbf{v}_{\mathbf{w}}^*)^\top \mathbf{M}_{\mathbf{w}}\mathbf{v}_{\mathbf{w}}^* = \mathbf{E}_{z\sim\mathcal{N}}[(\mathbf{v}_{\mathbf{w}}^*\cdot\mathbf{g}_{\mathbf{w}}(z))^2] = \|\mathbf{g}_{\mathbf{w}}(z)\cdot \mathbf{v}_{\mathbf{w}}^*\|_{L_2}^2$. Furthermore, as we have shown in Claim C.4, it holds $\|\widetilde{T}_{\cos\theta}\sigma'\|_{L_2} \leq \|T_{\cos\theta}\sigma'\|_{L_2} + \epsilon$; and note that we have $3\sqrt{\epsilon} \leq 3\sqrt{\mathrm{OPT}} \leq \|T_{\cos\theta}\sigma'\|_{L_2}$ (since we have assumed $1 \geq \sin\theta \geq 3\sqrt{\mathrm{OPT}}/\|T_{\cos\theta}\sigma'\|_{L_2}$). Thus, for small $\epsilon$ it holds $\sqrt{\epsilon} \leq \|T_{\cos\theta}\sigma'\|_{L_2}$ and we obtain $\|\widetilde{T}_{\cos\theta}\sigma'\|_{L_2} \leq 2\|T_{\cos\theta}\sigma'\|_{L_2}$. Using that $\|\mathbf{g}(z)\cdot\mathbf{v}_{\mathbf{w}}^*\|_{L_2} = (\mathbf{v}_{\mathbf{w}}^*\cdot\mathbf{M}_{\mathbf{w}}\mathbf{v}_{\mathbf{w}}^*)^{1/2}$ and $\|\widetilde{T}_{\cos\theta}\sigma'\|_{L_2} \leq 2\|T_{\cos\theta}\sigma'\|_{L_2}$ into Equation (7) yields

$$(\mathbf{v}_{\mathbf{w}}^* \cdot \mathbf{M}\mathbf{v}_{\mathbf{w}}^*)^{1/2}$$

$$\geq \frac{1}{2\sin\theta\|T_{\cos\theta}\sigma'(z)\|_{L_2}} \sum_{i=1}^{I} \mathop{\mathbf{E}}_{(\mathbf{x},y)\sim\mathcal{D}}[y(\mathbf{w}^*\cdot\mathbf{x}^{\perp\mathbf{w}})T_{\cos\theta}\sigma'(a_i)\mathbb{1}\{\mathbf{w}\cdot\mathbf{x}\in\mathcal{E}_i\}]$$

$$= \frac{1}{2\sin\theta\|T_{\cos\theta}\sigma'(z)\|_{L_2}} \sum_{i=1}^{I} \underbrace{\mathop{\mathbf{E}}_{(\mathbf{x},y)\sim\mathcal{D}}[y(\mathbf{w}^*\cdot\mathbf{x}^{\perp\mathbf{w}})(T_{\cos\theta}\sigma'(\mathbf{w}\cdot\mathbf{x}))\mathbb{1}\{\mathbf{w}\cdot\mathbf{x}\in\mathcal{E}_i\}]}_{(Q_1)}$$

$$+ \frac{1}{2\|T_{\cos\theta}\sigma'(z)\|_{L_2}} \sum_{i=1}^{I} \underbrace{\mathop{\mathbf{E}}_{(\mathbf{x},y)\sim\mathcal{D}}[y(\mathbf{v}_{\mathbf{w}}^*\cdot\mathbf{x})(T_{\cos\theta}\sigma'(a_i) - T_{\cos\theta}\sigma'(\mathbf{w}\cdot\mathbf{x}))\mathbb{1}\{\mathbf{w}\cdot\mathbf{x}\in\mathcal{E}_i\}]}_{(Q_2)} \,.$$

$$(8)$$

For the term $(Q_1)$, we apply Fact C.2 and obtain

$$(Q_1) = \mathop{\mathbf{E}}_{(\mathbf{x},y)\sim\mathcal{D}}[y(\mathbf{w}^*\cdot\mathbf{x}^{\perp\mathbf{w}})T_{\cos\theta}\sigma'(\mathbf{w}\cdot\mathbf{x})\mathbb{1}\{|\mathbf{w}\cdot\mathbf{x}| \leq M'\}]$$

$$\geq (2/3)\sin^2\theta\|T_{\cos\theta}\sigma'\|_{L_2}^2 - \mathop{\mathbf{E}}_{(\mathbf{x},y)\sim\mathcal{D}}[y(\mathbf{w}^*\cdot\mathbf{x}^{\perp\mathbf{w}})T_{\cos\theta}\sigma'(\mathbf{w}\cdot\mathbf{x})\mathbb{1}\{|\mathbf{w}\cdot\mathbf{x}| \geq M'\}]$$

$$\geq (2/3)\sin^2\theta\|T_{\cos\theta}\sigma'\|_{L_2}^2 - B\mathop{\mathbf{E}}_{\mathbf{x}\sim\mathcal{D}_{\mathbf{x}}}[|\mathbf{v}_{\mathbf{w}}^*\cdot\mathbf{x}|]\mathop{\mathbf{E}}_{\mathbf{x}\sim\mathcal{D}_{\mathbf{x}}}[|\sin\theta T_{\cos\theta}\sigma'(\mathbf{w}\cdot\mathbf{x})|\mathbb{1}\{|\mathbf{w}\cdot\mathbf{x}| \geq M'\}],$$

where in the second inequality we used $|y| \leq B$ and $\mathbf{v}_{\mathbf{w}}^*\sin\theta = (\mathbf{w}^*)^{\perp\mathbf{w}}$. Using the Cauchy-Schwarz inequality further yields

$$\mathop{\mathbf{E}}_{\mathbf{x}\sim\mathcal{D}_{\mathbf{x}}}[|\sin\theta T_{\cos\theta}\sigma'(\mathbf{w}\cdot\mathbf{x})|\mathbb{1}\{|\mathbf{w}\cdot\mathbf{x}| \geq M'\}] \leq \sin\theta\sqrt{\mathop{\mathbf{E}}_{\mathbf{x}\sim\mathcal{D}_{\mathbf{x}}}[(T_{\cos\theta}\sigma'(z))^2]\mathbf{Pr}[|\mathbf{w}\cdot\mathbf{x}| \geq M']}$$

$$\leq \sin\theta\|T_{\cos\theta}\sigma'\|_{L_2}(\epsilon/(CBL)),$$

where we used the fact that $M'$ satisfies $\mathbf{Pr}[|z| \geq M'] \leq \epsilon^2/(CBL)^2$ for some large absolute constant $C$; see the definition of $M'$ in (Grad). When $\sin\theta\|T_{\cos\theta}\sigma'\|_{L_2} \geq 3\sqrt{\mathrm{OPT}} \geq 3\sqrt{\epsilon}$, since $C$ is a large absolute constant and $L$ is a constant larger than 1, we have $\epsilon/(CBL) \leq (1/24B)\sin\theta\|T_{\cos\theta}\sigma'\|_{L_2}$. Therefore, it holds

$$(Q_1) \geq (2/3)\sin^2\theta\|T_{\cos\theta}\sigma'\|_{L_2}^2 - B\mathop{\mathbf{E}}_{\mathbf{x}\sim\mathcal{D}_{\mathbf{x}}}[|\mathbf{v}_{\mathbf{w}}^*\cdot\mathbf{x}|](1/24B)\sin^2\theta\|T_{\cos\theta}\sigma'\|_{L_2}^2$$

$$\geq (7/12)\sin^2\theta\|T_{\cos\theta}\sigma'\|_{L_2}^2 \,.$$

For the term $(Q_2)$, using Claim C.4 again we have

$$|(Q_2)| \leq \sum_{i=1}^{I} \mathop{\mathbf{E}}_{(\mathbf{x},y)\sim\mathcal{D}}[|y||\mathbf{v}_\mathbf{w}^* \cdot \mathbf{x}||\mathrm{T}_{\cos\theta}\sigma'(a_i) - \mathrm{T}_{\cos\theta}\sigma'(\mathbf{w}\cdot\mathbf{x})|\mathbb{1}\{\mathbf{w}\cdot\mathbf{x}\in\mathcal{E}_i\}]$$

$$\leq \sum_{i=1}^{I} B \mathop{\mathbf{E}}_{\mathbf{x}\sim\mathcal{D}_\mathbf{x}}[|\mathbf{v}_\mathbf{w}^* \cdot \mathbf{x}|] \mathop{\mathbf{E}}_{\mathbf{x}\sim\mathcal{D}_\mathbf{x}}[|\mathrm{T}_{\cos\theta}\sigma'(a_i) - \mathrm{T}_{\cos\theta}\sigma'(\mathbf{w}\cdot\mathbf{x})|\mathbb{1}\{\mathbf{w}\cdot\mathbf{x}\in\mathcal{E}_i\}]$$

$$\leq \sum_{i=1}^{M} B(\epsilon/B)\mathbf{Pr}[z\in\mathcal{E}_i] \leq \epsilon.$$

Since $\epsilon \leq \mathrm{OPT} \leq (1/12)\sin^2\theta\|\mathrm{T}_{\cos\theta}\sigma'\|_{L_2}^2$, we obtain that $|(Q_2)| \leq (1/12)\sin^2\theta\|\mathrm{T}_{\cos\theta}\sigma'\|_{L_2}^2$.

Plugging $(Q_1),(Q_2)$ back into Equation (8) yields:

$$(\mathbf{v}_\mathbf{w}^* \cdot \mathbf{M}_\mathbf{w}\mathbf{v}_\mathbf{w}^*)^{1/2} \geq \frac{7}{24}\sin\theta\|\mathrm{T}_{\cos\theta}\sigma'\|_{L_2} - \frac{1}{24}\sin^2\theta\|\mathrm{T}_{\cos\theta}\sigma'\|_{L_2} \geq \frac{1}{4}\sin\theta\|\mathrm{T}_{\cos\theta}\sigma'\|_{L_2}.$$

Thus, we have $\mathbf{v}_\mathbf{w}^* \cdot \mathbf{M}_\mathbf{w}\mathbf{v}_\mathbf{w}^* \geq (1/16)\sin^2\theta\|\mathrm{T}_{\cos\theta}\sigma'\|_{L_2}^2$. $\qquad\square$

The next lemma shows that any unit vector $\mathbf{u}$ that is orthogonal to $\mathbf{v}_\mathbf{w}^*$ has a small quadratic form.

**Lemma C.5.** *Let $\mathbf{u}$ be any unit vector that is orthogonal to $\mathbf{v}_\mathbf{w}^*$. Then $\mathbf{u}^\top\mathbf{M}_\mathbf{w}\mathbf{u} \leq 2\mathrm{OPT}$.*

*Proof.* First, if $\mathbf{u} = \mathbf{w}$, then since $\mathbf{g}_\mathbf{w}(z)\cdot\mathbf{w} = 0$, we have $\mathbf{w}^\top\mathbf{M}_\mathbf{w}\mathbf{w} = 0 \leq \mathrm{OPT}$. Now consider $\mathbf{u} \perp \mathbf{w}$. Since $\mathbf{u} \perp \mathbf{w}, \mathbf{v}_\mathbf{w}^*$ and since $\mathbf{w}^* = \cos\theta\mathbf{w} + \sin\theta\mathbf{v}_\mathbf{w}^*$, we have that $\mathbf{u}\cdot\mathbf{x}$ is independent of $\mathbf{w}^*\cdot\mathbf{x}$. Direct calculation gives (noting again that $\mathbf{Pr}_{\mathbf{x}\sim\mathcal{D}_\mathbf{x}}[\mathbf{w}\cdot\mathbf{x}\in\mathcal{E}_j] = \mathbf{Pr}_{z\sim\mathcal{N}}[z\in\mathcal{E}_j]$ since $\mathcal{D}_\mathbf{x}$ is the standard Gaussian):

$$\mathbf{u}^\top\mathbf{M}_\mathbf{w}\mathbf{u} = \mathop{\mathbf{E}}_{\mathbf{x}\sim\mathcal{D}_\mathbf{x}}[(\mathbf{u}\cdot\mathbf{g}_\mathbf{w}(z))^2]$$

$$= \mathop{\mathbf{E}}_{z\sim\mathcal{N}}\left[\left(\sum_{i=1}^{I}\mathop{\mathbf{E}}_{(\mathbf{x},y)\sim\mathcal{D}}[y(\mathbf{u}\cdot\mathbf{x}) \mid \mathbf{w}\cdot\mathbf{x}\in\mathcal{E}_i]\mathbb{1}\{z\in\mathcal{E}_i\}\right)^2\right]$$

$$= \sum_{i=1}^{I}\left(\mathop{\mathbf{E}}_{(\mathbf{x},y)\sim\mathcal{D}}\left[y(\mathbf{u}\cdot\mathbf{x})\mathbb{1}\{\mathbf{w}\cdot\mathbf{x}\in\mathcal{E}_i\}\right]/\mathbf{Pr}[\mathbf{w}\cdot\mathbf{x}\in\mathcal{E}_i]\right)^2\mathbf{Pr}[z\in\mathcal{E}_i]$$

$$= \sum_{i=1}^{I}\frac{1}{\mathbf{Pr}[z\in\mathcal{E}_i]}\left(\mathop{\mathbf{E}}_{(\mathbf{x},y)\sim\mathcal{D}}[(y-\sigma(\mathbf{w}^*\cdot\mathbf{x}))(\mathbf{u}\cdot\mathbf{x})\mathbb{1}\{\mathbf{w}\cdot\mathbf{x}\in\mathcal{E}_i\}]\right.$$

$$\left. + \mathop{\mathbf{E}}_{\mathbf{x}\sim\mathcal{D}_\mathbf{x}}[\sigma(\mathbf{w}^*\cdot\mathbf{x})(\mathbf{u}\cdot\mathbf{x})\mathbb{1}\{\mathbf{w}\cdot\mathbf{x}\in\mathcal{E}_i\}]\right)^2$$

$$= \sum_{i=1}^{I}\frac{1}{\mathbf{Pr}[z\in\mathcal{E}_i]}\left(\mathop{\mathbf{E}}_{(\mathbf{x},y)\sim\mathcal{D}}[(y-\sigma(\mathbf{w}^*\cdot\mathbf{x}))(\mathbf{u}\cdot\mathbf{x})\mathbb{1}\{\mathbf{w}\cdot\mathbf{x}\in\mathcal{E}_i\}]\right)^2,$$

where in the last inequality we used that $\mathbf{u}$ is orthogonal to $\mathbf{w}, \mathbf{w}^*$. Furthermore, it holds:

$$\left(\mathop{\mathbf{E}}_{(\mathbf{x},y)\sim\mathcal{D}}[(y-\sigma(\mathbf{w}^*\cdot\mathbf{x}))(\mathbf{u}\cdot\mathbf{x})\mathbb{1}\{\mathbf{w}\cdot\mathbf{x}\in\mathcal{E}_i\}]\right)^2$$

$$\overset{(i)}{\leq} \mathop{\mathbf{E}}_{(\mathbf{x},y)\sim\mathcal{D}}[(y-\sigma(\mathbf{w}^*\cdot\mathbf{x}))^2\mathbb{1}\{\mathbf{w}\cdot\mathbf{x}\in\mathcal{E}_i\}]\mathop{\mathbf{E}}_{\mathbf{x}\sim\mathcal{D}_\mathbf{x}}[(\mathbf{u}\cdot\mathbf{x})^2\mathbb{1}\{\mathbf{w}\cdot\mathbf{x}\in\mathcal{E}_i\}]$$

$$\overset{(ii)}{\leq} 2\mathop{\mathbf{E}}_{(\mathbf{x},y)\sim\mathcal{D}}[(y-\sigma(\mathbf{w}^*\cdot\mathbf{x}))^2\mathbb{1}\{\mathbf{w}\cdot\mathbf{x}\in\mathcal{E}_i\}]\mathbf{Pr}[z\in\mathcal{E}_i],$$

where in $(i)$ we applied Cauchy-Schwarz and in $(ii)$ we used the independence between $\mathbf{u}\cdot\mathbf{x}$ and $\mathbf{w}\cdot\mathbf{x}$ since $\mathbf{u}\perp\mathbf{w}$. Therefore,

$$\mathbf{u}^\top\mathbf{M}_\mathbf{w}\mathbf{u} \leq \sum_{i=1}^{I}2\mathop{\mathbf{E}}_{(\mathbf{x},y)\sim\mathcal{D}}[(y-\sigma(\mathbf{w}^*\cdot\mathbf{x}))^2\mathbb{1}\{\mathbf{w}\cdot\mathbf{x}\in\mathcal{E}_i\}]$$

$$\leq 2\mathop{\mathbf{E}}_{(\mathbf{x},y)\sim\mathcal{D}}[(\sigma(\mathbf{w}^*\cdot\mathbf{x})-y)^2\mathbb{1}\{\mathbf{w}\cdot\mathbf{x}\in[-M',M']\}] \leq 2\mathrm{OPT}.$$

$\square$

Then we can show that the top eigenvector $\mathbf{v_w}$ of $\mathbf{M_w}$ correlates strongly with the target direction $\mathbf{v_w^*}$.

**Lemma C.6.** *Let $\mathbf{v_w}$ be the top eigenvector of $\mathbf{M_w}$. Suppose $\sin\theta \geq 40\sqrt{\mathrm{OPT}}/\|\mathrm{T}_{\cos\theta}\sigma'\|_{L_2}$. Then $\mathbf{v_w} \cdot \mathbf{v_w^*} \geq \sqrt{3}/2$.*

*Proof.* Since $\mathbf{v_w}$ is the top eigenvector of $\mathbf{M_w}$, by Lemma C.3, it holds $\mathbf{v_w^\top M_w v_w} \geq (\mathbf{v_w^*})^\top \mathbf{M_w v_w^*} \geq (4/9)\sin^2\theta\|\mathrm{T}_{\cos\theta}\sigma'\|_{L_2}^2 > 0$. Since $\mathbf{Mw} = 0$, it must hold $\mathbf{v_w} \cdot \mathbf{w} = 0$, otherwise we would be able to find another eigenvector that has a larger eigenvalue. Suppose that $\mathbf{v_w} = a\mathbf{v_w^*} + b\mathbf{u}$, where $\mathbf{u} \perp \mathbf{w}, \mathbf{w}^*$ and $a^2 + b^2 = 1$. Then, we obtain

$$a^2(\mathbf{v_w^*})^\top \mathbf{M_w v_w^*} + b^2\mathbf{u}^\top \mathbf{M_w u} + 2ab\mathbf{u}^\top \mathbf{M_w v_w^*} = \mathbf{v_w^\top M_w v_w} \geq (\mathbf{v_w^*})^\top \mathbf{M_w v_w^*}.$$

By Lemma C.5 we have $\mathbf{u}^\top \mathbf{M_w u} \leq \mathrm{OPT}$. Furthermore, note that by Cauchy-Schwarz

$$\mathbf{u}^\top \mathbf{M_w v_w^*} = \mathop{\mathbf{E}}_{z\sim\mathcal{N}}[(\mathbf{u}\cdot\mathbf{g_w}(z))(\mathbf{v_w^*}\cdot\mathbf{g_w}(z))] \leq \sqrt{\mathop{\mathbf{E}}_{z\sim\mathcal{N}}[(\mathbf{u}\cdot\mathbf{g_w}(z))^2]\mathop{\mathbf{E}}_{z\sim\mathcal{N}}[(\mathbf{v_w^*}\cdot\mathbf{g_w}(z))^2]}$$

$$= \sqrt{\mathbf{u}^\top \mathbf{M_w u}}\sqrt{(\mathbf{v_w^*})^\top \mathbf{M_w v_w^*}} \leq \sqrt{\mathrm{OPT}}\sqrt{(\mathbf{v_w^*})^\top \mathbf{M_w v_w^*}}.$$

Since $a^2 + b^2 = 1$, we get

$$(1-a^2)\mathbf{v_w^*}\cdot\mathbf{M_w v_w^*} = b^2\mathbf{v_w^*}\cdot\mathbf{M_w v_w^*} \leq b^2\mathrm{OPT} + 2ab\sqrt{\mathrm{OPT}}\sqrt{\mathbf{v_w^*}\cdot\mathbf{M_w v_w^*}},$$

which implies that

$$\mathbf{v_w}\cdot\mathbf{v_w^*} = a \geq \frac{b}{2}\frac{\mathbf{v_w^*}\cdot\mathbf{M_w v_w^*} - \mathrm{OPT}}{\sqrt{\mathrm{OPT}}\sqrt{\mathbf{v_w^*}\cdot\mathbf{M_w v_w^*}}}.$$

Recall that $(\mathbf{v_w^*})^\top \mathbf{M_w v_w^*} \geq (1/16)\sin^2\theta\|\mathrm{T}_{\cos\theta}\sigma'\|_{L_2}^2$. Since we have assumed $\sin\theta\|\mathrm{T}_{\cos\theta}\sigma'\|_{L_2} \geq 40\sqrt{\mathrm{OPT}}$, it holds $(\mathbf{v_w^*})^\top \mathbf{M_w v_w^*} \geq 100\mathrm{OPT}$. Thus, we obtain

$$\mathbf{v_w}\cdot\mathbf{v_w^*} = \cos(\theta(\mathbf{v_w},\mathbf{v_w^*})) \geq \sin(\theta(\mathbf{v_w},\mathbf{v_w^*}))(99/20) \geq 3\sin(\theta(\mathbf{v_w},\mathbf{v_w^*})),$$

hence $\tan(\theta(\mathbf{v_w},\mathbf{v_w^*})) \leq 1/3$ and therefore $\mathbf{v_w}\cdot\mathbf{v_w^*} \geq \sqrt{3}/2$. $\square$

### C.2 Proof of Proposition C.1

Let $\mathbf{g_w}(z)$ be the vector defined in (Grad) and let $\mathbf{M_w} = \mathbf{E}_{z\sim\mathcal{N}}[\mathbf{g_w}(z)\mathbf{g_w}(z)^\top]$. In Lemma C.6 we proved that for any vector $\mathbf{w}$, whenever $\sin(\theta(\mathbf{w},\mathbf{w}^*)) \geq 40\sqrt{\mathrm{OPT}}\|\mathrm{T}_{\cos\theta(\mathbf{w},\mathbf{w}^*)}\sigma'\|_{L_2}$, one of the top eigenvector $\mathbf{v_w}$ of $\mathbf{M_w}$ correlates with $\mathbf{v_w^*}$: $\mathbf{v_w}\cdot\mathbf{v_w^*} \geq \sqrt{3}/2$, i.e., $\theta(\mathbf{v_w^*},\mathbf{v_w}) \leq \pi/6$. Next, in Lemma C.9, we proved that using $N = \Theta(d^2B^{12}L^8/\epsilon^{10}\log(dBL/(\delta\epsilon)))$ samples, for any vector $\mathbf{w}$, the empirical matrix $\widehat{\mathbf{M}}_\mathbf{w}$ satisfies $\|\widehat{\mathbf{M}}_\mathbf{w} - \mathbf{M_w}\|_2 \leq \epsilon$. Therefore, using Wedin's Theorem (Fact C.7), we know that for any vector $\mathbf{w}$, the top eigenvector $\widehat{\mathbf{v}}_\mathbf{w}$ of $\widehat{\mathbf{M}}_\mathbf{w}$ satisfies $\sin(\theta(\mathbf{v_w},\widehat{\mathbf{v}}_\mathbf{w})) \leq \epsilon/(\rho_1 - \rho_2 - \epsilon)$. Since in Lemma C.8 we showed that when $\sin(\theta(\mathbf{w},\mathbf{w}^*)) \geq 40\sqrt{\mathrm{OPT}}\|\mathrm{T}_{\cos\theta(\mathbf{w},\mathbf{w}^*)}\sigma'\|_{L_2}$, it holds $\rho_1 - \rho_2 \geq 60\mathrm{OPT} \geq 60\epsilon$, we then have that $\theta(\mathbf{v_w},\widehat{\mathbf{v}}_\mathbf{w}) \leq 1/59$, indicating $\theta(\widehat{\mathbf{v}}_\mathbf{w},\mathbf{v_w^*}) \leq \theta(\widehat{\mathbf{v}}_\mathbf{w},\mathbf{v_w}) + \theta(\mathbf{v_w},\mathbf{v_w^*}) \leq 1/59 + \pi/6 \leq \pi/4$. Therefore, let $\mathbf{u}$ be such eigenvector $\widehat{\mathbf{v}}_\mathbf{w}$ of $\widehat{\mathbf{M}}_\mathbf{w}$ that correlates positively with $\mathbf{v_w^*}$. Note that by definition of $\widehat{\mathbf{M}}_\mathbf{w}$, it must hold $\mathbf{u} \perp \mathbf{w}$. Thus we have $\mathbf{u}\cdot\mathbf{w}^* = \mathbf{u}\cdot(\cos(\theta(\mathbf{w},\mathbf{w}^*))\mathbf{w} + \sin(\theta(\mathbf{w},\mathbf{w}^*))\mathbf{v_w^*}) = \sin(\theta(\mathbf{w},\mathbf{w}^*))\mathbf{u}\cdot\mathbf{v_w^*} \geq (\sqrt{2}/2)\sin(\theta(\mathbf{w},\mathbf{w}^*))$. Letting $\delta = 0.01$ finishes the proof.

### C.3 Determining the Sample Complexity

Since we only have access to the empirical estimate $\widehat{\mathbf{M}}_\mathbf{w}$, we will use the following Wedin's theorem to bound the error between the empirical top eigenvector $\widehat{\mathbf{v}}_\mathbf{w}$ and the population top eigenvector $\mathbf{v_w}$:

**Fact C.7** (Wedin's theorem). *Let $\theta(\mathbf{v_w},\widehat{\mathbf{v}}_\mathbf{w})$ be the angle between the top eigenvectors $\mathbf{v_w} \in \mathbb{R}^d$ and $\widehat{\mathbf{v}}_\mathbf{w} \in \mathbb{R}^d$ of $\mathbf{M_w}$ and $\widehat{\mathbf{M}}_\mathbf{w}$ respectively. Let $\rho_1$ and $\rho_2$ be the first 2 eigenvalues of $\mathbf{M_w}$. Then, it holds that:*

$$\sin(\theta(\mathbf{v_w},\widehat{\mathbf{v}}_\mathbf{w})) \leq \frac{\|\mathbf{M_w} - \widehat{\mathbf{M}}_\mathbf{w}\|_2}{\rho_1 - \rho_2 - \|\mathbf{M_w} - \widehat{\mathbf{M}}_\mathbf{w}\|_2}.$$

Next, we bound the eigengap between the top eigenvalue and the rest.

**Lemma C.8.** *Suppose* $\sin\theta\|T_{\cos\theta}\sigma'\|_{L_2} \geq 40\sqrt{\text{OPT}}$. *Let* $\mathbf{p}$ *be any eigenvector of* $\mathbf{M_w}$ *orthogonal to* $\mathbf{v_w}$. *Then,* $\mathbf{v_w}^\top\mathbf{M_w}\mathbf{v_w} - \mathbf{p}^\top\mathbf{M_w}\mathbf{p} \geq (1/24)\sin^2\theta\|T_{\cos\theta}\sigma'\|_{L_2}^2 \geq 60\text{OPT}$.

*Proof.* In Lemma C.6, we showed that $\mathbf{v_w} = a\mathbf{v_w^*} + b\mathbf{u}$ with $\mathbf{u} \perp \mathbf{v_w^*}$, $a \geq \sqrt{3}/2$, $a^2 + b^2 = 1$. Assume that $\mathbf{p} = a_1\mathbf{v_w^*} + b_1\mathbf{u} + \mathbf{u}'$ where $\mathbf{u}'$ is a vector orthogonal to both $\mathbf{v_w^*}$ and $\mathbf{u}$ and $a_1^2 + b_1^2 \leq 1$. Since $\mathbf{p} \cdot \mathbf{v_w} = 0$, we have $aa_1 + bb_1 = 0$, which implies that $a^2a_1^2 = b^2b_1^2 \leq (1 - a^2)(1 - a_1^2)$. Rearranging the terms, it yields $a_1^2 \leq 1 - a^2 \leq 1/4$, and therefore we have $a_1 \leq 1/2$. Denote $b_1\mathbf{u} + \mathbf{u}' = \mathbf{v}'$, and we have $\mathbf{p} = a_1\mathbf{v_w^*} + \mathbf{v}'$ and $\|\mathbf{v}'\|_2 \leq 1$. Since $\mathbf{v}' \perp \mathbf{v_w^*}$, by Lemma C.5 we have $(\mathbf{v}')^\top\mathbf{M_w}\mathbf{v}' \leq 2\text{OPT}\|\mathbf{v}'\|_2^2 \leq 2\text{OPT}$. Then the eigenvalue of $\mathbf{p}$ is bounded above by

$$\mathbf{p}^\top\mathbf{M_w}\mathbf{p} = a_1^2\mathbf{v_w^*} \cdot \mathbf{M_w}\mathbf{v_w^*} + a_1\mathbf{v_w^*} \cdot \mathbf{M_w}\mathbf{v}' + \mathbf{v}' \cdot \mathbf{M_w}\mathbf{v}'$$
$$\leq (1/4)\mathbf{v_w^*} \cdot \mathbf{M_w}\mathbf{v_w^*} + (1/2)\sqrt{2\text{OPT}(\mathbf{v_w^*} \cdot \mathbf{M_w}\mathbf{v_w^*})} + 2\text{OPT},$$

where in the last inequality we applied Cauchy-Schwarz. Since we have assumed $\sin\theta\|T_{\cos\theta}\sigma'\|_{L_2} \geq 40\sqrt{\text{OPT}}$, then Lemma C.3 implies that $\mathbf{v_w^*} \cdot \mathbf{M_w}\mathbf{v_w^*} \geq (1/16)\sin^2\theta\|T_{\cos\theta}\sigma'\|_{L_2}^2 \geq 100\text{OPT}$, therefore we obtain $\mathbf{p}^\top\mathbf{M_w}\mathbf{p} \leq (1/3)\mathbf{v_w^*} \cdot \mathbf{M_w}\mathbf{v_w^*}$. Thus, the eigengap between $\mathbf{v_w} \cdot \mathbf{M_w}\mathbf{v_w}$ and $\mathbf{p}^\top\mathbf{M_w}\mathbf{p}$ can be bounded above by

$$\mathbf{v_w}^\top\mathbf{M_w}\mathbf{v_w} - \mathbf{p}^\top\mathbf{M_w}\mathbf{p} \geq (2/3)\mathbf{v_w^*} \cdot \mathbf{M_w}\mathbf{v_w^*} \geq (1/24)\sin^2\theta\|T_{\cos\theta}\sigma'\|_{L_2}^2 \geq (200/3)\text{OPT}.$$
$$\square$$

By Lemma C.8 we immediately get that $\rho_1 - \rho_2 \geq c\sin^2\theta\|T_{\cos\theta}\sigma'\|_{L_2}^2$, therefore, to guarantee that $\sin(\theta(\mathbf{v_w}, \widehat{\mathbf{v}}_{\mathbf{w}})) \ll 1$, if suffices to ensure that $\|\mathbf{M_w} - \widehat{\mathbf{M}}_{\mathbf{w}}\|_2 \leq \epsilon \lesssim \text{OPT} \lesssim \sin^2\theta\|T_{\cos\theta}\sigma'\|_{L_2}^2$.

**Lemma C.9** (Sample Complexity)**.** *Draw* $N = \Theta(d^2B^{12}L^8/\epsilon^{10}\log(dBL/(\delta\epsilon)))$ *independent samples from* $\mathcal{D}$. *Let*

$$\widehat{\mathbf{g}}_{\mathbf{w}}^{(j)} = \frac{1}{N}\sum_{i=1}^N y^{(i)}(\mathbf{x}^{(i)})^{\perp\mathbf{w}}\mathbb{1}\{\mathbf{w} \cdot \mathbf{x}^{(i)} \in \mathcal{E}_j\}, \ j \in [I] \ ,$$

$$\widehat{\mathbf{M}}_{\mathbf{w}} := \sum_{j=1}^I \frac{\widehat{\mathbf{g}}_{\mathbf{w}}^{(j)}(\widehat{\mathbf{g}}_{\mathbf{w}}^{(j)})^\top}{\mathbf{Pr}[z \in \mathcal{E}_j]} = \underset{z\sim\mathcal{N}}{\mathbf{E}}\left[\left(\sum_{j=1}^I \frac{\widehat{\mathbf{g}}_{\mathbf{w}}^{(j)}\mathbb{1}\{z \in \mathcal{E}_j\}}{\mathbf{Pr}[z \in \mathcal{E}_j]}\right)\left(\sum_{j=1}^I \frac{\widehat{\mathbf{g}}_{\mathbf{w}}^{(j)}\mathbb{1}\{z \in \mathcal{E}_j\}}{\mathbf{Pr}[z \in \mathcal{E}_j]}\right)^\top\right].$$

*Then, with probability at least* $1 - \delta$, *for any* $\|\mathbf{w}\|_2 = 1$, *we have* $\|\widehat{\mathbf{M}}_{\mathbf{w}} - \mathbf{M_w}\|_2 \leq \epsilon$.

*Proof.* Since $\mathbf{g_w}(z)$ is a piecewise constant function on the intervals $\mathcal{E}_j = [a_j, a_{j+1})$ where $a_j = -M' + j\Delta$ and $\Delta = \epsilon^2/(BL)^2$, as defined in (Grad), it suffices to approximate the (vector) value of $\mathbf{g_w}(z)$ on each interval. First, for any $j \in [I]$, we observe that $\mathbf{r}_j := y\mathbf{x}^{\perp\mathbf{w}}\mathbb{1}\{\mathbf{w} \cdot \mathbf{x} \in \mathcal{E}_j\}$ is a sub-Gaussian random variable with parameter $B$. To see this, it suffices to show that for any unit vector $\mathbf{u}$ it holds $\|\mathbf{u} \cdot \mathbf{r}_j\|_{L_p} \lesssim B\sqrt{p}$ for any $p \geq 1$. Since $\mathbf{w} \cdot \mathbf{r}_j = 0$, we only need to consider $\mathbf{u}$ that is orthogonal to $\mathbf{w}$. Direct calculation yields:

$$\underset{(\mathbf{x},y)\sim\mathcal{D}}{\mathbf{E}}[|\mathbf{u} \cdot \mathbf{r}_j|^p] = \underset{(\mathbf{x},y)\sim\mathcal{D}}{\mathbf{E}}\left[\left|y(\mathbf{x}^{\perp\mathbf{w}} \cdot \mathbf{u})\mathbb{1}\{\mathbf{w} \cdot \mathbf{x} \in \mathcal{E}_j\}\right|^p\right]$$
$$\leq B^p \underset{\mathbf{x}\sim\mathcal{D}_\mathbf{x}}{\mathbf{E}}[|\mathbf{u} \cdot \mathbf{x}|^p]\mathbf{Pr}[z \in \mathcal{E}_j] \leq B^p(c\sqrt{p})^p.$$

Thus, $\|\mathbf{u} \cdot \mathbf{r}_j\|_{L_p} \lesssim B\sqrt{p}$ for any unit vector $\mathbf{u}$ and hence $\mathbf{r}_j$ is $B$-sub-Gaussian.

Now sample $N$ independent samples from $\mathcal{D}$, $\{(\mathbf{x}^{(i)}, y^{(i)})\}_{i=1}^N$, creating $N$ independent vectors

$$\mathbf{r}_j^{(i)} := y^{(i)}(\mathbf{x}^{(i)})^{\perp\mathbf{w}}\mathbb{1}\{\mathbf{w} \cdot \mathbf{x}^{(i)} \in \mathcal{E}_j\}.$$

Then we know that $(1/N)\sum_{i=1}^N(\mathbf{r}_j^{(i)} - \mathbf{E}_{(\mathbf{x},y)\sim\mathcal{D}}[\mathbf{r}_j])$ is a sub-Gaussian random vector with parameter $B/\sqrt{N}$. Then, using standard sub-Gaussian vector concentration inequality, we obtain

$$\mathbf{Pr}\left[\left\|\frac{1}{N}\sum_{i=1}^N \mathbf{r}_j^{(i)} - \underset{(\mathbf{x},y)\sim\mathcal{D}}{\mathbf{E}}[\mathbf{r}_j]\right\|_2 \geq s\right] \leq \exp\left(-\frac{cs^2N}{B^2d}\right).$$

We let $s = \epsilon/(CB)(\epsilon/BL)^4$. Observe that $\mathbf{Pr}[z \in \mathcal{E}_j] \geq (a_{j+1} - a_j)\exp(-a_{j+1}^2/2) = \Delta\exp(-a_{j+1}^2/2)$, where $\Delta = \epsilon^2/(BL)^2$. Since $|a_{j+1}| \leq M'$ for all $j \in [I-1]$, we have $\exp(-a_{j+1}^2/2) \gtrsim \exp(-(M')^2/2) \gtrsim \mathbf{Pr}[|z| \geq M'] = \epsilon^2/(BL)^2$. Thus, we have $\mathbf{Pr}[z \in \mathcal{E}_j] \geq (\epsilon/BL)^4$ and hence $s \leq (\epsilon/(CB))\mathbf{Pr}[z \in \mathcal{E}_j]$ for all $j \in [I]$. Therefore choosing $N = \Theta(dB^{12}L^8/\epsilon^{10}\log(1/\delta))$, we have that with probability at least $1 - \delta$,

$$\left\|\frac{1}{N}\sum_{i=1}^N \mathbf{r}_j^{(i)} - \mathop{\mathbf{E}}_{(\mathbf{x},y)\sim\mathcal{D}}[\mathbf{r}_j]\right\|_2 \leq \frac{\epsilon\,\mathbf{Pr}[z \in \mathcal{E}_j]}{CB},$$

where $C$ is a large absolute constant. However, since there are $I = \tilde{O}((BL/\epsilon)^2)$ pieces of intervals $\mathcal{E}_j$ (by the definition of $I$ in (Grad)), to guarantee that $(1/N)\sum_{i=1}^N \mathbf{r}_j$ is close to $\mathbf{E}_{(\mathbf{x},y)\sim\mathcal{D}}[\mathbf{r}_j]$ on every interval, setting $\delta \leftarrow \delta\epsilon^2/(BL)^2$ and applying a union bound, we obtain that using $N = \Theta(dB^{12}L^8/\epsilon^{10}\log(dBL/(\delta\epsilon)))$ samples, with probability at least $1 - \delta$ it holds $\|(1/N)\sum_{i=1}^N \mathbf{r}_j - \mathbf{E}_{(\mathbf{x},y)\sim\mathcal{D}}[\mathbf{r}_j]\|_2 \lesssim \epsilon/(CB)\mathbf{Pr}[z \in \mathcal{E}_j]$ for every $j \in [I]$. Thus, letting

$$\widehat{\mathbf{g}}_{\mathbf{w}}^{(j)} := \frac{1}{N}\sum_{i=1}^N y^{(i)}(\mathbf{x}^{(i)})^{\perp\mathbf{w}}\mathbb{1}\{\mathbf{w}\cdot\mathbf{x}^{(i)} \in \mathcal{E}_j\} = \frac{1}{N}\sum_{i=1}^N \mathbf{r}_j^{(i)},$$

we have that with probability at least $1 - \delta$, it holds $\|\widehat{\mathbf{g}}_{\mathbf{w}}^{(j)} - \mathbf{E}_{(\mathbf{x},y)\sim\mathcal{D}}[y\mathbf{x}^{\perp\mathbf{w}}\mathbb{1}\{\mathbf{w}\cdot\mathbf{x} \in \mathcal{E}_j\}]\|_2 \leq \epsilon\,\mathbf{Pr}[z \in \mathcal{E}_j]/(CB)$, for any $j \in [I]$.

Now define

$$\widehat{\mathbf{M}}_{\mathbf{w}} := \sum_{j=1}^I \frac{\widehat{\mathbf{g}}_{\mathbf{w}}^{(j)}(\widehat{\mathbf{g}}_{\mathbf{w}}^{(j)})^\top}{\mathbf{Pr}[z \in \mathcal{E}_j]} = \mathop{\mathbf{E}}_{z\sim\mathcal{N}}\left[\left(\sum_{j=1}^I \frac{\widehat{\mathbf{g}}_{\mathbf{w}}^{(j)}\mathbb{1}\{z \in \mathcal{E}_j\}}{\mathbf{Pr}[z \in \mathcal{E}_j]}\right)\left(\sum_{j=1}^I \frac{\widehat{\mathbf{g}}_{\mathbf{w}}^{(j)}\mathbb{1}\{z \in \mathcal{E}_j\}}{\mathbf{Pr}[z \in \mathcal{E}_j]}\right)^\top\right].$$

Oberve that since $\widehat{\mathbf{g}}_{\mathbf{w}}^{(j)} \perp \mathbf{w}$ we have $\mathbf{w}\cdot\widehat{\mathbf{M}}_{\mathbf{w}}\mathbf{w} = 0$. Similarly we have $\mathbf{w}\cdot\mathbf{M}_{\mathbf{w}}\mathbf{w} = 0$. Now consider any unit vector $\mathbf{u}$ that is orthogonal to $\mathbf{w}$. Then, by the definition of $\mathbf{g}_{\mathbf{w}}(z)$ and that $\mathbf{M}_{\mathbf{w}} = \mathbf{E}_{z\sim\mathcal{N}}[\mathbf{g}_{\mathbf{w}}(z)\mathbf{g}_{\mathbf{w}}(z)^\top]$, we have

$$\left|\mathbf{u}^\top(\widehat{\mathbf{M}}_{\mathbf{w}} - \mathbf{M}_{\mathbf{w}})\mathbf{u}\right| = \left|\mathbf{u}^\top\widehat{\mathbf{M}}_{\mathbf{w}}\mathbf{u} - \mathbf{u}^\top\mathop{\mathbf{E}}_{z\sim\mathcal{N}}[\mathbf{g}_{\mathbf{w}}(z)\mathbf{g}_{\mathbf{w}}(z)^\top]\mathbf{u}\right|$$

$$= \left|\sum_{j=1}^I \frac{(\mathbf{u}\cdot\widehat{\mathbf{g}}_{\mathbf{w}}^{(j)})^2}{\mathbf{Pr}[z \in \mathcal{E}_j]} - \sum_{j=1}^I \frac{(\mathbf{u}\cdot\mathbf{E}_{(\mathbf{x},y)\sim\mathcal{D}}[y\mathbf{x}^{\perp\mathbf{w}}\mathbb{1}\{\mathbf{w}\cdot\mathbf{x} \in \mathcal{E}_j\}])^2}{\mathbf{Pr}[z \in \mathcal{E}_j]}\right|$$

$$\leq \sum_{j=1}^I \frac{1}{\mathbf{Pr}[z \in \mathcal{E}_j]}\left|\mathbf{u}\cdot(\widehat{\mathbf{g}}_{\mathbf{w}}^{(j)} - \mathop{\mathbf{E}}_{(\mathbf{x},y)\sim\mathcal{D}}[y\mathbf{x}^{\perp\mathbf{w}}\mathbb{1}\{\mathbf{w}\cdot\mathbf{x} \in \mathcal{E}_j\}])\right|\left|\mathbf{u}\cdot(\widehat{\mathbf{g}}_{\mathbf{w}}^{(j)} + \mathop{\mathbf{E}}_{(\mathbf{x},y)\sim\mathcal{D}}[y\mathbf{x}^{\perp\mathbf{w}}\mathbb{1}\{\mathbf{w}\cdot\mathbf{x} \in \mathcal{E}_j\}])\right|$$

$$\leq \sum_{j=1}^I \frac{1}{\mathbf{Pr}[z \in \mathcal{E}_j]}\left\|\widehat{\mathbf{g}}_{\mathbf{w}}^{(j)} - \mathop{\mathbf{E}}_{(\mathbf{x},y)\sim\mathcal{D}}[y\mathbf{x}^{\perp\mathbf{w}}\mathbb{1}\{\mathbf{w}\cdot\mathbf{x} \in \mathcal{E}_j\}]\right\|_2\left\|\widehat{\mathbf{g}}_{\mathbf{w}}^{(j)} + \mathop{\mathbf{E}}_{(\mathbf{x},y)\sim\mathcal{D}}[y\mathbf{x}^{\perp\mathbf{w}}\mathbb{1}\{\mathbf{w}\cdot\mathbf{x} \in \mathcal{E}_j\}]\right\|_2$$

Note that we have $\|\widehat{\mathbf{g}}_{\mathbf{w}}^{(j)} - \mathbf{E}_{(\mathbf{x},y)\sim\mathcal{D}}[y\mathbf{x}^{\perp\mathbf{w}}\mathbb{1}\{\mathbf{w}\cdot\mathbf{x} \in \mathcal{E}_j\}]\|_2 \leq \epsilon\,\mathbf{Pr}[z \in \mathcal{E}_j]/(CB)$, for any $j \in [I]$, therefore,

$$\left|\mathbf{u}^\top(\widehat{\mathbf{M}}_{\mathbf{w}} - \mathbf{M}_{\mathbf{w}})\mathbf{u}\right|$$

$$\leq \sum_{j=1}^I \frac{1}{\mathbf{Pr}[z \in \mathcal{E}_j]}\frac{\epsilon\,\mathbf{Pr}[z \in \mathcal{E}_j]}{CB}\left(\frac{\epsilon\,\mathbf{Pr}[z \in \mathcal{E}_j]}{CB} + 2\left\|\mathop{\mathbf{E}}_{(\mathbf{x},y)\sim\mathcal{D}}[y\mathbf{x}^{\perp\mathbf{w}}\mathbb{1}\{\mathbf{w}\cdot\mathbf{x} \in \mathcal{E}_j\}]\right\|_2\right)$$

$$\leq \sum_{j=1}^I \frac{\epsilon}{CB}\left(\frac{\epsilon\,\mathbf{Pr}[z \in \mathcal{E}_j]}{CB} + 2\left\|\mathop{\mathbf{E}}_{(\mathbf{x},y)\sim\mathcal{D}}[y\mathbf{x}^{\perp\mathbf{w}}\mathbb{1}\{\mathbf{w}\cdot\mathbf{x} \in \mathcal{E}_j\}]\right\|_2\right).$$

Observe that $\|\mathbf{E}_{(\mathbf{x},y)\sim\mathcal{D}}[y\mathbf{x}^{\perp\mathbf{w}}\mathbb{1}\{\mathbf{w}\cdot\mathbf{x} \in \mathcal{E}_j\}]\|_2 \lesssim B\,\mathbf{Pr}[z \in \mathcal{E}_j]$ since

$$\left\|\mathop{\mathbf{E}}_{(\mathbf{x},y)\sim\mathcal{D}}[y\mathbf{x}^{\perp\mathbf{w}}\mathbb{1}\{\mathbf{w}\cdot\mathbf{x} \in \mathcal{E}_j\}]\right\|_2 = \sup_{\|\mathbf{u}\|_2=1}\left|\mathop{\mathbf{E}}_{(\mathbf{x},y)\sim\mathcal{D}}[y(\mathbf{u}\cdot\mathbf{x}^{\perp\mathbf{w}})\mathbb{1}\{\mathbf{w}\cdot\mathbf{x} \in \mathcal{E}_j\}]\right|$$

$$\leq \sup_{\|\mathbf{u}\|_2=1}\mathop{\mathbf{E}}_{(\mathbf{x},y)\sim\mathcal{D}}[|y||(\mathbf{u}\cdot\mathbf{x}^{\perp\mathbf{w}})|\mathbb{1}\{\mathbf{w}\cdot\mathbf{x} \in \mathcal{E}_j\}] \lesssim B\,\mathbf{Pr}[z \in \mathcal{E}_j], \tag{9}$$

where we used the assumption that $|y| \leq B$ and the facts that $\mathbf{u} \cdot \mathbf{x}^{\perp \mathbf{w}}$ and $\mathbf{w} \cdot \mathbf{x}$ are independent and that $|\mathbf{E}_{\mathbf{x} \sim \mathcal{D}_{\mathbf{x}}}[\mathbf{u} \cdot \mathbf{x}^{\perp \mathbf{w}}]| \leq 2$. Thus, in summary, we have

$$\left| \mathbf{u}^{\top} (\widehat{\mathbf{M}}_{\mathbf{w}} - \mathbf{M}_{\mathbf{w}}) \mathbf{u} \right| \leq \sum_{j=1}^{I} \frac{\epsilon}{CB} (\epsilon/(CB) + B) \mathbf{Pr}[z \in \mathcal{E}_j] \leq \epsilon.$$

This implies that $\|\widehat{\mathbf{M}}_{\mathbf{w}} - \mathbf{M}_{\mathbf{w}}\|_2 \leq \epsilon$.

Finally, we show that a $\tilde{O}(\epsilon^3)$-net of $\mathbf{w}$'s covers all the functions $\mathbf{g}_{\mathbf{w}}(z)$ and consequently covers all the matrices $\mathbf{M}_{\mathbf{w}}$.

**Claim C.10.** *Given a unit vector $\mathbf{w}$, let $\mathbf{w}'$ be a vector such that $\|\mathbf{w}'\|_2 = 1$ and $\|\mathbf{w}' - \mathbf{w}\|_2 \leq \epsilon^3/(CB^4L^2\sqrt{\log(BL/\epsilon)})$ for some large absolute constant $C$. Then, for all $z \in \mathbb{R}$, it holds that $\|\mathbf{g}_{\mathbf{w}}(z) - \mathbf{g}_{\mathbf{w}'}(z)\|_2 \leq \epsilon/(CB)$.*

*Proof.* For any unit vector $\mathbf{w}'$ such that $\|\mathbf{w}' - \mathbf{w}\|_2 \leq \epsilon^3/(CB^4L^2\sqrt{\log(BL/\epsilon)}) \leq \epsilon^3/(CB^4L^2M')$ and any unit vector $\mathbf{u}$, we have

$$\mathbf{u} \cdot (\mathbf{g}_{\mathbf{w}}(z) - \mathbf{g}_{\mathbf{w}'}(z)) = \sum_{j=1}^{I} \frac{\mathbf{E}_{(\mathbf{x},y) \sim \mathcal{D}}[y((\mathbf{u} \cdot \mathbf{x}^{\perp \mathbf{w}})\mathbb{1}\{\mathbf{w} \cdot \mathbf{x} \in \mathcal{E}_j\} - (\mathbf{u} \cdot \mathbf{x}^{\perp \mathbf{w}'}\mathbb{1}\{\mathbf{w}' \cdot \mathbf{x} \in \mathcal{E}_j\})]}{\mathbf{Pr}[z \in \mathcal{E}_j]} \mathbb{1}\{z \in \mathcal{E}_j\}.$$

Let $\mathbf{w}' = \mathbf{w} + \mathbf{q}$ such that $\|\mathbf{q}\|_2 \leq \epsilon^3/(CB^4L^2\sqrt{\log(BL/\epsilon)})$. Then, $\|\mathbf{u}^{\perp \mathbf{w}} - \mathbf{u}^{\perp \mathbf{w}'}\|_2 = \|(\mathbf{w} \cdot \mathbf{u})\mathbf{w} - (\mathbf{w}' \cdot \mathbf{u})\mathbf{w}'\|_2 \leq \|\mathbf{q}\|_2 \leq \epsilon/(CB^4L^2)$. Furthermore, note that

$$\mathbf{E}_{\mathbf{x} \sim \mathcal{D}_{\mathbf{x}}} [|\mathbb{1}\{\mathbf{w} \cdot \mathbf{x} \in [a_j, a_{j+1})\} - \mathbb{1}\{\mathbf{w}' \cdot \mathbf{x} \in [a_j, a_{j+1})\}|]$$

$$= \mathbf{E}_{\mathbf{x} \sim \mathcal{D}_{\mathbf{x}}} [|\mathbb{1}\{\mathbf{w} \cdot \mathbf{x} \geq a_j\} - \mathbb{1}\{\mathbf{w}' \cdot \mathbf{x} \geq a_j\} - (\mathbb{1}\{\mathbf{w} \cdot \mathbf{x} \geq a_{j+1}\} - \mathbb{1}\{\mathbf{w}' \cdot \mathbf{x} \geq a_{j+1}\})|]$$

$$\leq \mathbf{E}_{\mathbf{x} \sim \mathcal{D}_{\mathbf{x}}} [|\mathbb{1}\{\mathbf{w} \cdot \mathbf{x} \geq a_j\} - \mathbb{1}\{\mathbf{w}' \cdot \mathbf{x} \geq a_j\}|] + \mathbf{E}_{\mathbf{x} \sim \mathcal{D}_{\mathbf{x}}} [|\mathbb{1}\{\mathbf{w} \cdot \mathbf{x} \geq a_{j+1}\} - \mathbb{1}\{\mathbf{w}' \cdot \mathbf{x} \geq a_{j+1}\}|].$$

It is well-known that $\mathbf{Pr}[\mathbb{1}\{\mathbf{w} \cdot \mathbf{x} \geq t\} \neq \mathbb{1}\{\mathbf{w}' \cdot \mathbf{x} \geq t\}] \leq \theta(\mathbf{w}, \mathbf{w}') \exp(-t^2)/(2\pi)$ (see Fact C.11 from [DKTZ22b]). Therefore, since $\theta(\mathbf{w}, \mathbf{w}') \lesssim \|\mathbf{w} - \mathbf{w}'\|_2 \leq \epsilon^3/(CB^4L^2M')$, for small $\epsilon$ we have

$$\mathbf{E}_{\mathbf{x} \sim \mathcal{D}_{\mathbf{x}}} [|\mathbb{1}\{\mathbf{w} \cdot \mathbf{x} \in \mathcal{E}_j\} - \mathbb{1}\{\mathbf{w}' \cdot \mathbf{x} \in \mathcal{E}_j\}|] \lesssim \frac{\epsilon^3}{B^4L^2M'} (\exp(-(a_{j+1} - \Delta)^2/2) + \exp(-a_{j+1}^2/2))$$

$$\lesssim \frac{\epsilon}{B^2} \frac{\epsilon^2}{B^2L^2} \frac{\exp(-a_{j+1}^2)}{a_{j+1}} \leq \frac{\epsilon}{B^2} \mathbf{Pr}[z \in \mathcal{E}_j].$$

Thus, suppose $z \in \mathcal{E}_j$, we have

$$|\mathbf{u} \cdot (\mathbf{g}_{\mathbf{w}}(z) - \mathbf{g}_{\mathbf{w}'}(z))| = \frac{1}{\mathbf{Pr}[z \in \mathcal{E}_j]} \left| \mathbf{E}_{(\mathbf{x},y) \sim \mathcal{D}}[y(\mathbf{u}^{\perp \mathbf{w}} - \mathbf{u}^{\perp \mathbf{w}'}) \cdot \mathbf{x}\mathbb{1}\{\mathbf{w} \cdot \mathbf{x} \in \mathcal{E}_j\}] \right|$$

$$+ \frac{1}{\mathbf{Pr}[z \in \mathcal{E}_j]} \left| \mathbf{E}_{(\mathbf{x},y) \sim \mathcal{D}}[y\mathbf{u}^{\perp \mathbf{w}} \cdot \mathbf{x}(\mathbb{1}\{\mathbf{w} \cdot \mathbf{x} \in \mathcal{E}_j\} - \mathbb{1}\{\mathbf{w}' \cdot \mathbf{x} \in \mathcal{E}_j\})] \right|$$

$$\leq \frac{1}{\mathbf{Pr}[z \in \mathcal{E}_j]} \mathbf{E}_{\mathbf{x} \sim \mathcal{D}_{\mathbf{x}}} \left[ B \left| \frac{\mathbf{u}^{\perp \mathbf{w}} - \mathbf{u}^{\perp \mathbf{w}'}}{\|\mathbf{u}^{\perp \mathbf{w}} - \mathbf{u}^{\perp \mathbf{w}'}\|_2} \cdot \mathbf{x} \right| \right] \mathbf{Pr}[z \in \mathcal{E}_j]\|\mathbf{u}^{\perp \mathbf{w}} - \mathbf{u}^{\perp \mathbf{w}'}\|_2$$

$$+ \frac{B}{\mathbf{Pr}[z \in \mathcal{E}_j]} \mathbf{E}_{\mathbf{x} \sim \mathcal{D}_{\mathbf{x}}}[|\mathbf{u}^{\perp \mathbf{w}} \cdot \mathbf{x}|] \mathbf{E}_{\mathbf{x} \sim \mathcal{D}_{\mathbf{x}}}[|\mathbb{1}\{\mathbf{w} \cdot \mathbf{x} \in \mathcal{E}_j\} - \mathbb{1}\{\mathbf{w}' \cdot \mathbf{x} \in \mathcal{E}_j\}|]$$

$$\lesssim \epsilon/B$$

Thus, this implies $\|\mathbf{g}_{\mathbf{w}}(z) - \mathbf{g}_{\mathbf{w}'}(z)\|_2 \lesssim \epsilon/B$. $\qquad\square$

Note that when $\|\mathbf{g}_{\mathbf{w}}(z) - \mathbf{g}_{\mathbf{w}'}(z)\|_2 \lesssim \epsilon/B$, we have (recall that in Equation (9) we showed $\|\mathbf{E}_{(\mathbf{x},y) \sim \mathcal{D}}[y\mathbf{x}^{\perp \mathbf{w}}\mathbb{1}\{\mathbf{w} \cdot \mathbf{x} \in \mathcal{E}_j\}]/\mathbf{Pr}[z \in \mathcal{E}_j]\|_2 \leq B$ hence by the definition of $\mathbf{g}_{\mathbf{w}}(z)$ we have

$\|\mathbf{g_w}(z)\|_2 \leq B$):

$$\forall \mathbf{u} \text{ s.t. } \|\mathbf{u}\|_2 = 1 : \quad |\mathbf{u}^\top \mathbf{g_{w'}}(z)\mathbf{g_{w'}}(z)^\top \mathbf{u} - \mathbf{u}^\top \mathbf{g_w}(z)\mathbf{g_w}(z)^\top \mathbf{u}|$$
$$= |\mathbf{u} \cdot (\mathbf{g_{w'}}(z) - \mathbf{g_w}(z))||\mathbf{u} \cdot (\mathbf{g_{w'}}(z) + \mathbf{g_w}(z))|$$
$$\leq \|\mathbf{g_{w'}}(z) - \mathbf{g_w}(z)\|_2 \|\mathbf{g_{w'}}(z) + \mathbf{g_w}(z)\|_2$$
$$\leq \frac{\epsilon}{CB}(2\|\mathbf{g_w}(z)\|_2 + \epsilon/B) \leq \epsilon,$$

indicating that

$$\|\mathbf{M_{w'}} - \mathbf{M_w}\|_2 = \sup_{\|\mathbf{u}\|_2} \mathop{\mathbf{E}}_{z \sim \mathcal{N}}[\mathbf{u}^\top(\mathbf{g_{w'}}(z)\mathbf{g_{w'}}(z)^\top - \mathbf{g_w}(z)\mathbf{g_w}(z)^\top)\mathbf{u}] \leq \epsilon.$$

Thus, constructing a $\tilde{O}(\epsilon^3/(B^4 L^2))$-cover $S$ on the unit sphere and requiring $\|(1/n)\sum_{i=1}^n \mathbf{M}_{\mathbf{w'}}^{(i)} - \mathbf{M_{w'}}\|_2 \leq \epsilon$ on all $\mathbf{w'} \in S$ suffices. Since a $\tilde{O}(\epsilon^3/(B^4 L^2))$-cover $S$ on the unit sphere contain $|S| = (O(\epsilon^3/(B^4 L^2)))^d$ vectors, applying a union bound we obtain that using $N = \Theta(d^2 B^{12} L^8/\epsilon^{10} \log(dBL/(\delta\epsilon)))$ samples, we guarantee that with probability at least $1 - \delta$, for any unit vector $\mathbf{w}$, it holds $\|\widehat{\mathbf{M}}_\mathbf{w} - \mathbf{M_w}\|_2 \leq \epsilon$. $\qquad\square$

# D  Proof of Main Theorem (Theorem 2.3)

We state and prove a more detailed version of the main theorem (Theorem 2.3) below:

**Theorem D.1** (Main Result). *Let $\epsilon > 0$. Fix parameters $B, L > 0$. Let $\mathcal{D}$ be a distribution over $(\mathbf{x}, y) \in \mathbb{R}^d \times \mathbb{R}$ with $\mathbf{x} \sim \mathcal{N}(\mathbf{0}, \mathbf{I}_d)$. Suppose there is a unit vector $\mathbf{w}^* \in \mathbb{R}^d$ and a monotone activation $\sigma \in \mathcal{H}_\epsilon(B, L)$ such that $\mathbf{E}_{(\mathbf{x},y)\sim\mathcal{D}}[(\sigma(\mathbf{w}^* \cdot \mathbf{x}) - y)^2] \leq \text{OPT}$. Then Algorithm 1 runs for at most $\text{poly}(B, L, 1/\epsilon)$ iterations, draws $\Theta(d^2 B^{12} L^8/\epsilon^{10} \log(dBL/\epsilon))$ samples, and returns a vector $\widehat{\mathbf{w}}$ and a Lipschitz and monotone activation $u : \mathbb{R} \to \mathbb{R}$, such that with probability at least $99\%$, it holds that $\mathbf{E}_{(\mathbf{x},y)\sim\mathcal{D}}[(u(\widehat{\mathbf{w}} \cdot \mathbf{x}) - y)^2] \leq O(\text{OPT}) + \epsilon$.*

*Proof of Theorem D.1.* We first show that if we appropriately choose the step size and the number of iterations in Algorithm 4, then for any initialization $\mathbf{w}^{(0)}$, Algorithm 4 with high probability returns a vector $\widehat{\mathbf{w}}$ with $\widehat{\theta} = \theta(\widehat{\mathbf{w}}, \mathbf{w}^*)$ so that $\sin\widehat{\theta} \leq O(\sqrt{\text{OPT}})/\|\mathrm{T}_{\cos\widehat{\theta}}\sigma'\|_{L_2}$ and $\widehat{\theta} \leq \bar{\theta} = \theta(\mathbf{w}^{(0)}, \mathbf{w}^*)$.

**Proposition D.2.** *Let $\epsilon > 0$. Fix parameters $B, L > 0$ and $\delta \in (0, 1)$. Let $\mathcal{D}$ be a distribution over $(\mathbf{x}, y) \in \mathbb{R}^d \times \mathbb{R}$ with $\mathbf{x} \sim \mathcal{N}(\mathbf{0}, \mathbf{I}_d)$. Suppose there is a unit vector $\mathbf{w}^* \in \mathbb{R}^d$ and an activation $\sigma \in \mathcal{H}_\epsilon(B, L)$ such that $\mathbf{E}_{(\mathbf{x},y)\sim\mathcal{D}}[(\sigma(\mathbf{w}^* \cdot \mathbf{x}) - y)^2] \leq \text{OPT}$. Then Algorithm 4, given $\bar{\theta}$, initialization vector $\mathbf{w}^{(0)}$ so that $\theta(\mathbf{w}^{(0)}, \mathbf{w}^*) \leq \bar{\theta}$ and $T \geq O(\log(L/\epsilon))$, with probability at least $1 - \delta$ returns a list of vectors $S^{\text{sol}}$ with size $|S^{\text{sol}}| = \text{poly}(1/\epsilon, L) \log(1/\delta)$ such that: there exists $\widehat{\mathbf{w}} \in S^{\text{sol}}$ so that: $\widehat{\theta} \leq \bar{\theta}$ and $\sin\widehat{\theta} \leq O(\sqrt{\text{OPT}})/\|\mathrm{T}_{\cos\widehat{\theta}}\sigma'\|_{L_2}$ where $\widehat{\theta} = \theta(\widehat{\mathbf{w}}, \mathbf{w}^*)$.*

*Proof of Proposition D.2.* In the proof, we denote the angle between $\mathbf{w}^{(t)}$ and $\mathbf{w}^*$ by $\theta_t = \theta(\mathbf{w}^{(t)}, \mathbf{w}^*)$. Furthermore, the algorithm uses the following parameters: $\phi_t = \bar{\theta}(1 - c^2/32)^t$ and $\eta_t = c\sin\phi_t/4$ where $c = 1/4 \leq \sqrt{2}/2$. Note that if $\epsilon \geq C\text{OPT}$, then we can run the algorithm with $\epsilon' = \epsilon/(2C)$ and assume that we have more noise of order $\text{OPT}' = 2\epsilon'$. In this case, the final error bound will be $C\text{OPT}' \leq \epsilon/2 \leq \text{OPT} + \epsilon$. So, without loss of generality, we can assume that $\epsilon \leq \text{OPT}$. According to Proposition C.1 as long as $\sin\theta_t \geq 40\sqrt{\text{OPT}}/\|\mathrm{T}_{\cos\theta_t}\sigma'\|_{L_2}$, with probability at least $1/2$, the vector $\mathbf{v}^{(t)}$ returned at Line (7) of Algorithm 4 satisfies $\mathbf{v}^{(t)} \cdot \mathbf{w}^* \leq -c\sin\theta_t$ and $\mathbf{v}^{(t)} \cdot \mathbf{w}^{(t)} = 0$. We denote as $\mathcal{P}_t$ the event that $\mathbf{v}^{(t)}$ negatively correlates with $\mathbf{w}^*$. We consider the following event

$$\mathcal{R}_t := \left\{ \sin\theta_t \geq \frac{C\sqrt{\text{OPT}}}{\|\mathrm{T}_{\cos\theta_t}\sigma'\|_{L_2}} \right\},$$

where $C > 0$ is an absolute constant.

First, we show that conditioning on the events $\mathcal{R}_t, \mathcal{P}_t$, for all $t \in T$, it holds that $\phi_t \geq \theta_t$.

**Claim D.3.** *Suppose the events $\mathcal{R}_t, \mathcal{P}_t, t \in [T_1]$, all hold for some $T_1 \geq 1$. Then for all $t \in [T_1]$, it holds that $\phi_t \geq \theta_t$.*

*Proof of Claim D.3.* We use induction for this proof. By assumption, we have that $\phi_0 \geq \theta_0$. Next, we assume that $\phi_t \geq \theta_t$. We need to show that $\phi_{t+1} \geq \theta_{t+1}$. We study the distance between $\mathbf{w}^{(t)}$ and $\mathbf{w}^*$ after one iteration from $t$ to $t+1$. Since $\mathbf{v}^{(t)}$ is orthogonal to $\mathbf{w}^{(t)}$, it must be $\|\mathbf{w}^{(t)} - \eta_t \mathbf{v}^{(t)}\|_2 \geq 1$, therefore, $\mathbf{w}^{(t+1)} = \mathrm{proj}_{\mathbb{B}}(\mathbf{w}^{(t)} - \eta_t \mathbf{v}^{(t)})$. By the non-expansiveness of the projection operator, we have

$$\|\mathbf{w}^{(t+1)} - \mathbf{w}^*\|_2^2 = \|\mathrm{proj}_{\mathbb{B}}(\mathbf{w}^{(t)} - \eta_t \mathbf{v}^{(t)}) - \mathbf{w}^*\|_2^2 \leq \|\mathbf{w}^{(t)} - \eta_t \mathbf{v}^{(t)} - \mathbf{w}^*\|_2^2$$
$$= \|\mathbf{w}^{(t)} - \mathbf{w}^*\|_2^2 + \eta_t^2 \|\mathbf{v}^{(t)}\|_2^2 - 2\eta_t \mathbf{v}^{(t)} \cdot (\mathbf{w}^{(t)} - \mathbf{w}^*)$$
$$= \|\mathbf{w}^{(t)} - \mathbf{w}^*\|_2^2 + \eta_t^2 + 2\eta_t \mathbf{v}^{(t)} \cdot \mathbf{w}^*. \tag{10}$$

Note that $\|\mathbf{w}^{(t)} - \mathbf{w}^*\|_2^2 - \|\mathbf{w}^{(t+1)} - \mathbf{w}^*\|_2^2 = 2(\cos\theta_{t+1} - \cos\theta_t)$ and using the identity about the sum of cosines, we have

$$4\sin\left(\frac{\theta_{t+1} - \theta_t}{2}\right)\sin\left(\frac{\theta_{t+1} + \theta_t}{2}\right) \leq \eta_t^2 + 2\eta_t \mathbf{v}^{(t)} \cdot \mathbf{w}^*.$$

First, consider the case where $2\theta_t \geq \phi_t \geq \theta_t$. From Proposition C.1, we have that $\mathbf{v}^{(t)} \cdot \mathbf{w}^* \leq -c\sin\theta_t$ where $c > 0$ is an absolute constant. Hence, since we chose $\eta_t = c\sin\phi_t/4$, it holds $\eta_t^2 + 2\eta_t \mathbf{v}^{(t)} \cdot \mathbf{w}^* \leq -c^2 \sin\phi_t \sin\theta_t/4$. Therefore, in this case, $\theta_{t+1} \leq \theta_t$ hence $\sin((\theta_{t+1} + \theta_t)/2) \leq \sin\theta_t$, and we have that

$$16\sin\left(\frac{\theta_t - \theta_{t+1}}{2}\right) \geq c^2 \sin\phi_t .$$

Using the inequality $x/4 \leq \sin x \leq x$ for $x \in (0, \pi/2)$, we get that

$$\theta_{t+1} \leq \theta_t(1 - c^2/32) ,$$

and using that $\phi_{t+1} = \phi_t(1 - c^2/32)$, we have that $\theta_{t+1} \leq \phi_{t+1}$.

Consider the remaining case where $\phi_t \geq 2\theta_t$. In this case, if $\theta_{t+1} \leq \theta_t$, then $\theta_{t+1} \leq \phi_{t+1}$ so we need to consider the case where $\theta_{t+1} \geq \theta_t$. We need to bound the maximum increase of $\theta_{t+1}$. Applying the triangle inequality and the non-expansiveness of the projection operator, it holds

$$2\sin(\theta_{t+1}/2) = \|\mathbf{w}^{(t+1)} - \mathbf{w}^*\|_2 = \|\mathrm{proj}_{\mathbb{B}}(\mathbf{w}^{(t)} - \eta_t \mathbf{v}^{(t)}(\mathbf{w}^{(t)})) - \mathbf{w}^*\|_2 \leq \|\mathbf{w}^{(t)} - \eta_t \mathbf{v}^{(t)} - \mathbf{w}^*\|_2$$
$$\leq \|\mathbf{w}^{(t)} - \mathbf{w}^*\|_2 + \eta_t \|\mathbf{v}^{(t)}\|_2 = 2\sin(\theta_t/2) + c\sin\phi_t/4 .$$

From the assumption, we have $\theta_t \leq \phi_t/2$, therefore, choosing $c \leq 1/4$ and since $\sin(x) \leq x$ for $x \in (0, \pi/2)$, we have that

$$\sin(\theta_{t+1}/2) \leq \sin(\phi_t/4) + c\sin\phi_t/8 \leq 9\phi_t/32 .$$

Therefore, since $(5/8)x \leq \sin x$ when $x \in (0, \pi/2)$ we have $\sin(\theta_{t+1}/2) \geq (5/16)\theta_{t+1}$ and thus, $\theta_{t+1} \leq (9/10)\phi_t \leq (1 - c^2/32)\phi_t \leq \phi_{t+1}$. This completes the proof. $\square$

Next, we condition on the event that all $\mathcal{P}_t$ are satisfied for $t \in [T]$. According to Claim D.3, as long as $\mathcal{R}_t$ is satisfied, we have $\phi_t \geq \theta_t$. Assume $\mathcal{R}_t$ is satisfied for all $t \in [T]$. If $T \geq C' \log(L/\epsilon)$ for a sufficiently large constant $C'$, then $\phi_T \leq \sqrt{\epsilon}/L$. This would imply $\theta_T \leq \sqrt{\epsilon}/L$. If $\mathcal{R}_T$ was satisfied, $\sin\theta_T \geq C\sqrt{\mathrm{OPT}}/\|\mathrm{T}_{\cos\theta_T}\sigma'\|_{L_2}$. So $\theta_T \geq \sin\theta_T \geq C\sqrt{\mathrm{OPT}}/L$ (since $\|\mathrm{T}_{\cos\theta_T}\sigma'\|_{L_2} \leq \|\sigma'\|_{L_2} \leq L$, by Fact A.2(f)). Then $\sqrt{\epsilon}/L \geq C\sqrt{\mathrm{OPT}}/L$, which means $\sqrt{\epsilon} \geq C\sqrt{\mathrm{OPT}}$. If $\epsilon \leq \mathrm{OPT}$, this is a contradiction for $C > 1$. This means that our assumption that $\mathcal{R}_t$ are satisfied for all $t \in [T]$ must be false. Therefore, there must exist some $T_1 \in [T]$ (we take $T_1$ to be the smallest one) such that $\mathcal{R}_{T_1}$ is not satisfied. For all $t < T_1$, $\mathcal{R}_t$ was satisfied (otherwise $T_1$ would be smaller), and thus $\phi_t \geq \theta_t$ for $t < T_1$. At $t = T_1$, $\mathcal{R}_{T_1}$ is false, meaning $\sin\theta_{T_1} < C\sqrt{\mathrm{OPT}}/\|\mathrm{T}_{\cos\theta_{T_1}}\sigma'\|_{L_2}$, and we also have $\theta_{T_1} \leq \phi_{T_1} \leq \bar{\theta}$. This gives us the desired vector $\widehat{\mathbf{w}} = \mathbf{w}^{(T_1)}$ such that $\widehat{\theta} \leq C\sqrt{\mathrm{OPT}}/\|\mathrm{T}_{\cos\widehat{\theta}}\sigma'\|_{L_2}$.

To conclude, we need to bound the probability that all $\mathcal{P}_t$ (correct direction choices) are satisfied. The events $\mathcal{P}_t$ are independent, and each occurs with probability at least $1/2$. The probability of $T_1$ such events occurring is at least $(1/2)^{T_1}$. Since $T_1 \leq T = O(\log(L/\epsilon))$, this probability is bounded below by $\delta' = (1/2)^T = \mathrm{poly}(\epsilon, 1/L)$. If we rerun the algorithm $K = O((1/\delta')\log(1/\delta))$ times ( Line 3 of Algorithm 4), by standard Chernoff bounds, with probability at least $1 - \delta$, there will be at least one run where all $\mathcal{P}_t$ are satisfied for $t \in [T_1]$. This completes the proof of Proposition D.2. $\square$

In Lemma 2.5, we showed that the initialization algorithm (Algorithm 2) uses $\tilde{O}(d\log(B)/\epsilon^2)$ samples and returns a list of vectors $S^{\mathrm{ini}}$, $|S^{\mathrm{ini}}| \leq B/\sqrt{\epsilon}$ that contains a vector $\mathbf{w}^{(0)}$ such that $\theta(\mathbf{w}^{(0)}, \mathbf{w}^*) \leq 1/M$ and $M$ is the threshold we can truncate $\sigma$ such that $\mathbf{E}_{z\sim\mathcal{N}}[(\sigma(z) - \sigma(M))^2\mathbb{1}\{|z| \geq M\}] \leq C(\mathrm{OPT} + \epsilon)$, i.e., we can truncate $\sigma$ at $M$ and the overall error is increased by $C(\mathrm{OPT} + \epsilon)$. By Fact A.10 (with the absolute constant $c = 2$ in Fact A.10), this initialized vector $\mathbf{w}^{(0)}$ ensures that for any unit vector $\mathbf{w}^{(t)}$ such that $\theta_t := \theta(\mathbf{w}^{(t)}, \mathbf{w}^*) \leq 2\theta(\mathbf{w}^{(0)}, \mathbf{w}^*)$, it holds

$$\mathop{\mathbf{E}}_{(\mathbf{x},y)\sim\mathcal{D}}[(\sigma(\mathbf{w}^{(t)}\cdot\mathbf{x}) - y)^2] \leq C\mathrm{OPT} + \sin^2\theta_t\|\mathrm{T}_{\cos\theta_t}\sigma'\|_{L_2}^2. \tag{11}$$

Therefore, any unit vector $\mathbf{w}^{(t)}$ such that $\theta_t \leq 2\theta(\mathbf{w}^{(0)}, \mathbf{w}^*)$ and $\sin^2\theta_t\|\mathrm{T}_{\cos\theta_t}\sigma'\|_{L_2}^2 \leq C\mathrm{OPT}$ is a constant factor approximate solution (i.e., it holds $\mathbf{E}_{(\mathbf{x},y)\sim\mathcal{D}}[(\sigma(\mathbf{w}^{(t)}\cdot\mathbf{x}) - y)^2] \leq C\mathrm{OPT} + \epsilon$). Since we are iterating through this list $S^{\mathrm{ini}}$, it is guaranteed we will encounter the correct $\mathbf{w}^{(0)}$. If $\mathbf{w}^{(0)}$ is an approximate solution, it will be tested and output by our testing subroutine (Algorithm 5). Now assume in the following that $\mathbf{w}^{(0)}$ is not a target solution.

It remains to show that we can choose the correct stepsize in Algorithm 4. If we choose $\bar{\theta}$ from the list $\Theta = \{\epsilon/L, \ldots, k\epsilon/L\}$ for $k = (\pi/2)L/\epsilon$, it is guaranteed that for the correct $\mathbf{w}^{(0)}$ in the initialization list, there exists an initial stepsize $\bar{\theta} \in \Theta$ so that $\theta(\mathbf{w}^{(0)}, \mathbf{w}^*) \in (\bar{\theta} - \epsilon/L, \bar{\theta})$ (see (5) of Algorithm 1). Our claim is that we are guaranteed to have $\bar{\theta} \leq 2\theta(\mathbf{w}^{(0)}, \mathbf{w}^*)$. That means according to Proposition D.2, setting $\delta = 0.01$, with probability at least 99% we have that there exist $\widehat{\mathbf{w}}$ in the list $S^{\mathrm{sol}}$, returned by Algorithm 4, that satisfies that $\widehat{\theta} = \theta(\widehat{\mathbf{w}}, \mathbf{w}^*) \leq \bar{\theta} \leq 2\theta(\mathbf{w}^{(0)}, \mathbf{w}^*)$ and $\sin\widehat{\theta} \leq O(\sqrt{\mathrm{OPT}})/\|\mathrm{T}_{\cos\widehat{\theta}}\sigma'\|_{L_2}$, indicating that $\mathbf{E}_{(\mathbf{x},y)\sim\mathcal{D}}[(\sigma(\widehat{\mathbf{w}}\cdot\mathbf{x}) - y)^2] \leq C\mathrm{OPT} + \epsilon$ by (11). Now we prove the claim that $\bar{\theta} \leq 2\theta(\mathbf{w}^{(0)}, \mathbf{w}^*)$. To show this, it suffices to prove that $\theta(\mathbf{w}^{(0)}, \mathbf{w}^*) \geq \epsilon/L$ because we choose $\bar{\theta} = k\epsilon/L$ for $k \in [(\pi/2)L/\epsilon]$.

**Claim D.4.** *Suppose $\sigma(\mathbf{w}^{(0)}\cdot\mathbf{x})$ is not a constant factor approximate solution. Then, it must hold that $\theta(\mathbf{w}^{(0)}, \mathbf{w}^*) \geq \epsilon/L$.*

*Proof.* Assuming that $\theta_0 \leq \epsilon/L$, we have

$$\mathop{\mathbf{E}}_{(\mathbf{x},y)\sim\mathcal{D}}[(\sigma(\mathbf{w}^{(0)}\cdot\mathbf{x}) - y)^2] \leq 2\mathop{\mathbf{E}}_{\mathbf{x}\sim\mathcal{D}_{\mathbf{x}}}[(\sigma(\mathbf{w}^{(0)}\cdot\mathbf{x}) - \sigma(\mathbf{w}^*\cdot\mathbf{x}))^2] + 2\mathop{\mathbf{E}}_{(\mathbf{x},y)\sim\mathcal{D}}[(\sigma(\mathbf{w}^*\cdot\mathbf{x}) - y)^2]$$

$$\leq 4\left(\mathop{\mathbf{E}}_{z\sim\mathcal{N}}[\sigma(z)^2] - \mathop{\mathbf{E}}_{z_1,z_2\sim\mathcal{N}}[\sigma(\cos\theta_0 z_1 + \sin\theta_0 z_2)\sigma(z_1)]\right) + 2\mathrm{OPT}.$$

By the definition of Ornstein–Uhlenbeck-semi-group and using the tower rule of expectation, we have that $\mathbf{E}_{z_1,z_2\sim\mathcal{N}}[\sigma(\cos\theta_0 z_1 + \sin\theta_0 z_2)\sigma(z_1)] = \mathbf{E}_{z_1\sim\mathcal{N}}[\sigma(z_1)\mathrm{T}_{\cos\theta_0}\sigma(z_1)]$, which yields that the error can be bounded from above by

$$\|\sigma(\mathbf{w}^{(0)}\cdot\mathbf{x}) - y\|_{L_2}^2 \lesssim \left(\mathop{\mathbf{E}}_{z\sim\mathcal{N}}[\sigma(z)^2] - \mathop{\mathbf{E}}_{z\sim\mathcal{N}}[\mathrm{T}_{\cos\theta_0}\sigma(z)\sigma(z)]\right) + \mathrm{OPT}.$$

Note that by the fundamental formula of calculus, the term in the parenthesis above can be written as

$$\mathop{\mathbf{E}}_{z\sim\mathcal{N}}[\sigma(z)^2] - \mathop{\mathbf{E}}_{z\sim\mathcal{N}}[\mathrm{T}_{\cos\theta_0}\sigma(z)\sigma(z)] = \mathop{\mathbf{E}}_{z\sim\mathcal{N}}[\sigma(z)(\sigma(z) - \mathrm{T}_{\cos\theta_0}\sigma(z))] = \mathop{\mathbf{E}}_{z\sim\mathcal{N}}\left[\int_{\cos\theta_0}^1 \sigma(z)\frac{\mathrm{d}}{\mathrm{d}\rho}\mathrm{T}_\rho\sigma(z)\,\mathrm{d}\rho\right]$$

$$= \int_{\cos\theta_0}^1 \mathop{\mathbf{E}}_{z\sim\mathcal{N}}\left[\sigma(z)\frac{\mathrm{d}}{\mathrm{d}\rho}\mathrm{T}_\rho\sigma(z)\,\mathrm{d}\rho\right],$$

where in the last equation, we used Fubini's theorem. Now applying Fact A.4, we have $\mathrm{d}\mathrm{T}_\rho\sigma(z)/\mathrm{d}\rho = (1/\rho)L\mathrm{T}_\rho\sigma(z)$ (Fact A.4 part 1), and using Fact A.4 part 2 we further obtain

$$\mathop{\mathbf{E}}_{z\sim\mathcal{N}}[\sigma(z)(\mathrm{d}\mathrm{T}_\rho\sigma(z)/\mathrm{d}\rho)] = (1/\rho)\mathop{\mathbf{E}}_{z\sim\mathcal{N}}[\sigma(z)L\mathrm{T}_\rho\sigma(z)] = (1/\rho)\mathop{\mathbf{E}}_{z\sim\mathcal{N}}[\sigma'(z)(\mathrm{T}_\rho\sigma(z))'].$$

Bringing in Fact A.2 part $(g)$, it finally yields

$$\mathop{\mathbf{E}}_{z\sim\mathcal{N}}[\sigma(z)^2] - \mathop{\mathbf{E}}_{z\sim\mathcal{N}}[\mathrm{T}_{\cos\theta_0}\sigma(z)\sigma(z)] = \int_{\cos\theta_0}^1 \mathop{\mathbf{E}}_{z\sim\mathcal{N}}[\sigma'(z)\mathrm{T}_\rho\sigma'(z)]\,\mathrm{d}\rho \leq \int_{\cos\theta_0}^1 \|\sigma'(z)\|_2\|\mathrm{T}_\rho\sigma'(z)\|_2\,\mathrm{d}\rho$$

$$\leq (1 - \cos\theta_0)L^2 = 2\sin^2(\theta_0/2)L^2.$$

Therefore, if $\sin\theta_0 \leq \epsilon/L$, we have $\|\sigma(\mathbf{w}^{(0)}\cdot\mathbf{x}) - y\|_{L_2}^2 \lesssim \mathrm{OPT} + \epsilon$, contradicting the assumption that $\mathbf{w}^{(0)}$ is not a constant-factor approximate solution. $\qquad\square$

Next, we show that given all the constructed candidate solutions, Algorithm 5 with high probability returns an activation and direction pair that achieves $O(\mathrm{OPT}) + \epsilon$ error. The proof of Lemma D.5 can be found in Appendix D.1.

**Lemma D.5** (Learning the Predictor and Testing). *Algorithm 5 given $n = \mathrm{poly}(B, L, 1/\epsilon)$ samples and a set $S^{\mathrm{sol}}$ of $\mathrm{poly}(B, L, 1/\epsilon)$ vectors, with probability at least $99\%$ returns a solution pair $(\widehat{u}_{\widehat{\mathbf{w}}}, \widehat{\mathbf{w}})$, with $\widehat{u}_{\widehat{\mathbf{w}}}$ being Lipschitz and monotone, and $\widehat{\mathbf{w}} \in S^{\mathrm{sol}}$, such that*

$$\mathop{\mathbf{E}}_{(\mathbf{x},y)\sim\mathcal{D}}[(\widehat{u}_{\widehat{\mathbf{w}}}(\widehat{\mathbf{w}} \cdot \mathbf{x}) - y)^2] \le C \min_{\mathbf{w}\in S^{\mathrm{sol}}} \mathop{\mathbf{E}}_{(\mathbf{x},y)\sim\mathcal{D}}[(\sigma(\mathbf{w} \cdot \mathbf{x}) - y)^2] + \epsilon \;,$$

*for some universal constant $C$.*

---

**Algorithm 5** Testing

---

1: **Input:** Parameters $B$, $L$, $\epsilon$; Data access $(\mathbf{x}, y) \sim \mathcal{D}$; $S^{\mathrm{sol}}$, empty set $S$
2: Draw $n$ samples $\{(\mathbf{x}^{(i)}, y^{(i)})\}_{i=1}^n$ and construct the empirical distribution $\widehat{\mathcal{D}}_n$.
3: **for** $\mathbf{w} \in S^{\mathrm{sol}}$ **do**
4:     Find $\widehat{u}_{\mathbf{w}} = \mathrm{argmin}_{u\in\mathcal{H}(B,L)} \mathbf{E}_{(\mathbf{x},y)\sim\widehat{\mathcal{D}}_n}[(u(\mathbf{w} \cdot \mathbf{x}) - y)^2]$.
5:     $S \leftarrow S \cup \{(u_{\mathbf{w}}, \mathbf{w})\}$
6: **Return:** $(\widehat{u}_{\widehat{\mathbf{w}}}, \widehat{\mathbf{w}}) = \mathrm{argmin}\{(u_{\mathbf{w}}, \mathbf{w}) \in S : \mathbf{E}_{(\mathbf{x},y)\sim\widehat{\mathcal{D}}_n}[(u_{\mathbf{w}}(\mathbf{w} \cdot \mathbf{x}) - y)^2]\}$.

---

Finally, using Lemma D.5, we know that drawing at most $n = \Theta(\log(BL/\epsilon)B^3 L/\epsilon^{3/2})$ new samples, with probability at least $99\%$, the testing algorithm (Algorithm 5) returns a solution pair $(\widehat{u}_{\widehat{\mathbf{w}}}, \widehat{\mathbf{w}})$ where $\widehat{u}_{\widehat{\mathbf{w}}} \in \mathcal{H}(B, L)$ and $\widehat{\mathbf{w}} \in S^{\mathrm{sol}}$ such that $\mathbf{E}_{(\mathbf{x},y)\sim\mathcal{D}}[(\widehat{u}_{\widehat{\mathbf{w}}}(\widehat{\mathbf{w}} \cdot \mathbf{x}) - y)^2] \le C\mathrm{OPT} + \epsilon$. This completes the proof of Theorem D.1. □

## D.1 Proof of the Testing Lemma (Lemma D.5)

Algorithm 1 generates a list of possible parameters $S^{\mathrm{sol}} = \{\mathbf{w}^{(i)}\}_{i=1}^m$, where $m = \mathrm{poly}(1/\epsilon, B, L)$, and we know that there exists a vector $\widehat{\mathbf{w}} \in S^{\mathrm{sol}}$ such that $\sin\theta \le \sqrt{\mathrm{OPT}}/\|\mathrm{T}_{\cos(\theta)}\sigma'\|_{L_2}$, where $\theta = \theta(\widehat{\mathbf{w}}, \mathbf{w}^*)$. To complete the task of learning SIMs, we need to: (1) find an activation $u \in \mathcal{H}(B, L)$ that is close to the target activation $\sigma$; (2) find the target vector $\widehat{\mathbf{w}} \in S^{\mathrm{sol}}$.

First, we note that given any vector $\mathbf{w}$ and $n$ samples $\{(\mathbf{x}^{(i)}, y^{(i)})\}_{i=1}^n$, there exists an efficient algorithm that computes a best fitting monotone and $\beta$-Lipschitz function on the sample set $\{(\mathbf{x}^{(i)}, y^{(i)})\}_{i=1}^n$, via solving the following constrained optimization problem:

$$\min_{v_i, i\in[n]} \sum_{i=1}^n (v_i - y^{(i)})^2 \tag{Iso}$$
$$\text{s.t. } 0 \le v_{i+1} - v_i \le \beta(\mathbf{w} \cdot \mathbf{x}^{(i+1)} - \mathbf{w} \cdot \mathbf{x}^{(i)}).$$

We remark that $\mathbf{x}^{(i)}$'s are sorted so that $\mathbf{w} \cdot \mathbf{x}^{(i)}$'s are in increasing order.

We use the following fact:

**Fact D.6** (Proposition 1 [HTY25]). *Given a sample set $\{(\mathbf{x}^{(i)}, y^{(i)})\}_{i=1}^n$, there exists an algorithm that exactly solves (Iso) in $O(n\log^2(n))$ time.*

Observe that given the solution $\{v_i\}_{i=1}^n$ of (Iso), we can construct a function $\widehat{u}_{\mathbf{w}}(z)$ by linearly interpolating the points $\{(z_i, v_i)\}_{i=1}^n$ where $z_i = \mathbf{w} \cdot \mathbf{x}_i$. Then, the function $\widehat{u}_{\mathbf{w}}(z)$ is guaranteed to be a monotone and $\beta$-Lipschitz function. In the following claim, we show that the function class $\sigma \in \mathcal{H}(B, L)$ is covered by Lipschitz-continuous functions.

**Claim D.7.** *It is without loss of generality to assume that $\tilde{\sigma} \in \mathcal{H}_\epsilon(B, L)$ is $BL/\sqrt{\epsilon}$-Lipschitz.*

*Proof.* Let $\sigma \in \mathcal{H}(B, L)$ such that $\|\sigma - \tilde{\sigma}\|_{L_2}^2 \le \epsilon$. By Fact A.5, we have that for any $\rho \in (0, 1)$ it holds $\|\mathrm{T}_\rho\sigma - \sigma\|_{L_2}^2 \le (1 - \rho^2)\|\sigma'\|_{L_2}^2 \le (1 - \rho^2)L^2$. Therefore, choosing $\rho^2 = 1 - \epsilon/L^2$ we have that for any function $\sigma \in \mathcal{H}(B, L)$, there exists a function $\mathrm{T}_\rho\sigma \in \mathcal{H}(B, L)$ such that

$\|\mathrm{T}_\rho\sigma - \sigma\|_{L_2}^2 \leq \epsilon$ and hence $\|\mathrm{T}_\rho\sigma - \tilde{\sigma}\|_{L_2}^2 \leq 2\epsilon$. Furthermore, the function $\mathrm{T}_\rho\sigma(z)$ is $BL/\sqrt{\epsilon}$-Lipschitz since according to Fact A.2 part (c) we have $\|(\mathrm{T}_\rho\sigma)'\|_{L_\infty} \leq \|\sigma\|_{L_\infty}/\sqrt{1-\rho^2} \leq BL/\sqrt{\epsilon}$. Therefore, the function class $\mathcal{H}_\epsilon(B,L)$ is covered by $BL/\sqrt{\epsilon}$-Lipschitz functions and hence it is without loss of generality to assume that $\sigma \in \mathcal{H}_\epsilon(B,L)$ is $BL/\sqrt{\epsilon}$-Lipschitz. $\qquad\square$

We are now ready to prove the sample complexity and the correctness of the testing algorithm. We restate and prove Lemma D.5.

**Lemma D.5** (Learning the Predictor and Testing). *Algorithm 5 given $n = \mathrm{poly}(B,L,1/\epsilon)$ samples and a set $S^{\mathrm{sol}}$ of $\mathrm{poly}(B,L,1/\epsilon)$ vectors, with probability at least 99% returns a solution pair $(\widehat{u}_{\widehat{\mathbf{w}}}, \widehat{\mathbf{w}})$, with $\widehat{u}_{\widehat{\mathbf{w}}}$ being Lipschitz and monotone, and $\widehat{\mathbf{w}} \in S^{\mathrm{sol}}$, such that*

$$\mathop{\mathbf{E}}_{(\mathbf{x},y)\sim\mathcal{D}}[(\widehat{u}_{\widehat{\mathbf{w}}}(\widehat{\mathbf{w}}\cdot\mathbf{x}) - y)^2] \leq C \min_{\mathbf{w}\in S^{\mathrm{sol}}} \mathop{\mathbf{E}}_{(\mathbf{x},y)\sim\mathcal{D}}[(\sigma(\mathbf{w}\cdot\mathbf{x}) - y)^2] + \epsilon \,,$$

*for some universal constant $C$.*

*Proof.* Let $\beta = BL/\sqrt{\epsilon}$ (if we know that the target activation $\sigma$ is $b$-Lipschitz, let $\beta = b$). Let $\{(\mathbf{x}^{(i)}, y^{(i)})\}_{i=1}^n$ be a set of $n$ samples and let $\widehat{u}_{\mathbf{w}}(z)$ be a solution of (Iso) (via linear interpolation), i.e.,

$$\widehat{u}_{\mathbf{w}}(z) \in \mathop{\mathrm{argmin}}_{u:\beta-\mathrm{Lipschictz},u'\geq 0} (1/n)\sum_{i=1}^n (u(\mathbf{w}\cdot\mathbf{x}^{(i)}) - y^{(i)})^2.$$

Note that in Claim D.7 we showed that all the functions in $\mathcal{H}_\epsilon(B,L)$ are $\beta$-Lipschitz functions, hence we have

$$(1/n)\sum_{i=1}^n (\widehat{u}_{\mathbf{w}}(\mathbf{w}\cdot\mathbf{x}^{(i)}) - y^{(i)})^2 \leq \mathop{\mathrm{argmin}}_{u\in\mathcal{H}(B,L),u'\geq 0} (1/n)\sum_{i=1}^n (u(\mathbf{w}\cdot\mathbf{x}^{(i)}) - y^{(i)})^2.$$

Let us denote $\varphi_{u,\mathbf{w}}(\mathbf{x}) := u(\mathbf{w}\cdot\mathbf{x})$ and let $\mathcal{U} := \{\varphi_{u,\mathbf{w}} : u \text{ is } \beta\text{-Lipschitz}, u'\geq 0, \mathbf{w}\in S^{\mathrm{sol}}\}$ be the family of all such $\varphi_{u,\mathbf{w}}$. Let $\mathcal{L}(\varphi_{u,\mathbf{w}}) := \mathbf{E}_{(\mathbf{x},y)\sim\mathcal{D}}[(\varphi_{u,\mathbf{w}}(\mathbf{x}) - y)^2]$. Denote the empirical distribution on $\{(\mathbf{x}^{(i)}, y^{(i)})\}_{i=1}^n$ by $\widehat{\mathcal{D}}_n$, we define $\widehat{\mathcal{L}}(\varphi_{u,\mathbf{w}}) = \mathbf{E}_{(\mathbf{x},y)\sim\widehat{\mathcal{D}}_n}[(\varphi_{u,\mathbf{w}}(\mathbf{x}) - y)^2]$. Furthermore, let

$$\widehat{\varphi}^* := \mathop{\mathrm{argmin}}_{\varphi\in\mathcal{U}} \widehat{\mathcal{L}}(\varphi), \ \mathcal{L}^* := \min_{\varphi\in\mathcal{U}} \mathcal{L}(\varphi).$$

Then, since we know there exist an activation $\sigma \in \mathcal{H}_\epsilon(B,L)$ and a vector $\widehat{\mathbf{w}} \in S^{\mathrm{sol}}$ such that $\mathbf{E}_{(\mathbf{x},y)\sim\mathcal{D}}[(\sigma(\widehat{\mathbf{w}}\cdot\mathbf{x}) - y)^2] \leq C\mathrm{OPT} + \epsilon$, it holds that $\mathcal{L}^* \leq C\mathrm{OPT} + \epsilon$. Furthermore, by definition we have

$$\widehat{\varphi}^* = \mathop{\mathrm{argmin}}_{\varphi\in\mathcal{U}} \mathop{\mathbf{E}}_{(\mathbf{x},y)\sim\widehat{\mathcal{D}}_n}[(\varphi(\mathbf{x}) - y)^2] = \mathop{\mathrm{argmin}}_{u:\beta-\mathrm{Lipschitz},u'\geq 0,\mathbf{w}\in S^{\mathrm{sol}}} \mathop{\mathbf{E}}_{(\mathbf{x},y)\sim\widehat{\mathcal{D}}_n}[(y - u(\mathbf{w}\cdot\mathbf{x}))^2],$$

indicating that $\widehat{\varphi}^*$ is a solution of the problem (Iso) with respect to some vector $\widehat{\mathbf{w}} \in S^{\mathrm{sol}}$, i.e., $\widehat{\varphi}^*(\mathbf{x}) = \varphi_{\widehat{u}_{\widehat{\mathbf{w}}},\widehat{\mathbf{w}}}(\mathbf{x}) = \widehat{u}_{\widehat{\mathbf{w}}}(\widehat{\mathbf{w}}\cdot\mathbf{x})$ for some $\beta$-Lipschitz function $\widehat{u}_{\widehat{\mathbf{w}}}$ and $\widehat{\mathbf{w}} \in S^{\mathrm{sol}}$.

We use the following fact to show that $\widehat{\mathcal{L}}(\varphi_{u,\mathbf{w}})$ are close to $\mathcal{L}(\varphi_{u,\mathbf{w}})$ for all $\varphi_{u,\mathbf{w}} \in \mathcal{U}$:

**Fact D.8** (Theorem 1, [SST10]). *Suppose there exists a constant $b > 0$ such that for any $\varphi \in \mathcal{U}$, and any $(\mathbf{x},y) \sim \mathcal{D}$ it holds $(\varphi(\mathbf{x}) - y)^2 \leq b$. Let $\widehat{\mathcal{R}}_n(\mathcal{U})$ be the empirical Rademacher complexity of function class $\mathcal{U}$ with respect to some sample set $(\mathbf{x}^{(1)}, \ldots, \mathbf{x}^{(n)})$ and let $\mathcal{R}_n(\mathcal{U}) = \sup_{(\mathbf{x}^{(1)},\ldots,\mathbf{x}^{(n)})} \widehat{\mathcal{R}}_n(\mathcal{U})$. We have that with probability at least $1 - \delta$ over the random sample set of size $n$, for any $\varphi \in \mathcal{U}$, it holds for some universal constant $C' > 0$ that*

$$\mathcal{L}(\varphi) \leq \widehat{\mathcal{L}}(\varphi) + C'\left(\sqrt{\widehat{\mathcal{L}}(\varphi)}\left(\log^{1.5}(n)\mathcal{R}_n(\mathcal{U}) + \sqrt{\frac{b\log(1/\delta)}{n}}\right) + \log^3(n)\mathcal{R}_n^2(\mathcal{U}) + \frac{b\log(1/\delta)}{n}\right),$$

*and*

$$\mathcal{L}(\widehat{\varphi}^*) \leq \mathcal{L}^* + C'\left(\sqrt{\mathcal{L}^*}\left(\log^{1.5}(n)\mathcal{R}_n(\mathcal{U}) + \sqrt{\frac{b\log(1/\delta)}{n}}\right) + \log^3(n)\mathcal{R}_n^2(\mathcal{U}) + \frac{b\log(1/\delta)}{n}\right).$$

Note first that since $u \in \mathcal{U}$ by definition we have $|u| \leq B$ and since $|y| \leq B$ as well, it holds $|\varphi(\mathbf{x})| \leq B$ and $(\varphi(\mathbf{x}) - y)^2 \lesssim B^2$ for any $\varphi \in \mathcal{U}$.

Fact D.8 implies that if $n$ is large enough such that $\mathcal{R}_n(\mathcal{U}) \leq \sqrt{\epsilon}/\log^{3/2}(n)$ and $B^2 \log(1/\delta)/n \leq \epsilon$, then we are guaranteed that: (1) for any activation $u$ and any vector $\mathbf{w} \in S^{\mathrm{sol}}$, it holds $\mathbf{E}_{(\mathbf{x},y)\sim\mathcal{D}}[(u(\mathbf{w} \cdot \mathbf{x}) - y)^2] \leq \mathbf{E}_{(\mathbf{x},y)\sim\widehat{\mathcal{D}}_n}[(u(\mathbf{w} \cdot \mathbf{x}) - y)^2]$; (2) for the solution pair $(\widehat{u}_{\widehat{\mathbf{w}}}, \widehat{\mathbf{w}})$ that achieves the minimal empirical loss among all the vectors $\mathbf{w}$ from $S^{\mathrm{sol}}$ and all $\beta$-Lipschitz activations, i.e., $\widehat{\varphi}^* = \varphi_{\widehat{u}_{\widehat{\mathbf{w}}},\widehat{\mathbf{w}}} \in \operatorname{argmin}_{\varphi\in\mathcal{U}} \mathbf{E}_{(\mathbf{x},y)\sim\widehat{\mathcal{D}}_n}[(\varphi(\mathbf{x}) - y)^2]$, we have $\mathcal{L}(\varphi_{\widehat{u}_{\widehat{\mathbf{w}}},\widehat{\mathbf{w}}}) = \mathbf{E}_{(\mathbf{x},y)\sim\mathcal{D}}[(u_{\widehat{\mathbf{w}}}(\widehat{\mathbf{w}} \cdot \mathbf{x}) - y)^2] \leq (C' + 1)\mathcal{L}^* \leq CC'\mathrm{OPT} + \epsilon$. Therefore, by solving (Iso) and finding the besting fitting activation $u_{\mathbf{w}}$ for each $\mathbf{w} \in S^{\mathrm{sol}}$ and outputting the solution pair $(u_{\widehat{\mathbf{w}}}, \widehat{\mathbf{w}})$ with the smallest empirical error, we are ensured that $u_{\widehat{\mathbf{w}}}(\widehat{\mathbf{w}} \cdot \mathbf{x})$ is a constant factor approximate solution such that $\mathbf{E}_{(\mathbf{x},y)\sim\mathcal{D}}[(u_{\widehat{\mathbf{w}}}(\widehat{\mathbf{w}} \cdot \mathbf{x}) - y)^2] \leq C\mathrm{OPT} + \epsilon$.

Thus, it remains to bound the Rademacher complexity of the function class $\mathcal{U}$ and choose $n$ such that $\mathcal{R}_n(\mathcal{U}) \leq \sqrt{\epsilon}/\log^{3/2}(n)$ and $B^2 \log(1/\delta)/n \leq \epsilon$. To this aim, we use the following fact:

**Fact D.9** (Lemma A.3, [SST10]). *For any function class $\mathcal{U}$, let $N_2(\epsilon, \mathcal{U}, n)$ be the $\epsilon$-cover of $\mathcal{U}$ with respect to $\ell_2$ norm on sample set $(\mathbf{x}^{(1)}, \ldots, \mathbf{x}^{(n)})$. Let $\widehat{\mathcal{D}}_n$ be the empirical distribution on $(\mathbf{x}^{(1)}, \ldots, \mathbf{x}^{(n)})$. Then,*

$$\widehat{\mathcal{R}}_n(\mathcal{U}) \leq \inf_{\alpha>0}\left\{4\alpha + 10\int_\alpha^{\sup_{\varphi\in\mathcal{U}}\sqrt{\mathbf{E}_{\mathbf{x}\sim\widehat{\mathcal{D}}_n}[\varphi^2(\mathbf{x})]}} \sqrt{\frac{\log(|N_2(\epsilon,\mathcal{U},n)|)}{n}}\,\mathrm{d}\epsilon\right\}.$$

Let $\mathcal{F}$ be the family of monotone and $\beta$-Lipschitz functions that maps $[-\bar{M}, \bar{M}]$ to $[-B, B]$ (recall that for all $\sigma(z) \in \mathcal{H}_\epsilon(B, L)$ we can truncate the domain of $\sigma(z)$ to $[-\bar{M}, \bar{M}]$ where $\bar{M} \lesssim \sqrt{\log(B/\epsilon)}$, as shown in Fact A.9, therefore, it is sufficient to consider the function class of monotone $\beta$-Lipschitz functions $u$ that maps from $[-\bar{M}, \bar{M}]$ to $[-B, B]$ that contains the target activation $\sigma$). Then, standard results showed that $|N_2(\epsilon, \mathcal{F}, n)| \leq |N_\infty(\epsilon, \mathcal{F}, n)| = (B/\epsilon)2^{\bar{M}\beta/\epsilon}$ (one can show this via constructing a grid of width $\epsilon/\beta$ on the domain $[-\bar{M}, \bar{M}]$ and another grid of width $\epsilon$ on the codomain, see e.g., Lemma 6, [KKSK11]). Hence, since $\mathcal{U} = \mathcal{F} \circ S^{\mathrm{sol}}$, we have $|N_2(\epsilon, \mathcal{U}, n)| \leq |N_\infty(\epsilon, \mathcal{U}, n)| \lesssim (B/\epsilon)2^{\bar{M}\beta/\epsilon}|S^{\mathrm{sol}}|$. Then, choosing $\alpha = 1/n$ in Fact D.9, and noting that $\sqrt{\mathbf{E}_{\mathbf{x}\sim\widehat{\mathcal{D}}_n}[\varphi^2(\mathbf{x})]} \leq \|\varphi\|_{L_\infty} \leq B$, we obtain

$$\widehat{\mathcal{R}}_n(\mathcal{U}) \lesssim \frac{1}{n} + \sqrt{\frac{1}{n}}\int_{1/n}^B \sqrt{\log(|S^{\mathrm{sol}}|) + (\bar{M}\beta/\epsilon)\log(2) + \log(B/\epsilon)}\,\mathrm{d}\epsilon$$

$$\lesssim \frac{1}{n} + B\frac{\sqrt{\log(|S^{\mathrm{sol}}|)}}{n} + \sqrt{\frac{1}{n}}\int_{1/n}^B \sqrt{(\bar{M}\beta/\epsilon)}\,\mathrm{d}\epsilon + \sqrt{\frac{1}{n}}\int_{1/n}^B \sqrt{\log(B/\epsilon)}\,\mathrm{d}\epsilon$$

$$\lesssim \frac{1}{n} + B\frac{\sqrt{\log(|S^{\mathrm{sol}}|)}}{n} + \sqrt{\frac{\bar{M}\beta B}{n}} \lesssim \sqrt{\frac{\bar{M}\beta B^2 \log(|S^{\mathrm{sol}}|)}{n}}\,.$$

Thus, $\mathcal{R}_n(\mathcal{U}) \lesssim \sqrt{\bar{M}\beta B^2 \log(|S^{\mathrm{sol}}|)/n}$. Recall that we have $|S^{\mathrm{sol}}| = \mathrm{poly}(1/\epsilon, B, L)$, $\bar{M} \leq \sqrt{\log(B/\epsilon)}$, and $\beta = BL/\sqrt{\epsilon}$ (Claim D.7), therefore to guarantee that $\mathcal{R}_n(\mathcal{U}) \leq \sqrt{\epsilon}/\log^{3/2}(n)$ and $B^2 \log(1/\delta)/n \leq \epsilon$, it suffices to choose $n = \Theta(\log(BL/\epsilon)\log(1/\delta)B^3L/\epsilon^{3/2})$. Letting $\delta = 0.01$ completes the proof. $\square$

