# OpenReview forum: "Robustly Learning Monotone Single-Index Models"
_NeurIPS.cc/2025/Conference — NeurIPS 2025 poster_

### Official Review · Reviewer_Kf2k · 2025-06-05

**Clarity:** 3
**Significance:** 2
**Originality:** 3
**Rating:** 4
**Confidence:** 3

**Summary:**

The paper makes progress for the problem on agnostically learning monotone single index model.

**Questions:**

I feel like some ingredients in the new algorithm including the alignment part, the spectral subroutine and the approximation part have all been used for related problems. Can the authors comment on such related work or (boldly) claim these are novel.

**Ethical Concerns:**

["NO or VERY MINOR ethics concerns only"]

**Final Justification:**

The paper seems to contain nice technical contributions but the questions studies is a bit too specific.

**Limitations:**

Yes

**Quality:**

3

**Strengths And Weaknesses:**

The paper shows technical competency in improving on a natural problem.
However, both in terms of the proofs and in terms of the results there is a concern about how incremental the result is.

Obtaining a *constant-factor* approximation for learning *monotone* single-index model for an *unknown* activation was indeed unknown before. But removing any of the highlighted conditions either makes the results invalid or known. So the progress is for a somewhat narrow problem. Also the case where the activation has slightly stronger assumption is studied in ZWDD24.

---

> ### Author Rebuttal · Authors · 2025-07-31
>
> We thank the reviewer for their time and effort in reviewing our paper. We address the reviewer’s points below.
>
>
> **Significance**
> - Response: While the problem setting we consider is specific, it resolves a well-known open question (see, e.g., prior works [WZDD23], [WZDD24], [ZWDD24], [ZWDD25] cited in the main body). This is common in learning theory papers (as ours), that are within the scope of the conference (as per the call for papers). Our work provides the first constant-factor approximation algorithm for agnostically learning SIMs under the Gaussian distribution, where the only assumption on the activation is that it is monotone. Prior work either required significantly stronger assumptions or achieved qualitatively weaker error guarantees (see lines 71-87 and [GGSK23, ZWDD24]). In particular, the [ZWDD24] paper that the reviewer referred to required significantly stronger assumptions on the activation class—that excludes basic activations like biased ReLUs, sigmoids, Linear Threshold Functions, etc.-- whereas our algorithm can deal with *any monotone* activation, a class that includes the activation class considered in [ZWDD24] as a mere subset. Furthermore, the error guarantee obtained in [ZWDD24] is considerably weaker than ours: in [ZWDD24] the final error is $O((\max_{z\geq 0}\sigma’(z))/(\min_{z\geq 0}\sigma’(z)))^4\cdot OPT$ (infinite for activations like the Linear Threshold Function and exponentially large for activations like sigmoid), whereas ours scales as $C\cdot OPT$ with $C$ being an *absolute constant,* independent of *any* problem parameters. Obtaining our result (which, we reiterate, was posed as an open problem in these prior works) requires new algorithmic and analytic insights, as we have illustrated in Section 1.1 of our submission.
>
> **Overlap with “concurrent work” [ZWDD25]**
> - Response: We note that [ZWDD25] is a **prior work**. It appeared on arxiv in February and its conference version recently appeared in COLT 2025. We reiterate that [ZWDD25] posed as an open question the problem we solve (among other prior works, please see our first response bullet above). At the technical level, our algorithm utilizes two key ingredients from that work (the initialization and the error-angle connection). We emphasize that these components alone are by no means sufficient to solve the problem with an unknown activation function, which is the core technical challenge we solve. As we have thoroughly discussed in the manuscript (see e.g., lines 119-131), the gradient alignment property that is crucial to [ZWDD25] no longer holds in our setting, due to the fact that the activation $\sigma$ is no longer a prior known to the learner. Further, a high-accuracy estimate of $\sigma$ is also unavailable due to the agnostic noise. The gradient alignment property is the main structural result in [ZWDD25] that established the convergence guarantee of their algorithm. Regarding the novelty of our techniques, please see below.
>
> **(Question): Novelty of Techniques**
> - Response: We provide a summary in the following paragraphs. A more detailed description is given in our submission (see references to specific lines below).
>
> 1. Our **spectral subroutine**, based on a duality argument, is **novel** and of independent interest. In particular, the spectral subroutine identifies an optimal vector field for the gradient descent steps that is substantially different from prior works, as noted in our manuscript (lines 97-100): ’On a conceptual level, our work makes the case for stepping outside the usual boundaries of gradient based methods and instead looking to directly design a vector field that can be computed from the information given to the algorithm and that carries useful information about the location of target solutions.’ The overview of the spectral subroutine is presented in lines 156-168.
>
> 2. Our optimization method is fundamentally different from prior approaches. Our algorithm **does not need to estimate the activation function** at each step. This is yet another major difference from prior works, which required an alternating approach to estimate the best-fitting activation $u$ on each update and then update the parameter $w$. This is another novel feature of our algorithm, as we have clearly stated in lines 129-131, that ‘...it is unclear whether prior SIM learning frameworks [KKSK11, ZWDD24, HTY25]— alternating between the “best-fit” updates for activation u and gradient-style updates for w—can resolve this issue [of creating alignment between the gradient and the optimal parameter w*]. Hence, our work represents a departure from this seemingly natural approach.’
>
> 3. We are not aware of any prior work that uses a **random walk-based optimization procedure** in a similar context and that is proven to converge in polynomial time (lines 185-197).
>
> [WZDD23] P. Wang, N. Zarifis, I. Diakonikolas, and J. Diakonikolas. Robustly learning a single neuron via sharpness. 40th International Conference on Machine Learning, 2023.
>
> [WZDD24] P. Wang, N. Zarifis, I. Diakonikolas, and J. Diakonikolas. Sample and computationally efficient robust learning of Gaussian single-index models. The Thirty-Eighth Annual Conference on Neural Information Processing Systems, 2024.
>
> [ZWDD24] N. Zarifis, P. Wang, I. Diakonikolas, and J. Diakonikolas. Robustly learning single-index models via alignment sharpness. In Proceedings of the 41st International Conference on Machine Learning, 2024.
>
> [ZWDD25] N. Zarifis, P. Wang, I. Diakonikolas, and J. Diakonikolas. Robustly learning monotone generalized linear models via data augmentation. COLT 2025.
>
> [GGSK23] A. Gollakota, P. Gopalan, A. R. Klivans, and K. Stavropoulos. Agnostically learning single-index models using omnipredictors. In Thirty-seventh Conference on Neural Information Processing Systems, 2023.

---

### Official Review · Reviewer_vBoH · 2025-06-26

**Clarity:** 1
**Significance:** 2
**Originality:** 2
**Rating:** 4
**Confidence:** 3

**Summary:**

This paper tackles the problem of learning generalized linear models with Gaussian inputs $x$ where both the parameters and the activation/link function are unknown. The parameters are assumed to be bounded and $\sigma$ is assumed to belong to the class of monotone, Lipschitz functions (or monotone functions with bounded $2 + \delta$-moment). The goal is to jointly learn both the activation and parameters. The core contribution of this is the development and analysis of an algorithm for learning $(\sigma, w)$. The algorithm is based on the spectrum of a matrix constructed from the data whose eigenvectors are supposed to strongly correlate with the optimal parameters $w^\star$. The algorithm is shown to achieve a constant factor approximation of the optimum from a polynomial number of samples (with respect to the input dimension).

**Questions:**

1. Line 126 : "Thus, to ensure $\|T_{cos \theta} u' - T_{cos \theta} \sigma'\|$  small, it necessitates approximating the low degree coefficients of $\sigma'$ (under adversarial noise and without the knowledge of $\sigma$), imposing formidable technical challenges." Could the authors elaborate on these challenges?
2. On line 174, the author argues that because the labels are adversarially corrupted, estimating $g^\star_w(z)$ as $E[yx^{\perp w} | w \cdot x \in (a,b)]$ is not possible. Could the authors elaborate on why this is the case?

**Ethical Concerns:**

["NO or VERY MINOR ethics concerns only"]

**Final Justification:**

It is not clear from the manuscript that the contributions of this work (in terms of results or techniques) warrant a publication at this stage. I believe it is interesting and of interest, but it would require additional work that is too substantial for a minor camera-ready revision. I don't feel strongly enough to recommend it for acceptance, but I also would not stand in its way.

**Limitations:**

Yes

**Quality:**

2

**Strengths And Weaknesses:**

## Strengths

The main contribution of this work is the core construction of the update direction. It involves two non-trivial steps: posing the choice of update direction as a "correlation maximization problem" and estimating the solution of this problem from the spectrum of a matrix that can be constructed from data.


## Weaknesses

The manuscript is difficult to read and at times informal or unclear. This makes it very difficult to parse the exact contributions of the work and assess their correctness. For instance, the main text contains several instances of notation that are either used before being defined or not defined at all. For instance, Section 1.1 refers to $T_{\rho}$, $x^{\perp w}$, and $\gtrsim$ only defined later in Sec 1.2. This makes the section (which is almost 1/3 of the paper) almost impossible before having read through the full paper (with appendices). These issues are compounded with the continuous use of inline math, including to introduce proof sketches (e.g., line 190-197, which do not even point to a specific result later in the paper). This makes the section virtually unreadable. While I understands that conference papers have space constraints, I do not believe it is an excuse to jeopardize clarity.

The manuscript (and appendices) also repeatedly refer to results in the literature without explicit references, which makes them impossible to assess. This is particularly critical with respect to [ZWDD25] (which contains a plethora of results), e.g.,
    - Line 122-125 : "Even though it is possible to control the L2 distance between $u(z)$ and $\sigma(z)$ by $\theta(w, w^{\star})$ following **similar steps as in [ZWDD24]**, it is unclear how to show that $T_{cos \theta}u'(z)$ is close to $T_{cos \theta}$ \sigma'(z)$ so that [...] and **the arguments of [ZWDD25] can go through**." (which steps/arguments?)
    - Line 162 : "and we **know from [ZWDD25]** that ..." (specific proposition? theorem? proof?)
    - Line 258 : "and transform the regression problem to a robust halfspace learning problem, following **the same procedure** as in [ZWDD25]" (which procedure?)
It is difficult to understand these references without knowing the specific results the manuscript is aluding to.

The comparisons to other works are also not very clear. For instance, this work extends [ZWDD25], which also assumes Gaussian inputs, but only mentions [GGKS23] which works with similar assumptions on the activation function, but *no distributional assumptions* (while labels are confined to [0,1], the labels in the current manuscript are also bounded given the that f is uniformly bounded). It is not straightforward, from the discussions in the paper, to see how the results are considerably weaker in this setting, seen as the sample complexity look comparable and the proportionality constants of the optimality factor are not clearly described. While [GGKS23] does depend on $W$, the radius of the space, it is not clear that this is worst than the constants for Algorithms 1-4. Particularly, since the bound on $\|w\|$ in [GGKS23] is replaced here by a uniform bound $B$ on the activation/link function (I believe that is in fact part of the argument in Remark 2.2 leveraging the homogeneity of $\sigma$). Again, it is hard to adjudicate these matters from the presentations in the manuscript.

On a minor point, it is not clear to which extent the labels are per se "adversarial." Though the agnostic learning setting can be seen as incorporating a form of "adversarial label noise" (as in, e.g., [DKMR22]), the setting of this paper is not fully agnostic since $x$ is assumed to be Gaussian. I imagine that what is meant by adversarial label noise is the fact that no assumptions are made on the conditional $y|x$, but this point should be explicitly addressed in the manuscript.

While the paper might have relevant contributions, it requires considerable improvement in its presentation for the novelty and correctness of the contributions to be evaluated. I therefore cannot recommend the manuscript be published in its current form.

---

> ### Author Rebuttal · Authors · 2025-07-31
>
> We thank the reviewer for their time and effort in reviewing our paper. We address the reviewer’s points below.
>
> **(Weaknesses 1 & 2): Presentation**
> - Response: In the introductory section (Section 1), we intentionally adopted a degree of informality to avoid introducing too many technical details (while focusing on the key novel ideas of our approach). We believe this enhances readability. Regarding the notation pointed out by the reviewer: First we note that this notation is fairly standard in this subfield of theoretical machine learning and specific definitions (such as $T_\rho$, for which a forward reference is provided the first time it is mentioned) are not crucial to explaining the intuition, which is the purpose of Section 1.1. Importantly, we provide all required definitions and notation in Section 1.2 (see page 5). That said, we will revise this section to ensure forward references are included for all the utilized notation.
> Regarding references to prior work: We must respectfully but firmly push back on this assessment.  While in the introduction our goal was to provide high-level context, we provided full technical details and references to all needed results in the supplementary material (see Facts A.8, A.9, B.1, B.2, C.2).
>
> **(Weakness 3 & 4): Comparison to Prior Work**
> - Response: A detailed summary of prior work is given in lines 71-87, where we provide a comparison to [ZWDD25], [GV24], [ZWDD24], and [GGKS23], which are closest to our work. Specifically, regarding the comparison with the work of [GGKS23], we reiterate the point made in our submission: Our algorithm achieves an error of **$C \cdot OPT + \epsilon$**, where $C$ is a **universal constant–independent of any problem parameters**–in the case where the activation function is **unknown**.  This answers a recognized open problem in the literature. For example, this precise question was posed as the main open problem in both [ZWDD24] and [ZWDD25] (see the “Conclusion” section of  [ZWDD24] and the “Conclusions and Open Problems” section of [ZWDD25]).
>
> The prior work of [GGKS23] obtains an error of $O(W) \sqrt{OPT} + \epsilon$, where $W$ is the diameter of the space. The parameter $W$ can depend polynomially on the dimension $d$ and cannot be viewed as a “constant”. Additionally, their error guarantee is also sub-optimal as a function of the parameter $OPT$. We want to emphasize here that this distinction (constant vs non-constant approximation ratio) is not new. A number of prior published works in this field make this distinction clear and pose the question of obtaining a constant factor approximation as an open question. Concrete recent examples include lines 78-82 and page 2 of [ZWDD24] and Appendix A of [ZWDD25]. Earlier references include [DGK+20] and [DKTZ22a], cited in our work.
>
> A point of confusion from the reviewer may be the notion of “boundedness” for the labels. In our setting, even though the labels are bounded (and this is w.l.o.g., as discussed in the paper), the upper bound is not necessarily an absolute constant: instead, the upper bound can scale polynomially with the diameter of the space (hence, also the dimension) and also depend on the desired accuracy. Specifically, for the basic case of a ReLU activation, the effective bound on the magnitude is roughly $W\sqrt{\log(1/\epsilon)}$, where W is the radius of the space and $\epsilon$ an accuracy parameter. In summary, there is a major qualitative difference with the prior work [GGKS23]. It is essential here that our approximation ratio in the error is *independent* of this quantity—and any other problem scaling (i.e., possibly rescaling the labels or the problem space via a change of variable has no impact on the multiplicative approximation guarantee and this is in contrast to prior work).
>
> **(Weakness 4): Regarding adversarial label noise/agnostic noise**
> - Response: While this point was deemed minor by the reviewer, we have the following comments. The terminology we use (i.e., interchangeably using the terms “distribution-specific agnostic learning” and “learning with adversarial label noise”) is standard in the literature; see, e.g., [KKMS05], [ABL17] for the task of learning Linear Threshold Functions and [DKTZ22a] for the task of robustly learning GLMs (a special case of our task). Recall that we work in the distribution-specific setting where the feature vectors are assumed to be drawn from the standard normal (this is a standard and extensively studied regime in the literature). Importantly, no assumptions are made on the labels and our goal is to compete against the best-fit function in the class. An essentially equivalent phrasing is that we start from clean labels (i.e., consistent with the target function in the class) and then an adversary is allowed to arbitrarily corrupt them, subject to the constraint that the total misspecification (with respect to the $L_2$ loss) is bounded above by OPT. See for example lines 28-34 for a brief explanation.
>
> **(Question 1) Approximation of $\sigma’$:**
> - Response: The challenge in approximating the derivative $\sigma'$ is significant because this task is equivalent to learning the polynomial approximation of $\sigma$. The latter task requires sample complexity $d^{m/2}$, where $m$ is the degree of the polynomial approximation (which may be large). (We note here that learning with respect to the $L_2$-norm is equivalent to finding a function that has a closely approximating polynomial expansion.) On the other hand, the sample complexity of our algorithm is $O(d^2)$, which does not depend on the degree of the polynomial expansion of the unknown activation.
>
> **(Question 2) Estimation of** $g_w^* (z)$
> - Response: The core challenge is that an adversary can manipulate labels at specific (individual) points without affecting the overall loss, which makes naive estimation impossible. Since we cannot efficiently sample at an exact point $w \cdot x = z$ (as this is an event with probability measure zero), a natural choice would be to sample from a small band or interval. This approach would indeed work (efficiently) in the realizable case, when the labels $y$ are equal to $\sigma(w^{\ast} \cdot x)$, for a monotone activation $\sigma$. Unfortunately, this fails in the agnostic/adversarial label noise setting we consider for the following reason: an adversary can place extreme noise at one point $z$. As this is a single point, the measure is zero, and this manipulation does not actually change the overall loss. However, if we try to estimate the value of $ g_w^* (z) $ by averaging $yx$ over the interval containing $z$, we will never sample this specific point $z$ and we will never observe the extreme value. This makes our local estimate of $g_w^*(z)$  inaccurate (with error we can make arbitrarily high by increasing the magnitude of the noise).
>
> Instead of trying to estimate the corrupted function directly, our approach is to find a function that performs at least as well as the unknown activation, specifically $T_{\cos\theta}\sigma'$. A key property of the Ornstein-Uhlenbeck semi-group is that it is a **smoothing operator**. This means that the function $T_{\cos\theta}\sigma'$ is smooth and, therefore, nearly constant within small intervals. This property justifies approximating it with a **piecewise-constant function**, as the error introduced will be controllably small. By estimating the average value over a band, we thus obtain a piecewise-constant estimate. Our analysis shows that this approach is sufficient because this approximation is good enough to effectively compete with the smooth target, successfully overcoming the adversary's potential manipulations.
>
> [KKMS05] Adam Tauman Kalai, Adam R. Klivans, Yishay Mansour, and Rocco A. Servedio. Agnostically learning halfspaces. FOCS 2005
>
> [ABL17] Awasthi, P., Balcan, M. F., and Long, P. M. The power of localization for eﬃciently learning linear separators with noise. Journal of the ACM, 2017
>
> [DKTZ22a] I. Diakonikolas, V. Kontonis, C. Tzamos, and N. Zarifis. Learning a single neuron with adversarial label noise via gradient descent. COLT 2022
>
> [GGKS23] A. Gollakota, P. Gopalan, A. R. Klivans, and K. Stavropoulos. Agnostically learning single-index models using omnipredictors. NeurIPS 2023
>
> [ZWDD24] N. Zarifis, P. Wang, I. Diakonikolas, and J. Diakonikolas. Robustly learning single-index models via alignment sharpness. In Proceedings of the 41st International Conference on Machine Learning, 2024.
>
> [ZWDD25] N. Zarifis, P. Wang, I. Diakonikolas, and J. Diakonikolas. Robustly learning monotone generalized linear models via data augmentation. COLT 2025
>
> [GV24] A. Guo and A. Vijayaraghavan. Agnostic learning of arbitrary ReLU activation under
> Gaussian marginals. COLT 2025

---

> ### Comment · Reviewer_vBoH · 2025-08-05
>
> I appreciate the responses of the authors and they have addressed several of my concerns (in quite some detail in fact). I reiterate and clarify below two points that I believe they should consider in reviewing their manuscript, since I felt some resistance in their response in making modifications to address these concerns (I use feel to underscore that I may be mistaken since this is not a factual argument). I will not keep the paper hostage over this as I trust the willingness of the authors to incorporate constructive suggestions in their manuscript.
>
> **Presentation**: I thank the authors for explaining the rationale in their presentation choices. Granted, some presentation aspects are a matter of style over which we can disagree. Others, however, are conventions in mathematical writing that severely hurt readability. So I encourage the authors to review Section 1 as its informality currently hinders rather than enhance readability. I also strongly encourage them to shy away from assuming that notation is "fairly standard" in a "subfield of theoretical machine learning" and consider writing their paper for a broader theoretical research audience. The reader should not read half of the manuscript before being introduced to notation (5 pages, until Section 1.2).
>
> **Prior work**: Perhaps my concerns were not clear in my first comment. The authors do provide comparison to prior work in their original manuscript, but I would not evaluate this comparison as transparent:
>
> - While the current algorithm does achieve a constant factor approximation as opposed to [GGKS23], it does so under stronger assumption (Gaussian vs distribution-free). That is not a criticism of the data generation assumption, but it is hardly an "improvement" seen as it holds in a different setting. It could well be, for instance, that the Gaussian assumption is necessary to obtain this constant-factor approximation.
> - While the problem is interesting, the manuscript and the response does not provide sufficient evidence that it is a "recognized open problem in the literature" (as they point out to two papers from the same authors: [ZWDD24] and [ZWDD25]).
> It is particularly important to make this things clear so as to position this paper in the broader literature (which is an issue also raised by Reviewer Kf2k).

---

> > ### Author Response · Authors · 2025-08-06
> > **Response to reviewer**
> >
> > We thank the reviewer for engaging in this discussion. We respond to the reviewer’s points below.
> >
> > Regarding presentation, in addition to forward references, we will add informal definitions of the notation to improve the flow of the technical overview, as the reviewer suggested.
> >
> > **Comparison to Prior Work:**
> >
> > The comparison to prior work in our submission is given in lines 71-85, following the formal definition of the problem we study and the statement of our main theorem.
> >
> > *Regarding the reviewer’s first point:*
> >
> > “While the current algorithm does achieve a constant factor approximation as opposed to [GGKS23], it does so under stronger assumption (Gaussian vs distribution-free). That is not a criticism of the data generation assumption, but it is hardly an "improvement" seen as it holds in a different setting. It could well be, for instance, that the Gaussian assumption is necessary to obtain this constant-factor approximation.”
> >
> >
> > We agree with the reviewer that our result is not an “improvement” over [GGKS23]. *In fact, nowhere in the submission or our first rebuttal did we make such a statement.* Instead, our result is framed as a generalization of [GV24] (which handled the case of a general ReLU) and [ZWDD25] (which handled a *known* monotone activation) under the same distributional assumptions. Both these papers appeared in COLT’25.
> >
> > *Some additional relevant remarks:*
> >
> > * Under the distributional assumptions of [GGKS23], it is computationally hard to achieve *any* constant factor approximation [see line 50].
> >
> > * Even under the Gaussian distribution and a ReLU activation, relaxing the error guarantee to some constant factor approximation is necessary, if we are shooting for a polynomial time algorithm [see line 49].
> >
> > * Despite significant effort by the learning theory community, even for a single Linear Threshold function (potentially biased), the only distribution for which we have a constant-factor approximation (in polynomial time) is the Gaussian.
> > Hence extending our results beyond Gaussian distributions in open even for this very special case of a *known* activation. (Please also see response to Reviewer Be7r.)
> >
> > *Regarding the reviewer's second point:*
> >
> > “While the problem is interesting, the manuscript and the response does not provide sufficient evidence that it is a "recognized open problem in the literature" (as they point out to two papers from the same authors: [ZWDD24] and [ZWDD25]). It is particularly important to make this things clear so as to position this paper in the broader literature (which is an issue also raised by Reviewer Kf2k).”
> >
> >
> > Please see lines 51-57 of our submission, where we describe the context of this work.
> > We explain these points in more detail below:
> >
> > * The pioneering work of [KKSK11] developed the “Isotron” algorithm for learning Single-Index Models for the class of monotone and Lipschitz activations. The motivation for considering this class of activations is in part due to the fact that most activations used as gates in deep neural networks are in fact monotone.
> > The algorithm of [KKSK11] succeeds for any distribution on the unit ball. However, it requires the crucial assumption that the labels are either realizable or have zero mean random noise.
> >
> > * Since that initial work, it has been a folklore open question in this community to generalize the [KKSK11] result to the agnostic/adversarial label noise setting. The line of works we cite on Lines 54-55 (note that these are works by several different subsets of authors and it is a partial list) all consider and make progress on *special cases* of this question.
> >
> > * In particular, all of them study the agnostic learning task for subsets of monotone activations.
> > With the exception of [GGKS23], all other works in the list make strong distributional assumptions (like Gaussianity)
> > with the goal of achieving a constant factor approximation. The reason is that an approximation factor that scales with the dimension or the diameter of the space is not particularly useful.
> >
> > * The reason that the reference to the general open problem (that we answer) is explicitly listed on “two papers from the same coauthors” (papers in ICML’24 and COLT’25) is mainly due to the fact that over the past couple of years the community felt that there is hope to resolve the general question as a next step. Again, until COLT’25, it was not even known how to obtain a constant-factor approximation for a general biased ReLU—let alone for an arbitrary unknown monotone function.
> >
> > * We do acknowledge that for a reviewer outside this area, it may not be immediately obvious that this is a “recognized” open question. The aforementioned discussion attempts to explain the context. That said, there are several experts in this area that the AC could reach out to for that purpose, assuming this remained in doubt.

---

### Official Review · Reviewer_Be7r · 2025-07-01

**Clarity:** 4
**Significance:** 3
**Originality:** 4
**Rating:** 6
**Confidence:** 2

**Summary:**

This paper provides the first computationally efficient algorithm for robustly learning a Single-Index Model (SIM) of the form $f(x) = \sigma(w \cdot x)$ when the activation function $\sigma$ is unknown but belongs to a broad class of monotone functions, the input distribution is Gaussian, and the labels are subject to adversarial noise. The algorithm achieves a constant-factor approximation to the optimal squared loss, resolving a significant open problem in the algorithmic theory of SIMs. The main technical innovation is a novel optimisation framework that moves beyond gradient-based methods, instead using a carefully constructed vector field, identified via spectral methods, to guide the search for the optimal parameter vector.

**Questions:**

- Your spectral method relies on constructing a moment matrix $M_w$. To what extent are the key properties required for your analysis (specifically, the spectral gap between the top eigenvalue and the rest of the spectrum) tied to the Gaussianity of the input distribution? Could this spectral alignment framework be extended to other distributions, such as isotropic log-concave distributions?

- The algorithm's final stage relies on a random walk to overcome the sign ambiguity of the identified eigenvector, requiring $poly(1/\epsilon, B, L)$ repetitions to guarantee success with high probability. Could you comment on the practical magnitude of this number of repetitions? Furthermore, is there a potential path to "de-randomise" this component?

- What are the primary obstacles to extending this spectral alignment approach to robustly learning a multi-index model? Would the top eigenvectors of a similarly constructed moment matrix still capture the subspace of interest, or would the interactions between different index directions fundamentally complicate the spectral picture?

- Given the work is purely theoretical, what specific insights do you believe an empirical study would provide? For example, how might the constants hidden in your asymptotic analysis behave in a finite-sample setting, and what is your hypothesis on the algorithm's robustness to minor deviations from the Gaussian data assumption?

**Ethical Concerns:**

["NO or VERY MINOR ethics concerns only"]

**Final Justification:**

I have read the authors' rebuttal and the other reviews. My assessment of the paper remains unchanged.

​This paper makes a significant theoretical contribution by providing the first polynomial-time algorithm for robustly learning single-index models with unknown monotone activations, a long-standing open problem. The technical approach is novel and the exposition is clear.

​The authors have satisfactorily addressed my main concerns. Their justification for focusing on the Gaussian distribution is reasonable, it tackles a known open problem, and extensions are non-trivial even for much simpler cases. While I maintain that the lack of experiments is a limitation, I agree with the authors that it is not a prerequisite for acceptance for a purely theoretical work of this caliber.

**Limitations:**

The primary limitations of this work are threefold. First, its theoretical guarantees are confined to the standard Gaussian input distribution, which restricts its direct applicability. Second, the algorithm's high polynomial complexity and multi-stage, randomised nature raise questions about its practical feasibility despite its theoretical efficiency. Finally, the work is entirely theoretical, with no empirical validation provided in either the main paper or the supplementary material. The absence of experiments means there is no insight into the algorithm's finite-sample performance, the practical values of the constants in the bounds, or its robustness to even minor deviations from the strict distributional assumptions.

**Quality:**

3

**Strengths And Weaknesses:**

*Disclaimer: I am not an expert of the field and the proofs in the SM have not been checked.*

# Strengths

- It successfully provides the first polynomial-time, constant-factor approximation algorithm for agnostically learning SIMs with a very general class of monotone activations. This class includes all monotone Lipschitz functions and discontinuous ones, which covers many cases of practical and theoretical interest.

- The main technical idea is novel. The authors' departure from standard gradient-based optimisation and designed a custom moment matrix $M_w$ that can be analysed with spectral methods to guide the gradient.

- For a paper with high technical density, the exposition is remarkably clear.

# Weaknesses

- The entire analysis is contingent on the assumption that the input data $x$ follows a standard multivariate Gaussian distribution. This is a strong limitation, as the analytical tools used (e.g., properties of the Ornstein-Uhlenbeck semigroup, Gaussian integration by parts) are highly specific to this setting. The authors rightly note that generalising beyond Gaussian marginals is a major open question, even for simpler models. As it stands, the algorithm's direct applicability to non-Gaussian, real-world data is limited.

- While the algorithm is proven to be "efficient" in the theoretical sense (i.e., runs in polynomial time), its complexity appears prohibitive for practical use. The sample complexity is $N=d^{2}poly(B,L,1/\epsilon)$, and the overall algorithm involves multiple nested subroutines, including an initialisation procedure that grids over a parameter space and a main loop that performs a random walk, requiring the core routine to be repeated many times to ensure success.

- The work is entirely theoretical and lacks any experimental validation. This absence prevents an assessment of the algorithm's finite-sample performance, the practical magnitude of the constants hidden in the theoretical bounds, and its robustness to even minor deviations from the strict Gaussian assumption.

---

> ### Author Rebuttal · Authors · 2025-07-31
>
> We thank the reviewer for their effort in evaluating our submission and for their insightful questions. We are  encouraged by the reviewer’s positive assessment, particularly on the **significance** of the paper, the **novelty** of the techniques, and the **clarity** of our presentation. Below, we respond to the reviewer’s comments and questions.
>
> **Gaussian Assumption**:
> - Response: The reviewer is correct that our algorithmic result makes essential use of the Gaussian distribution. As pointed out in the submission, the question we resolve (robustly learning monotone SIMs under Gaussian marginals within a constant factor) has been a recognized open problem in the field (see, e.g., the prior works [WZDD23], [WZDD24], [ZWDD24], [ZWDD25] all cited from the main body). As the reviewer noted and is stated in our paper, to achieve a constant-factor approximate learner beyond Gaussian data, one would first need to make progress on (very) special cases of the problem when the activation is known (e.g., for a general Linear Threshold Function or a general, potentially biased, ReLU).
>
> **Practicality and Experimental Evaluation**
> - Response: As the reviewer notes, our work is in the area of learning theory. The primary focus was to provide the first polynomial (in the dimension $d$ and $1/\epsilon$) sample and time algorithm for our learning task. As such, we did not optimize the absolute constants or the constant powers of $1/\epsilon$ in the resulting complexity bounds. That said, we are confident that a more careful analysis may yield a significantly improved upper bound.  While experimental evaluation of our algorithm would be interesting, we note that the NeurIPS call for papers invites purely theoretical works (for which experimental evaluation is not required).
>
> **(Question 1): To what extent are the key properties required for your analysis tied to the Gaussianity of the input distribution?**
> - Response: Our work crucially relies on the Gaussianity of the feature vectors in two main aspects:
>
> 1. **Spectral Subroutine:** To derive our main structural result—that the top eigenvector of our designed matrix aligns with the target direction—we relied on Stein’s lemma and the properties of the Ornstein-Uhlenbeck (OU) semi-group. It is not immediately clear how to extend this result to non-Gaussian distributions.
>
> 2. **Initialization Subroutine:** This part of our algorithm transforms our learning problem into the task of agnostically learning halfspaces under Gaussian marginals. Unfortunately, even for this special case, there is no known efficient algorithm that achieves a constant-factor approximation beyond the Gaussian case; see the conclusion section, line 391-393.
>
> We are hopeful that similar ideas might be adapted to non-Gaussian distributions and leave this as an important direction for future work.
>
> **(Question 2): Practical magnitude of number of repetitions in random walk and potential derandomization.**
> - Response: The total number of iterations incurred by this step is on the order of $(L/\epsilon)^c$ for some universal constant c, which we did not optimize. The upper bound in our analysis relies on the worst case (monotone) function in the class. That said, for natural subsets of activations (e.g., activations with heavy tails or exponential tails) it is possible to obtain faster convergence rates. Regarding de-randomization, it is currently unclear to us how to circumvent the random walk approach. One potential approach would be to determine the correct sign of the eigenvector by testing which direction yields a smaller error. However, obtaining such an accurate estimate for a general monotone function under agnostic noise is a significant challenge in itself.
>
> **(Question 3): What are the primary obstacles to extending this spectral alignment approach to robustly learning a multi-index model?**
> - Response: Extending our approach to multi-index models (MIMs) is challenging for a number of reasons. The primary obstacle is that the interactions between different index directions would fundamentally complicate the spectral field of the moment matrix. For the case of SIMs that we study, it turns out that second moment information suffices for our purposes. For the more general setting of MIMs, the second moment matrix would likely no longer suffice to capture the subspace of interest, and leveraging higher-order moment information would become necessary. Very recent work has provided some theoretical justification of this observation for learning MIMs, via SQ and low-degree polynomial lower bounds; see, e.g., [DIKZ25].
>
>
> **(Question 4): Potential Insights of Empirical Study**
> - Response:  In an empirical study, we expect (1) better performance than what is predicted by the theoretical bounds, as our focus was on getting a polynomial time algorithm and constant factor approximation, which was the primary open question we were aiming to address. In particular, we expect the algorithm to perform well even with a smaller sample size and to have a reasonably small constant approximation ratio. (2) The algorithm should remain robust to minor deviations from Gaussian data, as the data augmentation itself should have a "correcting effect" for such deviations, making the data "look and behave more Gaussian." (3) Furthermore, an empirical study could investigate the algorithm's performance on certain classes of non-monotone activations. While our theoretical guarantees for key components like the initialization and the error analysis crucially rely on the monotonicity assumption, the core optimization routine itself is more general. Observing strong empirical performance even on some non-monotone functions would be a compelling result, potentially motivating future theoretical work to relax this assumption.
>
>
> [WZDD23] P. Wang, N. Zarifis, I. Diakonikolas, and J. Diakonikolas. Robustly learning a single neuron via sharpness. 40th International Conference on Machine Learning, 2023.
>
> [WZDD24] P. Wang, N. Zarifis, I. Diakonikolas, and J. Diakonikolas. Sample and computationally efficient robust learning of gaussian single-index models. The Thirty-Eighth Annual Conference on Neural Information Processing Systems, 2024.
>
> [ZWDD24] N. Zarifis, P. Wang, I. Diakonikolas, and J. Diakonikolas. Robustly learning single-index models via alignment sharpness. In Proceedings of the 41st International Conference on Machine Learning, 2024.
>
> [ZWDD25] N. Zarifis, P. Wang, I. Diakonikolas, and J. Diakonikolas. Robustly learning monotone generalized linear models via data augmentation. COLT 2025
>
> [DIKZ25] Ilias Diakonikolas, Giannis Iakovidis, Daniel M. Kane, and Nikos Zarifis
> Robust Learning of Multi-index Models via Iterative Subspace Approximation 2025, arXiv:2505.21475

---

> > ### Comment · Reviewer_Be7r · 2025-08-01
> >
> > After considering the other reviewers’ comments and the authors’ responses, I find no substantial new information that would alter my evaluation. I therefore maintain my original score. While I continue to believe that empirical validation could further strengthen the paper, I agree with the authors that it is not essential for acceptance.

---

### Comment · Area_Chair_Whd6 · 2025-08-01
**Author-reviewer discussions**

Dear Reviewers,

Thank you for your time in writing the constructive comments. The authors have provided detailed responses. Would you please read these replies and see whether your comments have been addressed (e.g., the assumption on the data and the comparison with related work)?Thank you very much for your cooperations.

Best regards, AC

---

### Decision · Program_Chairs · 2025-09-17

**Decision:**

Accept (poster)

**Comment:**

This paper proposes a computationally efficient algorithm to learn single index models $f(x)=\sigma(w\cdot x)$, where both the activation function $\sigma:\mathbb{R}\to \mathbb{R}$ and $w\in \mathbb{R}^d$ are learned from data with adversarial label noises. The primary contribution of the paper is the development of the first polynomial-time, constant-factor approximation algorithm for agnostically learning SIMs with a broad class of monotone Lipschitz activation functions, addressing an open problem in the literature. Additionally, the paper introduces several novel techniques, including the use of a spectral subroutine to identify an optimal vector field for gradient descent steps and a random walk-based optimization procedure.

Meanwhile, the paper has some limitations: it assumes the input data follow a Gaussian distribution, the algorithm's complexity may be high for practical applications, and it lacks empirical validation. Reviewers also suggest improvements in the presentation for enhanced clarity. Overall, the reviewers are generally satisfied with the authors' responses, acknowledging that while the results are interesting, substantial efforts are needed to improve the clarity of the presentation and the articulation of the main contributions.